# Parent-of-origin effects on complex traits in up to 236,781 individuals

Robin J. Hofmeister[1,2,3,4 ✉], Théo Cavinato[1,2], Roya Karimi[5], Adriaan van der Graaf[1,2], Fanny-Dhelia Pajuste[4], Jaanika Kronberg[4], Nele Taba[4], Estonian Biobank research team*, Reedik Mägi[4], Marc Vaudel[5,6], Simone Rubinacci[7], Stefan Johansson[5,8], Lili Milani[4,9], Olivier Delaneau[10] & Zoltán Kutalik[1,2,3 ✉]

Parent-of-origin effects (POEs) occur when the effect of a genetic variant depends on its parental origin[1]. Traditionally linked to genomic imprinting, POEs are believed to occur due to parental conflict over resource allocation to offspring, resulting in opposing parental influences[2]. Despite their importance, POEs remain underexplored in complex traits, owing to the lack of parental genomes. Here we present an approach to infer the parent of origin of alleles without parental genomes, leveraging interchromosomal phasing, mitochondrial and X chromosome data, and sex-specific crossover in siblings. Applied to the UK Biobank, this enabled parent-of-origin inference for up to 109,385 individuals. Genome-wide association study scans for 59 complex traits and over 14,000 protein quantitative trait loci contrasting maternal and paternal effects identified over 30 POEs and confirmed more than 50% of known associations. More than one third of these showed opposite parental influences, especially for traits related to growth (for example, IGF1 and height) and metabolism (for example, type 2 diabetes and triglyceride levels). Replication in up to 85,050 individuals from the Estonian Biobank and 42,346 offspring from the Norwegian Mother, Father and Child Cohort Study (MoBa) validated 87% of testable associations. Overall, our findings highlight the contribution of POEs to complex traits and support the parental conflict hypothesis, providing compelling evidence for this understudied evolutionary phenomenon.

Genome-wide association studies (GWASs) traditionally identify additive genetic effects, assuming that the phenotypic effect depends on the number of copies of a given allele, regardless of their parental origin. However, some sequence variants can have distinct phenotypic effects depending on whether they are maternally or paternally inherited, a phenomenon known as parent-of-origin effects (POEs). Traditionally, POEs have been linked to genomic imprinting, in which only one parental gene is expressed depending on its parental origin. This selective expression is believed to arise from an evolutionary conflict over parental investment, with paternally inherited alleles promoting offspring growth at the cost of maternal resources, whereas maternally inherited alleles prioritize resource conservation for future reproduction[2]. This antagonism can lead to opposite parental effects at imprinted loci, particularly for traits related to growth, metabolism and energy storage. However, as most studies have identified only isolated POEs, evidence supporting the conflict hypothesis across diverse traits remains scarce. Although the conflict hypothesis provides a compelling rationale for imprinting, the presence of POEs at non-imprinted loci also suggests more complex

mechanisms[3,4]. Some of these effects may also reflect parental rearing or environmental influences, rather than strictly direct genetic mechanisms. This perspective calls for the broadening of investigations beyond classical imprinted regions and exploring alternative pathways leading to POEs.

Studying POEs requires large cohorts with parent-of-origin (PofO) information, traditionally derived from parental genomes[5]. An alternative approach uses close relatives as surrogate parents to estimate parental haplotypes[6], which allows PofO inference when combined with genealogical data[7]. However, large-scale biobanks often lack parental genomic data and detailed genealogies[8]. Building on the surrogate parent concept, our previous work introduced a method leveraging X chromosome sharing to identify maternal relatives without genealogical information[4], substantially increasing the available sample size. However, this method was limited to male participants, constraining statistical power.

To overcome these challenges, we developed a novel multistep approach for inferring the PofO of alleles in many additional pedigree situations. First, we partitioned genomes into maternal and paternal

[1]Department of Computational Biology, University of Lausanne, Lausanne, Switzerland. [2]Swiss Institute of Bioinformatic (SIB), University of Lausanne, Lausanne, Switzerland. [3]University Center for Primary Care and Public Health, Lausanne, Switzerland. [4]Estonian Genome Centre, Institute of Genomics, University of Tartu, Tartu, Estonia. [5]Mohn Center for Diabetes Precision Medicine, Department of Clinical Science, University of Bergen, Bergen, Norway. [6]Department of Genetics and Bioinformatics, Health Data and Digitalization, Norwegian Institute of Public Health, Oslo, Norway. [7]Institute for Molecular Medicine Finland, Helsinki, Finland. [8]Department of Pediatrics, Haukeland University Hospital, Bergen, Norway. [9]Estonian Biobank, Institute of Genomics, University of Tartu, Tartu, Estonia. [10]Regeneron Genetics Center, Tarrytown, NY, USA. *A list of authors and their affiliations appears at the end of the paper. ✉e-mail: robin.j.hofmeister@gmail.com; zoltan.kutalik@unil.ch

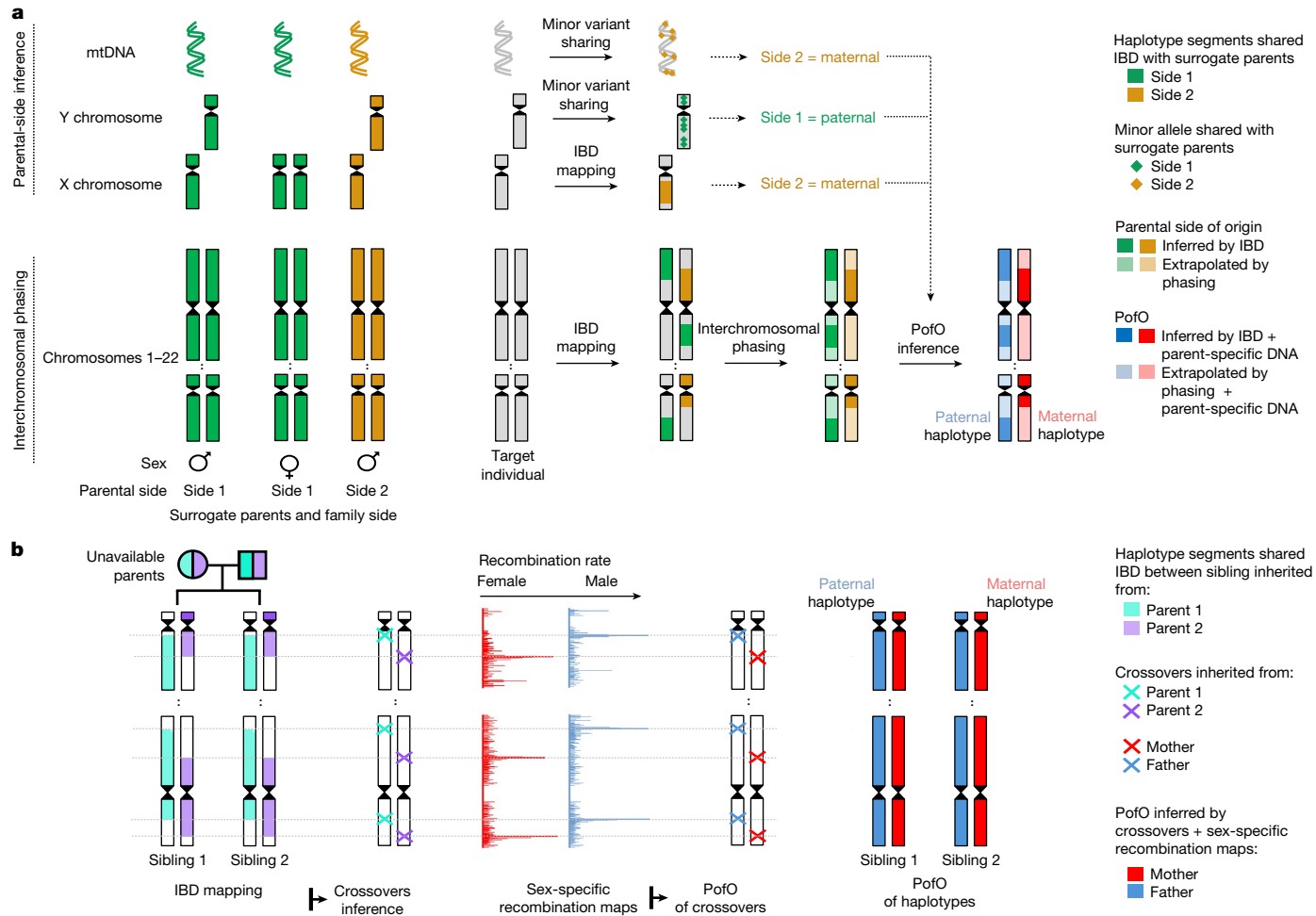

**Fig. 1 | Interchromosomal phasing and overview of PofO methods.**
**a**, Interchromosomal phasing and PofO inference. We initially clustered relatives (second, third and fourth degree) into surrogate parent groups, segregating the relatives into one family side (green) versus the other family side (orange). We then combined two steps to infer the PofO of a focal individual: on the one hand, we leveraged IBD sharing with surrogate parent groups to perform interchromosomal phasing – for instance, green IBD tracks were forced to align on the first haplotype, simultaneously correcting for intrachromosomal phasing errors (see Supplementary Note 1) – and on the other hand, we examined genetic similarities on the X chromosome and mtDNA between the focal individual and its surrogate parents, to assign surrogate parent groups to a parental side. Finally, we deduced the PofO of interchromosomally phased haplotypes from the parental side of the surrogate parent in IBD (see Methods for details). **b**, To determine the PofO of siblings, we first inferred crossover events using IBD haplotypes. We then overlapped inferred crossovers with sex-specific genetic maps to derive the likelihood of a crossover originating from the mother or the father. This subsequently allowed us to deduce the PofO of the haplotype carrying the crossover.

haplotypes through statistical interchromosomal phasing, using inferred surrogate parents (Fig. 1a). We then determined the PofO of these haplotypes by integrating several predictors: X chromosome sharing in males and mitochondrial DNA (mtDNA) whole-genome sequences (Fig. 1a). In addition, we assigned the PofO of crossovers inferred in siblings using sex-specific recombination maps[9,10] (Fig. 1b).

Applied to the UK Biobank, we successfully inferred the PofO of alleles for 109,385 white British individuals – a fourfold increase in sample size compared with our previous work[4]. For replication, we extended our analysis to the Estonian Biobank (up to 85,050 individuals) and to 42,346 offspring from the Norwegian Mother, Father and Child Cohort Study (MoBa). We identified over 30 POEs, affecting mainly traits linked to resource allocation, such as growth and metabolism, and protein levels. A substantial fraction exhibited bipolar effects, characterized by opposite parental influences, consistent with the conflict hypothesis. These findings highlight the potential evolutionary origin of POEs and their importance in shaping complex traits and diseases, offering new insights into their broader biological role.

## Method overview

We developed a comprehensive, multistep strategy to infer the PofO of sequence variants in large-scale biobanks. A brief summary of the methods is outlined below, followed by a detailed analysis of POEs.

### Interchromosomal phasing

We used kinship estimates, age and sex to identify parent–offspring trios, duos and siblings. For more distant relatives, we grouped them into two parental sides[4], without distinguishing maternal from paternal side. This was possible for 274,525 UK Biobank participants (Supplementary Fig. 1). In subsequent sections, we refer to these family groups as 'surrogate parents'. Among them, 2,141 individuals also had parental genomes, forming a 'validation cohort' to assess the concordance of surrogate parent-based PofO inference and those based on actual parents (Supplementary Fig. 2). We then used identity-by-descent (IBD) information from surrogate parents to conduct interchromosomal phasing, effectively identifying variant sets co-inherited within and across chromosomes (Fig. 1a). This showed 99% interchromosomal phasing accuracy (Supplementary Figs. 3 and 4a–c), further improving

traditional intrachromosomal phasing (Supplementary Fig. 5 and Supplementary Note 1). In subsequent analyses, we refer to this dataset as 'interchromosomally phased'.

## PofO inference for 109,385 UK samples

To infer the PofO of each complete parental haplotype set simultaneously, we used two key approaches depending on the available surrogate parents. For individuals with second- to fourth-degree surrogate parents, we examined X chromosome and mtDNA inheritance patterns (Fig. 1a and Supplementary Figs. 6–10). This allowed us to determine whether the used surrogate parent (and all IBD segments shared on other chromosomes) belonged to the maternal (or paternal) side. For sibling pairs, we inferred crossover positions and probabilistically assigned their PofO using sex-specific recombination maps[9,10] (Fig. 1b), prioritizing interchromosomally phased data that improved accuracy by aggregating crossover events across all chromosomes (see Methods; Supplementary Figs. 11–13).

Combined with inference made from available parental genomes (see Methods), we obtained PofO estimates for 286,666 individuals. Of note, over 40% of these individuals ($n$ = 123,716) had high-confidence estimates (PofO probability ≥ 0.99; Supplementary Fig. 14a,b), with 94% concordance across the three predictors (X chromosome, mtDNA and crossovers). We then restricted our analysis to 109,385 white British individuals. A detailed breakdown of sample selection, filtering and inference yield is provided in Supplementary Fig. 15.

Using the validation cohort with known PofO, we estimated an overall PofO inference accuracy of 97.94% (Supplementary Figs. 4d and 14c), with over 80% of individuals exceeding 99% accuracy. Compared with our previous approach[11], this novel inference led to 4.7 times larger effective sample size, while maintaining high accuracy (prediction concordance with the previous approach > 97%; Supplementary Note 2).

Next, we used the resulting PofO-resolved genotype data to perform PofO-specific association scans on 59 selected complex traits (Supplementary Table 1) and over 14,000 PofO-unaware protein quantitative trait loci (pQTLs). To assess POEs, we selected a wide range of complex traits: biomarkers, morphological measures, traits associated with resource allocation, such as growth-related traits (for example, height and fat-free mass), metabolic traits (for example, glucose and lipid levels, and basal metabolic rate) and traits related to energy storage (for example, regional fat percentages and BMI), allowing us to test the conflict hypothesis. Our analysis used three main approaches: (1) we first examined imprinted regions with established imprinting potential, (2) we then focused on regions exhibiting additive (PofO-unaware) associations with the trait (referred to as 'additively associated regions' in subsequent sections), and (3) we finally scanned for POEs genome-wide to identify POEs without prior assumptions.

## Standards for POE detection

Declaring POEs often lacked a standardized statistical framework in the past. Many studies have defined POEs on the basis of parent-specific associations, but this can be misleading due to unequal statistical power. To address this, we propose the POE differential test $P$ value ($P_D$) to formally assess whether maternal and paternal effects differ significantly. In the next sections, we systematically required $P_D$ to meet a strict significance threshold, ensuring that POEs met the same high standards as additive GWAS associations.

For studies lacking $P_D$ but providing parent-specific estimates, we computed $P_D$ (see Methods). Adopting this standard ensured rigour, reproducibility and comparability across POE studies.

In addition, we introduce a classification of POE patterns based on the relative magnitudes and directions of paternal and maternal effects (see Methods; Supplementary Fig. 16).

## Replicating known POEs

We first replicated POEs previously reported in the literature[4,5,7,12–16]. Out of 39 previously published independent POEs that could be tested for replication in our cohort (same variant–trait pair), we retained 18 with discovery $P_D$ < 10[−3]. We successfully replicated 8 of them (44.4%) at $P_D$ < 0.05/18. Of the eight POEs reported for birth weight, none was replicated in our cohort (see Supplementary Note 4). For the other traits, we successfully replicated 80% (8 of 10) of the reported POEs (Supplementary Table 2).

## POEs within imprinted regions

To identify POEs within imprinted regions, we restricted our POE analysis to variants located within a 500-kb window of known imprinted loci[7] (Supplementary Table 3). We scanned 59 complex traits for association (Supplementary Table 1) and identified 11 significant POEs in which $P_D$ met our phenome-wide imprinted region-focused significance threshold (0.05/(16,574 × 48); Table 1; see Methods). In addition, we identified 16 'suggestive' POEs in which $P_D$ met our imprinted region-focused significance threshold, uncorrected for the number of independent traits (0.05/16,574; Table 1; see Methods).

A systematic analysis of lead POE variants across traits revealed two key patterns. POEs at pleiotropic loci on chromosomes 7, 11 and 20 often involved growth-related and metabolism-related traits (Fig. 2a). Furthermore, we found a significant enrichment (OR = 5.35, $P$ = 0.018; Supplementary Note 19) of POEs on growth and metabolism traits compared with other trait categories (Supplementary Table 1). In addition, a large proportion of these POEs (19 of 27 total, 7 of 15 independent SNP–trait pairs) exhibited opposite parental effects, in which alleles inherited from one parent increased, but when inherited from the other parent, decreased the trait value (Fig. 2b). This pattern, known as bipolar dominance[1,17], highlights the distinct and opposing contributions of the two parental alleles. Of note, such effects are often missed in traditional additive GWASs, in which the opposing parental effects cancel out in heterozygotes (Supplementary Fig. 16).

In the following sections, we describe key POE loci identified using our imprinted region-focused approach, showcasing the diversity and pleiotropy of POEs across multiple regions and traits. A detailed description of all associations is available in Supplementary Note 6.

### Bipolar POE variant at 7q32.2

We identified a bipolar effect of rs62471721 on triglyceride levels at the 7q32.2 imprinted region (Fig. 3a,b). Functionally, this variant acts as an expression QTL (eQTL) for several imprinted genes in adipose tissue under an additive model[18], including *KLF14* (maternally expressed) and *MEST* (paternally expressed). Such parent-specific pleiotropy provides a plausible mechanistic explanation for the observed bipolar effect: assuming that both *KLF14* and *MEST* influence triglyceride levels, the maternal allele of rs62471721 preferentially affects *KLF14*, whereas the paternal allele alters *MEST*, resulting in opposite directions of effect depending on the parental origin of the allele (Fig. 3c). Two additional variants, also eQTLs for maternally and paternally expressed genes[18], showed suggestive bipolar POEs on high-density lipoprotein cholesterol (HDL-C) and sex hormone-binding globulin (SHBG). Colocalization analysis suggested shared causal POE variant for the three traits at this locus (Supplementary Note 6.1).

In the same imprinted region, but in low linkage disequilibrium ($r^2$ < 0.001), we identified two independent suggestive maternal effects on hip circumference (Table 1 and Extended Data Fig. 1). SNP rs6467315 (Extended Data Fig. 6a,b), an eQTL for the maternally expressed *KLF14* gene, was in high linkage disequilibrium ($r^2$ > 0.9) with previously reported maternal-effects variant on TG and HDL-C (rs12154627)[5], and on type 2 diabetes (T2D) (rs4731702)[7], all of which showed moderate

## Table 1 | Significant POEs identified in this study

| Chr. | Pos. | SNP ID | A1 | β-Paternal | s.e. Paternal | P paternal | β-Maternal | s.e. Maternal | P maternal | P differential | Trait | Locus | Scan | POE |
|---|---|---|---|---|---|---|---|---|---|---|---|---|---|---|
| 11 | 1914139 | rs576603 | T | −0.027 | 0.004 | $8.53 \times 10^{-11}$ | 0.005 | 0.004 | $2.03 \times 10^{-1}$ | $3.86 \times 10^{-8}$ | Standing height[a,n] (c) | H19, IGF2 | I | P |
| 11 | 2813322 | rs143840904 | T | −0.023 | 0.015 | $1.37 \times 10^{-1}$ | −0.183 | 0.016 | $9.13 \times 10^{-32}$ | $5.21 \times 10^{-13}$ | Standing height[a,k] | KCNQ1 | IAG | M |
| 11 | 2040272 | rs77708343 | G | −0.055 | 0.011 | $2.21 \times 10^{-7}$ | 0.034 | 0.010 | $1.07 \times 10^{-3}$ | $2.18 \times 10^{-9}$ | Standing height[a,k] | H19, IGF2 | I | B |
| 11 | 2040272 | rs77708343 | G | −0.048 | 0.011 | $1.25 \times 10^{-5}$ | 0.035 | 0.011 | $1.33 \times 10^{-3}$ | $7.53 \times 10^{-8}$ | BMR[n] | H19, IGF2 | I | B |
| 11 | 2040272 | rs77708343 | G | −0.052 | 0.011 | $1.28 \times 10^{-6}$ | 0.033 | 0.011 | $2.06 \times 10^{-3}$ | $1.80 \times 10^{-8}$ | Leg fat-free mass[a,n] | H19, IGF2 | I | B |
| 11 | 2040272 | rs77708343 | G | −0.044 | 0.010 | $3.08 \times 10^{-5}$ | 0.033 | 0.010 | $1.23 \times 10^{-3}$ | $1.63 \times 10^{-7}$ | Whole-body water mass[n] | H19, IGF2 | I | B |
| 11 | 2041348 | rs78507815 | T | −0.031 | 0.009 | $7.03 \times 10^{-4}$ | 0.033 | 0.009 | $1.86 \times 10^{-4}$ | $3.61 \times 10^{-7}$ | Trunk fat-free mass[n] | H19, IGF2 | I | B |
| 11 | 1920285 | rs4264135 | G | −0.033 | 0.006 | $1.83 \times 10^{-7}$ | 0.014 | 0.006 | $3.31 \times 10^{-2}$ | $2.22 \times 10^{-7}$ | Urate level[n] | H19, IGF2 | I | P |
| 11 | 1998031 | rs170102 | A | −0.041 | 0.006 | $5.96 \times 10^{-11}$ | 0.026 | 0.006 | $3.19 \times 10^{-5}$ | $5.98 \times 10^{-14}$ | Cystatin C level[a,n] | H19, IGF2 | IG | B |
| 11 | 2003944 | rs217215 | G | 0.030 | 0.006 | $2.53 \times 10^{-7}$ | −0.024 | 0.006 | $2.68 \times 10^{-5}$ | $4.94 \times 10^{-11}$ | Creatinine level[a,n] | H19, IGF2 | IG | B |
| 11 | 1703564 | rs4417225 | T | 0.029 | 0.008 | $9.26 \times 10^{-5}$ | −0.028 | 0.008 | $2.04 \times 10^{-4}$ | $7.40 \times 10^{-8}$ | Glucose level[t] | H19, IGF2 | I | B |
| 11 | 1702929 | rs10838787 | A | 0.136 | 0.027 | $3.88 \times 10^{-7}$ | −0.091 | 0.027 | $7.55 \times 10^{-4}$ | $1.87 \times 10^{-9}$ | T2D[a,k] | H19, IGF2 | I | B |
| 11 | 1702929 | rs10838787 | A | 0.050 | 0.007 | $2.34 \times 10^{-14}$ | −0.028 | 0.007 | $1.80 \times 10^{-5}$ | $2.77 \times 10^{-17}$ | HbA1c level[a,t] | H19, IGF2 | IG | B |
| 11 | 2858295 | rs2299620 | T | −0.011 | 0.018 | $5.22 \times 10^{-1}$ | −0.129 | 0.017 | $1.50 \times 10^{-13}$ | $2.23 \times 10^{-6}$ | HbA1c level[t] | KCNQ1 | IA | M |
| 7 | 130009312 | rs10239342 | G | 0.022 | 0.007 | $1.36 \times 10^{-3}$ | −0.024 | 0.007 | $4.15 \times 10^{-4}$ | $1.87 \times 10^{-6}$ | SHBG level[n] | KLF14, MEST | I | B |
| 7 | 130016470 | rs62471721 | A | −0.023 | 0.007 | $9.70 \times 10^{-4}$ | 0.033 | 0.007 | $1.57 \times 10^{-6}$ | $6.51 \times 10^{-9}$ | Triglyceride level[a,n] | KLF14, MEST | I | B |
| 7 | 130017940 | rs4731690 | G | 0.019 | 0.006 | $1.97 \times 10^{-3}$ | −0.022 | 0.006 | $2.76 \times 10^{-4}$ | $1.92 \times 10^{-6}$ | HDL-C level[n] | MEST | I | B |
| 7 | 130463192 | rs6467315 | G | 0.001 | 0.006 | $9.02 \times 10^{-1}$ | −0.042 | 0.006 | $5.81 \times 10^{-11}$ | $2.20 \times 10^{-6}$ | Hip circ.[t] | KLF14 | I | M |
| 7 | 130400698 | rs3847104 | A | −0.016 | 0.010 | $1.01 \times 10^{-1}$ | 0.056 | 0.010 | $1.94 \times 10^{-8}$ | $3.81 \times 10^{-7}$ | Hip circ.[n] (c) | KFL14 | I | M |
| 20 | 57216538 | rs80116540 | G | 0.035 | 0.010 | $4.38 \times 10^{-4}$ | −0.043 | 0.010 | $2.04 \times 10^{-5}$ | $3.39 \times 10^{-8}$ | Arm fat %[a,n] | GNAS | I | B |
| 20 | 57216538 | rs80116540 | G | 0.037 | 0.010 | $2.65 \times 10^{-4}$ | −0.037 | 0.010 | $2.27 \times 10^{-4}$ | $2.06 \times 10^{-7}$ | Body fat %[n] | GNAS | I | B |
| 20 | 57226079 | rs6026426 | G | 0.045 | 0.012 | $3.08 \times 10^{-4}$ | −0.040 | 0.012 | $1.37 \times 10^{-3}$ | $1.29 \times 10^{-6}$ | Trunk fat %[n] | GNAS | I | B |
| 20 | 57484934 | rs3730173 | T | 0.036 | 0.010 | $3.18 \times 10^{-4}$ | −0.035 | 0.010 | $3.95 \times 10^{-4}$ | $5.32 \times 10^{-7}$ | Leg fat %[n] | GNAS | I | B |
| 6 | 144274210 | rs12528876 | C | −0.036 | 0.009 | $2.58 \times 10^{-5}$ | 0.021 | 0.009 | $1.60 \times 10^{-2}$ | $2.96 \times 10^{-6}$ | IGF1 level[n] | PLAGL1 | I | B |
| 16 | 3412861 | rs7188903 | G | −0.012 | 0.007 | $1.03 \times 10^{-1}$ | 0.041 | 0.007 | $5.35 \times 10^{-8}$ | $6.14 \times 10^{-7}$ | IGF1 level[n] | ZNF597, NAA60 | I | M |
| 15 | 25983333 | rs146982369 | G | 0.101 | 0.032 | $1.88 \times 10^{-3}$ | −0.106 | 0.032 | $9.99 \times 10^{-4}$ | $2.59 \times 10^{-6}$ | Total protein level[n] | ATP10A | I | B |
| 14 | 101185187 | rs59228823 | C | −0.010 | 0.008 | $2.04 \times 10^{-1}$ | −0.092 | 0.008 | $3.87 \times 10^{-34}$ | $1.92 \times 10^{-14}$ | Platelet count[a,k] | MEG3 | IAG | M |
| 3 | 169482335 | rs2293607 | C | −0.122 | 0.008 | $2.26 \times 10^{-55}$ | −0.068 | 0.008 | $5.36 \times 10^{-18}$ | $5.70 \times 10^{-7}$ | TS ratio[n] | TERC | A | PA |
| 4 | 164012901 | rs11100479 | T | 0.079 | 0.008 | $1.82 \times 10^{-22}$ | 0.022 | 0.008 | $6.61 \times 10^{-3}$ | $6.25 \times 10^{-7}$ | TS ratio[n] | NAF1 | A | P |
| 5 | 1285974 | rs7705526 | A | 0.038 | 0.007 | $2.09 \times 10^{-7}$ | 0.106 | 0.007 | $1.27 \times 10^{-47}$ | $5.82 \times 10^{-11}$ | TS ratio[k] | TERT | AG | M |

Far left column indicates the relevant chromosome (Chr.). Second column indicates position (Pos.) in the human reference genome hg19. 'SNP ID' (third column) indicates the variant identifier (rsID). 'A1' (fourth column) indicates the assessed allele. β-, s.e. and P (fifth through tenth columns) indicate effect size, standard error and P value, respectively, computed using REGENIE[38] (two-sided test). In the 'Trait' column, a superscripted 'a' indicates a POE identified using the imprinted region-focused approach that is also robust with correction for the number of phenotypes tested (see Methods); superscripted 'n', 'k' and 't' indicate, respectively, novel POE, known POE and tagging POE (that is, novel associations that share substantial similarities with previously reported type 2 diabetes associations); and '(c)' indicates a POE identified in a conditional analysis. Trait abbreviations: BMR, basal metabolic rate; T2D, type 2 diabetes; HbA1c, glycated haemoglobin; SHBG, sex hormone-binding globulin; HDL-C, high-density lipoprotein cholesterol; Hip circ., hip circumference; IGF1, insulin-like growth factor 1; TS ratio, relative leucocyte telomere length. 'Scan' (penultimate column) indicates the approach: A, additively associated region focus; G, genome-wide scan; I, imprinted region focus. Far right column indicates the POE pattern (see Methods): B, bipolar; M, maternal; P, paternal; PA, paternal asymmetric.

association with our lead POE variant (Supplementary Note 6.2). SNP rs3847104 (Extended Data Fig. 1c,d), located 62 kb away and not in linkage disequilibrium with rs6467315 or known lipid-associated POEs ($r^2 < 0.001$), also exhibited an independent maternal-only effect on hip circumference when conditioned on the lead variant (Extended Data Fig. 1 and Supplementary Note 6.2).

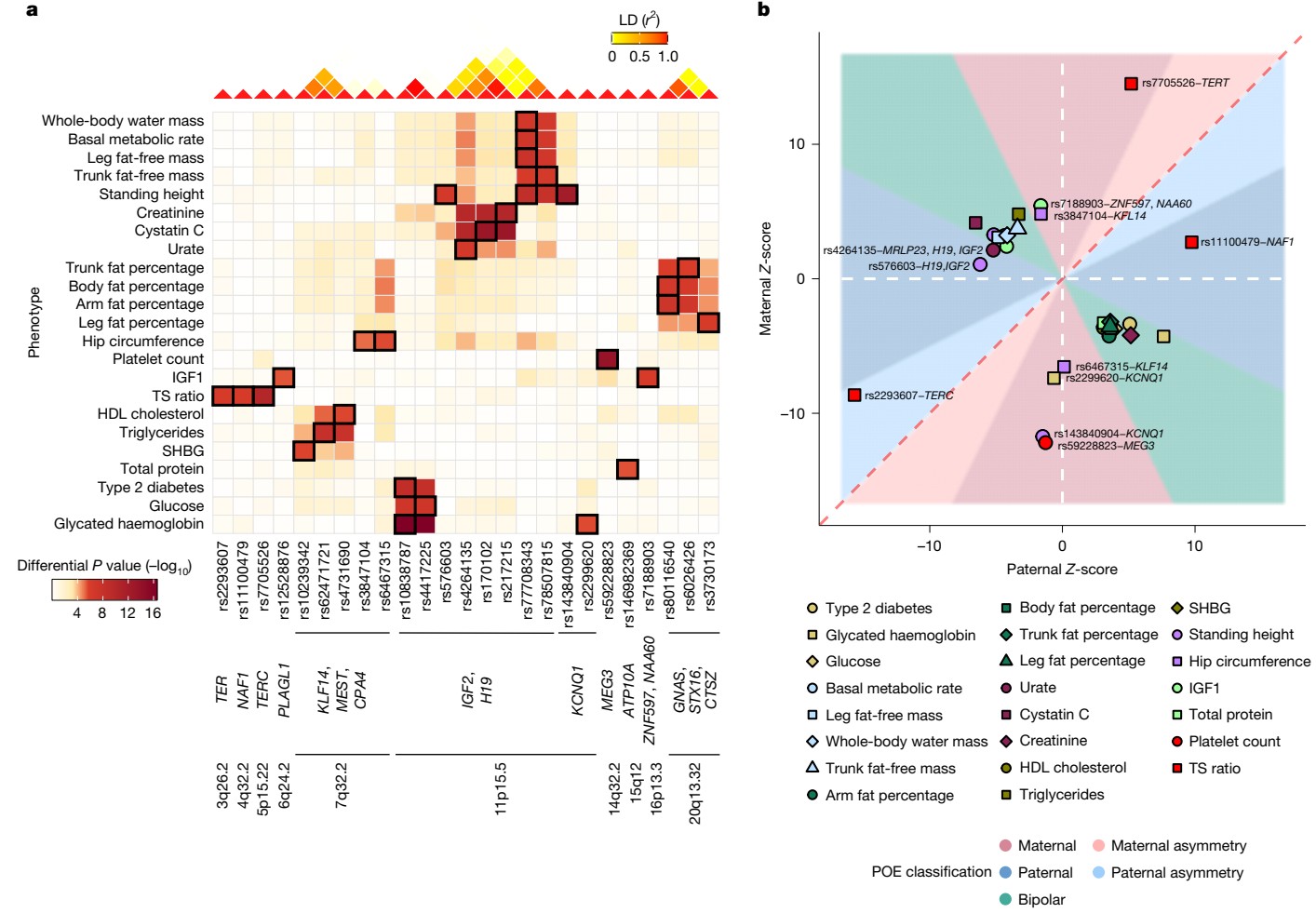

**Fig. 2 | Significant POEs. a**, Heatmap summarizing the differential *P* value for all significant POEs identified in this study, computed using REGENIE[38] (two-sided test). Columns correspond to genetic variants ordered by genetic position and annotated with the genes and chromosome bands, whereas rows represent phenotypes. The colour intensity represents the magnitude of the differential *P* value, with darker shades indicating stronger differential effects. Cells with black rectangles indicate significant POEs identified in this SNP–trait pair and reported in Table 1. The linkage disequilibrium (LD) heatmap (top) shows the linkage disequilibrium (*r*²) between the variants. **b**, Scatter plot illustrating the parental *Z*-scores for all significant POEs identified in this study and reported in Table 1. Each point represents a significant POE, coloured and shaped by phenotype. The dashed red line represents the line of equality, and the dashed white lines represent zero values for paternal and maternal *Z*-scores, respectively. Areas are filled according to the POE classification, also shown in Table 1 (see Methods for details on the classification). Labelled points correspond to POEs classified as paternal or maternal effects. Unlabelled points correspond to bipolar POEs.

## Novel and pleiotropic POEs at *H19*/*IGF2*

We identified multiple POEs at the *H19*/*IGF2*-imprinted region, including three independent effects on standing height (Table 1 and Extended Data Fig. 2a–d). A novel association involved the paternal rs576603 T allele being linked to reduced height (Extended Data Fig. 2e). This variant is a known splice QTL for the maternally expressed *H19* gene and an eQTL for the paternally expressed *IGF2* gene[18].

Two additional independent POEs − rs143840904 (maternal effect) and rs77708343 (bipolar effect) − were in strong linkage disequilibrium (*r*² > 0.6) with previously reported height-associated POEs[12,15,16] (Extended Data Fig. 2f,g; see the section 'Replicating known POEs'). Although rs143840904 and rs77708343 influenced both sitting and standing height, rs576603 was specific to standing height (Supplementary Note 6.3). Of note, rs77708343 also showed suggestive POEs on metabolic traits, whereas the other variants were height specific (Extended Data Fig. 3a and Supplementary Note 6.4).

At the same locus, we identified two novel significant bipolar POEs of rs170102 and rs217215 on cystatin C and creatinine levels, respectively,

and a novel suggestive paternal effect of rs4264135 on urate levels (Table 1, Extended Data Fig. 3a and Supplementary Note 6.5).

Given the extensive pleiotropy at 11p15.5 (Fig. 2 and Extended Data Fig. 3a), we performed a colocalization analysis, revealing two distinct clusters of traits probably influenced by separate causal variants: rs217215 for cystatin C and creatinine levels (*H*₄ = 0.92) and rs77708343 for standing height and metabolic traits (*H*₄ ≥ 0.92), but only partial colocalization with urate levels (*H*₄ = 0.27–0.47; Extended Data Fig. 3b).

## Revisiting POEs on T2D

We detected significant bipolar POEs of rs10838787 on T2D at 11p15.5, where the paternal A allele increased T2D risk (OR_pat = 1.14, 95% confidence interval (CI) of 1.08–1.21), but the maternal A allele was protective (OR_mat = 0.91, 95% CI of 0.86–0.96; Extended Data Fig. 4a,b).

Although the imprinting status of this locus is not yet established, our lead POE variant is located within 350 kb of a well-characterized imprinted gene cluster and is in high linkage disequilibrium (*r*² > 0.9) with rs2334499, a variant previously reported to exhibit a similar bipolar effect on T2D[7]. Conditional analysis suggested rs10838787

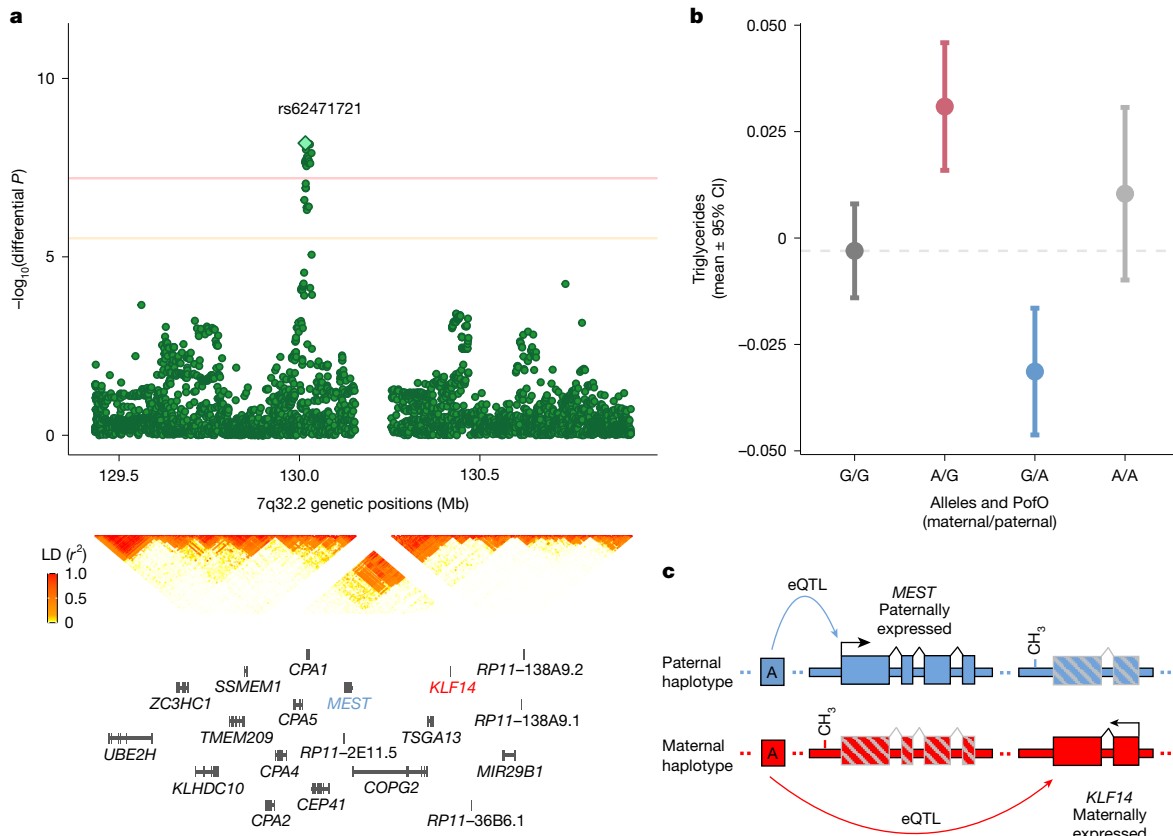

**Fig. 3 | Significant bipolar POE on triglycerides. a**, Differential GWAS shows the association strength ($-\log_{10}$($P$ value), $y$ axis) against the genomic position ($x$ axis) on triglycerides. Each point represents a genetic variant. The diamond indicates the variant with independent POEs found in this study and listed in Table 1. The red and orange lines represent the significance threshold used to identify significant and suggestive significant POEs, respectively (see Methods). The linkage disequilibrium pattern (middle) ranges from 0 (white) to 1 (red). Gene positions (bottom) are shown along the genomic positions ($x$ axis). The horizontal lines show the gene start and end positions. The vertical lines show the exons start and end positions. Gene names are shown below the corresponding gene coordinates. Imprinted genes mentioned in the main text are highlighted in red (maternally expressed) and blue (paternally expressed). **b**, Genotype and PofO of alleles of the lead POE variant ($x$ axis) effects on triglyceride levels ($y$ axis). The dots show mean values, and the error bars show 95% CI computed as mean ± 1.96 × s.e. The grey dashed line represents the mean value of individuals carrying no alternative allele (that is, genotype 0). The dot and bar colours indicate maternal heterozygotes (red), paternal heterozygotes (blue) and homozygotes (dark and light grey). **c**, We hypothesize that bipolar POEs at this locus may arise because SNP rs62471721 is an eQTL for two imprinted genes. Specifically, the paternal copy of the A allele affects expression of the paternally expressed gene *MEST*, whereas the maternal copy influences expression of the maternally expressed gene *KLF14*.

to be a more likely causal variant in the UK Biobank cohort (Supplementary Note 6.6). Thus, our result refines rather than introduces a novel association at this locus, and provides robust confirmation of a bipolar POE on T2D, an effect that had not been reliably replicated since its initial report[7]. This locus potentially ranks among the SNPs WITH the largest effect on T2D with the odds of T2D being 1.25 times higher (95% CI of 1.16–1.35) when the A allele of rs10838787 is paternally versus maternally inherited. For comparison, one of the most influential genetic factors for T2D, an intronic variant to *TCF7L2* (rs7903146), has an OR ≈ 1.4 (ref. 19).

At the same locus, we also observed a bipolar POE of rs10838787 on glycated haemoglobin (HbA1c), a well-established diagnostic biomarker for T2D[20] (Extended Data Fig. 4c), and a suggestive bipolar POE of rs4417225 on glucose, a variant in high linkage disequilibrium ($r^2 > 0.99$) with rs10838787 (Extended Data Fig. 4d and Supplementary Note 6.6). Although the POEs on T2D and HbA1c were consistent across sexes, our sex-specific analysis (Supplementary Note 9) revealed a significant male-only POE on glucose ($P_{D\text{males}} = 4.7 \times 10^{-9}$ and $P_{D\text{females}} = 0.19$; Extended Data Fig. 5 and Supplementary Table 11).

We identified an additional suggestively significant maternal-only effect of rs2299620 on HbA1c, intronic to *KCNQ1* and in high linkage disequilibrium ($r^2 = 0.5$) with a previously reported T2D-risk variant (rs2237892)[7] (Extended Data Fig. 4c). Consistently, we also found a moderate protective effect of the maternal T allele of rs2299620 on T2D ($OR_{\text{mat}} = 0.69$, 95% CI of 0.60–0.79). Conditional analysis confirmed rs2299620 as the more likely driver of the POE on HbA1c levels (Supplementary Note 6.6).

### 20q13.32 POEs on fat distribution

At 20q13.32, we identified a novel bipolar POE of rs80116540 on arm fat percentage. This variant is an eQTL for the bidirectionally imprinted gene *GNAS* in blood (under an additive model)[21], a gene producing both maternally and paternally derived proteins[22], supporting the bipolar POE that we observed at this locus. Additional suggestive bipolar POEs on diverse body fat compartments indicated that this locus has a broader influence on fat distribution (Supplementary Note 6.7).

### POEs in additively associated regions

We identified six significant POEs within additively associated regions ($P_D < 0.05/1{,}812 = 2.75 \times 10^{-5}$; Table 1, Fig. 2 and Supplementary Note 7; see Methods). Two were novel associations with telomere length

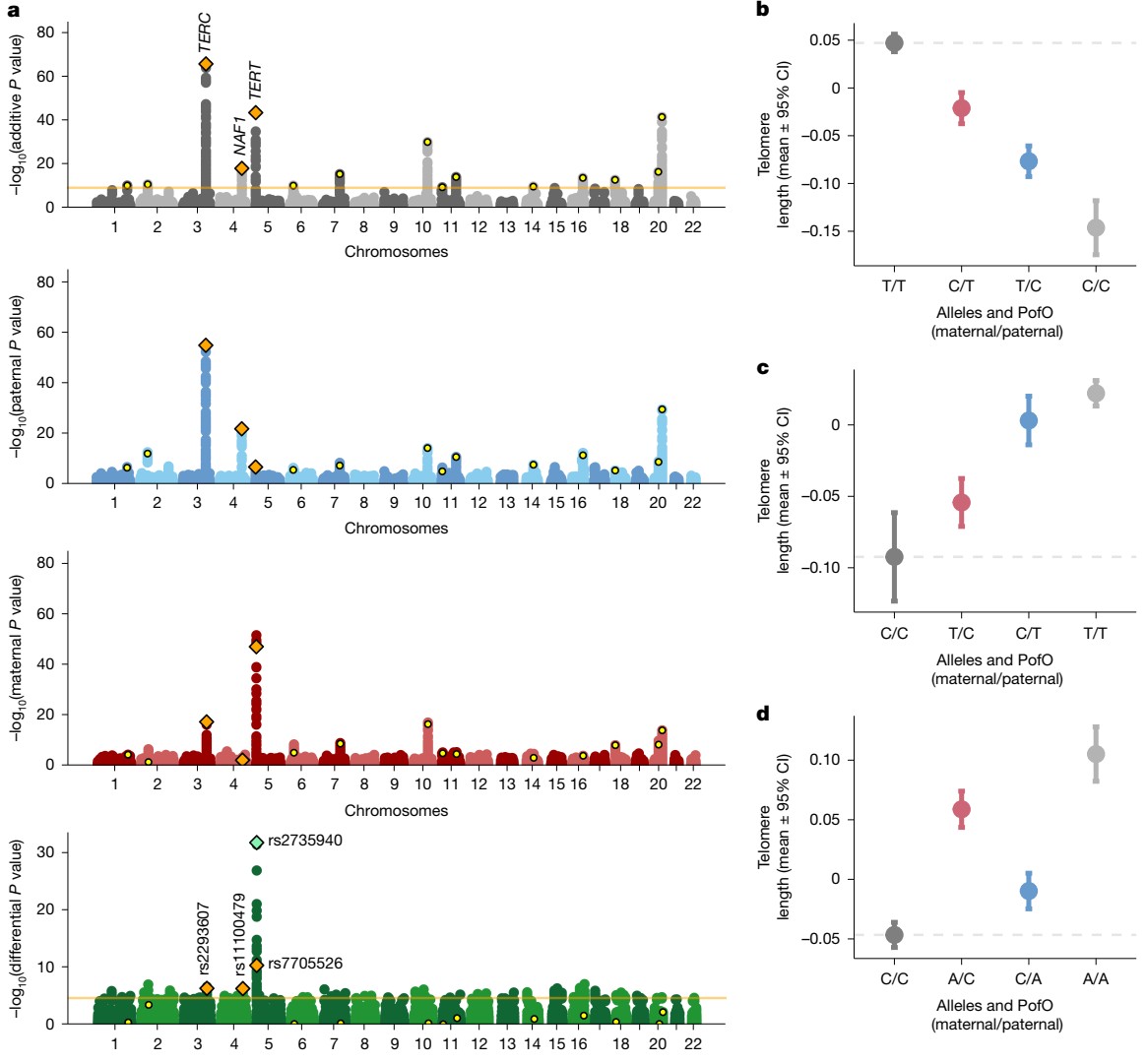

**Fig. 4 | PofO associations on telomere length. a**, Genome-wide associations for additive (grey), paternal (blue), maternal (red) and differential (green) effects represented as association strength ($-\log_{10}(P$ value), $y$ axis) against the genomic position ($x$ axis). Each point represents a genetic variant. The yellow dots indicate additive genome-wide significant variants (those then tested for POE). The orange diamonds indicate variants exhibiting significant POEs among the lead additive effect variants. The light green diamond indicates the lead POE variant identified in the genome-wide scan. The orange lines indicate significant thresholds ($1 \times 10^{-9}$ for the additive effects and $2.75 \times 10^{-5}$ for POEs).

**b–d**, Genotypes and PofO of allele effects ($x$ axis) on telomere length ($y$ axis) at the *TERC* locus (rs2293607; **b**), *NAF1* locus (rs11100479; **c**) and *TERT* locus (rs7705526; **d**). The dots show mean values, and the error bars show 95% CI computed as $\beta \pm 1.96 \times$ s.e. and derived from $n = 109{,}385$ individuals. The grey dashed lines represent the mean value of individuals carrying no alternative allele (that is, genotype 0). The dot and bar colours indicate maternal heterozygotes (red), paternal heterozygotes (blue) and homozygotes (dark and light grey).

(Fig. 4): the C allele of rs2293607 (*TERC*) decreased telomere length more strongly when paternally inherited (Fig. 4a,b), whereas the T allele of rs11100479 (*NAF1*) increased telomere length when paternally inherited (Fig. 4a,c).

We also found a maternal effect of rs7705526 (*TERT*) on telomere length, in moderate linkage disequilibrium ($r^2 = 0.27$) with rs2735940, previously implicated in telomere length in our earlier work[11]. Our genome-wide analysis suggested rs2735940 as the more likely lead POE variant at this locus (see Supplementary Note 8), which was confirmed by conditional analysis ($P_{Dc} = 1.8 \times 10^{-23}$ and $P_{Dc} = 0.1$ for rs2735940 and rs7705526, respectively).

The three remaining associations, namely, with platelet count, standing height and HbA1c level, were in high linkage disequilibrium with associations reported in our imprinted regions-focused analysis (Supplementary Note 7).

## POEs in early life

We tested whether loci found to have POE on adult height and obesity-related traits exhibit POE on these traits at early life. For this, we used self-reported childhood data in the UK Biobank and longitudinal height and BMI measurements in 42,346 children (with genetic data available for both parents) from 6 weeks to 8 years of age (11 time points; Supplementary Table 8) in MoBa[23] (Supplementary Note 10). Two of the three adult height loci, including rs77708343 and our novel *IGF2* variant rs576603, showed significant POEs on 'comparative height size at age 10' in the UK Biobank (Supplementary Table 7 and Supplementary Note 10.1). The bipolar POE of rs77708343 was also associated with infant height across all time points in MoBa (Extended Data Fig. 6a, Supplementary Table 9 and Supplementary Note 10.2.1). These effects mirrored those seen in adulthood,

suggesting that POEs influence early growth trajectories with long-lasting effects.

SNP rs6467315 showed POEs on infant BMI in MoBa at seven time points. The maternal G allele was associated with higher BMI in infancy but lower BMI and hip circumference in adulthood, with the effect gradually reversing over time (Extended Data Fig. 6b and Supplementary Note 10.2.2). This probably explains the absence of POEs at 10 years of age in the UK Biobank. A similar pattern on BMI has been previously reported for rs287621 ($r^2 = 0.36$ with rs6467315)[23]. UK Biobank conditional analysis supported rs6467315 as the primary signal (Supplementary Note 10.2.2).

We further confirmed that the observed POEs on early-life height and BMI were not attributable to maternal untransmitted alleles (Extended Data Fig. 6 and Supplementary Tables 9 and 10), indicating genuine imprinting effects instead of maternal rearing.

## POE SNP heritability

We used linkage disequilibrium score regression to estimate SNP heritability ($h^2$), which represents the proportion of phenotypic variability explained by additive SNP effects. We decomposed $h^2$ into two orthogonal components: average parental (additive) effects and parental differential effects (see Methods). Hence, $h^2_{POE}$ represents the variance explained by differential parental effects at heterozygous sites, beyond that of the additive model.

For most traits, $h^2_{POE}$ was modest compared with additive effects (Extended Data Fig. 7a and Supplementary Note 11.1). For six traits (IGF1 level, triglyceride level, eosinophil count, lymphocyte count, birth weight and basophil count), we observed nominally significant enrichment of $h^2_{POE}$ within imprinted regions compared with the rest of the genome.

Further exploration of maternal versus paternal $h^2$ showed similar contributions for most traits, with a few notable differences for T2D, arm fat percentage, birth weight, glucose level and basophil count (Extended Data Fig. 7b,c and Supplementary Note 11.2).

## POE pQTLs

We investigated POEs at the protein level in the UK Biobank plasma proteomics data, focusing on (1) variants identified as POEs for complex traits in this study, and (2) known additive pQTLs[24] (Supplementary Note 12).

For complex trait POEs, we could test eight variant–protein pairs involving genes *CPA4* and *GNAS*. One significant ($P_D < 0.05/8$; Supplementary Note 12.1) POE pQTL was detected: the G allele of rs4731690, exerting a bipolar effect on HDL-C, increased *CPA4* protein levels when paternally inherited. No significant POEs were observed for *GNAS*.

Among 14,287 previously reported pQTLs, we tested 10,611 directly and assessed proxy variants ($r^2 > 0.8$) for the remainder. We identified four significant POE pQTLs ($P_D < 3.5 \times 10^{-6}$; Supplementary Note 12.2) for *DLK1*, *CPA4*, *ADAM23* and *PER3* (Extended Data Table 1 and Extended Data Fig. 8). POEs at *DLK1* and *CPA4* aligned with their known imprinting: paternal effects and maternal effects, respectively. For *CPA4*, the lead POE pQTL differed from the top additive pQTL. POEs at *ADAM23* and *PER3*, both outside known imprinted regions, showed paternal-specific effects, with lead POE signals showing moderate linkage disequilibrium with those reported in additive pQTL studies (Supplementary Note 12.2).

## Replication in the Estonian Biobank

To replicate the POEs identified in our study, we followed a similar multistep approach in the Estonian Biobank cohort[25], allowing us to examine POEs in up to 85,050 individuals (Supplementary Figs. 17–22 and Supplementary Note 13).

Owing to the limited overlap of phenotypes in the Estonian Biobank cohort, we were able to test only seven POEs identified in our study using the exact same variants and seven additional associations, utilizing linkage disequilibrium proxies. Out of these 14 associations, we successfully replicated 10 ($P_D < 0.05/14 = 3.35 \times 10^{-3}$), with two additional associations at nominal significance (Supplementary Table 12).

## Discussion

We have introduced a novel multistep approach to infer the PofO of haplotypes, enabling large-scale PofO-informed analysis. Applied to the UK Biobank, we inferred allelic PofO for 109,385 white British individuals. To replicate our findings, we used 85,050 individuals in the Estonian Biobank, and a further 42,346 offspring from MoBa. This combined dataset of 236,781 individuals allowed systematic discovery and replication of POEs across diverse traits. In total, we identified over 30 POEs, many of which had been previously undetected.

Our approach overcomes key limitations of traditional methods that rely on parental genomes, known genealogies, or X chromosome sharing in male individuals. By incorporating mtDNA from whole-genome sequencing, we extend inference to female individuals, increasing the sample size and enabling sex-specific analysis. We also introduced a sibling-based crossover inference method, which is applicable to SNP array-based datasets. A major strength of our approach is its scalability to biobank-sized cohorts, allowing PofO inference across all ages. Unlike trio-based studies, in which parent–offspring trios are typically recruited together, resulting in a cohort predominantly composed of young offspring, our method enables analysis of late-onset phenotypes, such as T2D, illustrated by our replication and refinement of a POE that went unreplicated for over 15 years. Limitations include dependence on the relatedness structure of the cohort (Supplementary Note 14). Indeed, accuracy was higher in the Estonian Biobank (average of 12 relatives per individual compared with 1.6 in the UK Biobank). Consanguinity also influences inference quality, although we found no evidence that consanguinity reduces accuracy (Supplementary Note 14). Finally, as we only inferred parental transmitted alleles, we cannot distinguish between parental rearing and true imprinting effects (Supplementary Note 15).

Given the comparable power of POE and additive GWAS analyses (Supplementary Note 16), the lower number of POEs probably reflects their scarcity. To boost statistical power, we applied two SNP pre-filtering strategies to reduce multiple testing burden: focusing on SNPs with additive associations and variants in imprinted regions. These analyses revealed a broad spectrum of POEs, uncovering previously uncharacterized mechanisms. In additively associated regions, we identified two novel POEs on telomere length near the genes *TERC* and *NAF1*, which are key regulators of telomere biology[26–28]. These associations align with previous work, suggesting that telomere regulation may be influenced by imprinting[29–31]. Their location outside of known imprinted regions suggests that genomic imprinting mechanisms may operate beyond established imprinted regions or may hint that additional regions may be subject to imprinting. Of note, the POEs near *TERC* exhibited a pattern distinct from the current classification of POEs, here termed 'asymmetric polar' parental effects: both parental alleles affect the trait in the same direction, but with significantly different magnitudes of effect (Supplementary Fig. 16). This novel POE pattern may be due to incomplete imprinting, in which both parental alleles are expressed but at different levels.

Our scan of imprinted regions uncovered many bipolar POEs, for which maternal and paternal alleles influence traits in opposite directions, including SHBG, triglyceride, HDL-C and glucose levels, standing height, cystatin C and creatinine levels, basal metabolic rate, and various fat-free and fat mass measures. Such effects have only been scarcely reported[1,7], mainly in regions subject to methylation[32,33].

Our findings reveal that such effects are, in fact, relatively common at imprinted loci. Although their underlying mechanisms remain unclear, our follow-up analysis at the 7q32.2 imprinted region — exhibiting bipolar POEs on triglyceride, HDL-C and SHBG levels — revealed a possible explanation for this bipolar phenomenon. It appears that the lead POE variant is an (additive) eQTL for the maternally expressed *KLF14* and *CPA4* and also an eQTL for the paternally expressed *MEST*. If both of those genes impact the associated complex traits, such gene expression-mediated effects may explain certain bipolar patterns for complex traits, suggesting a rather indirect antagonism. These findings underscore the complexity of POEs and the need for functional studies to elucidate imprinting mechanisms in metabolic traits.

Phenotype-rich biobanks allow in-depth POE pheWAS follow-up, which can pinpoint pleiotropic mechanisms underlying multitrait associations and opens new avenues to follow-up, such as colocalization and Mendelian randomization. For instance, at 11p15.5, a POE associated with height was also linked to basal metabolic rate, fat-free mass and whole-body water mass, suggesting that parental effects influence energy allocation beyond regulating stature. Similarly, POEs at 7q32.2 affected triglyceride, HDL-C and SHBG levels, which are established indicators of metabolic health and are closely tied to insulin resistance and metabolic syndrome (Supplementary Note 17).

We leveraged the wide range of available phenotypes to examine contrasting parental effects across traits and loci. The parental conflict hypothesis predicts that paternally inherited alleles promote growth, whereas maternally inherited alleles conserve resources. Therefore, we expect to see more bipolar effects for growth-related and metabolism-related traits. Consistent with this, all 19 identified bipolar effects (7 independent SNP–trait pairs) were restricted to growth and metabolic traits.

In addition to locus-specific examples consistent with the conflict hypothesis (for example, at the 11p15.5 locus for height, fat-free mass, kidney function biomarkers and metabolic traits), we also observed genome-wide trends through parent-specific heritability estimates ($h^2_{POE}$; for example, for IGF1 level, birth weight and triglyceride level) reflecting differential selection pressures on maternal and paternal alleles (Supplementary Note 18).

Moving beyond complex traits, we identified four significant POE pQTLs, including two in known imprinted genes (*DLK1* and *CPA4*). Of note, POE pQTL analysis of a triglyceride-associated variant revealed new candidate genes at this locus, such as *KLF14* and *MEST* (Supplementary Note 19). Two additional paternal POE pQTLs were found for *ADAM23* and *PER3*, outside known imprinted regions but exhibiting father-specific expression regulation[34,35], supporting potential incomplete or context-dependent imprinting. In addition, our POE pQTL for *PER3* is further supported by previous evidence of a POE methylation QTL at the same gene[33] (Supplementary Note 19). Several known imprinted genes lacked significant POE pQTLs, possibly due to post-transcriptional regulation that masks direct PofO protein effects; thus, integrating POE eQTLs in such analyses will be an important future direction.

Finally, our validation efforts in both the Estonian Biobank and the MoBa cohort demonstrate that whereas some POEs are robustly replicated (up to 87% of testable findings; Supplementary Table 13), such as our novel bipolar POE on triglyceride, HDL-C and creatinine levels (Supplementary Note 20), others require larger sample sizes to achieve significance, such as the sex-specific POE on glucose.

In summary, our study introduces a scalable approach to PofO inference at the biobank scale, which enabled us to create one of the largest POE-informed cohort thus far. By uncovering over 30 robust POEs, many of which exhibit bipolar effects on growth and metabolic traits, we have provided compelling evidence that POEs contribute meaningfully to complex trait genetic architecture. These effects, often undetectable in standard additive models, offer new insights into the potential evolutionary origin of imprinting, supporting the parental conflict hypothesis. Our findings highlight the need for a broad integration of PofO inference into genetic studies to lay the foundation for future efforts to map their molecular mechanisms and clinical relevance. Moving forwards, expanding sample sizes through meta-analyses of multiple familial biobanks, such as Finngen[36] or HUNT[37], will be critical. Such efforts will be pivotal in uncovering the full genetic architecture of POEs and understanding their implications for complex traits.

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

**Estonian Biobank research team**

**Andres Metspalu[4], Lili Milani[4,9], Tõnu Esko[4], Reedik Mägi[4], Mari Nelis[4] & Georgi Hudjashov[4]**

## Methods

### UK Biobank genotype processing

We used the UK Biobank Axiom Array data provided in PLINK format[39] and converted it to variant call format (VCF) using PLINK (v1.90b5)[40]. We then used the UK Biobank SNPs quality control file (UK Biobank resource 1955) to filter the data using BCFtools (v1.8) to keep only variants used for the official phasing of the original UK Biobank data release[39], resulting in 670,741 variant sites across the 22 autosomes and 16,601 variant sites on the X chromosome. We then used the SHAPEIT5 phase_common tool[11] with default parameters and no filter on allele frequency to perform an initial phasing of autosomes. For the X chromosome, we proceed as described in the official phasing report of the UK Biobank whole-genome sequencing (WGS) data, interim release of 200,031 samples[41]. In brief, we removed pseudoautosomal regions, identified male individuals as genetically determined, forced reference allele homozygosity at heterozygotes sites in male individuals, and finally we provided the list of male individuals as input in the –haploid option of SHAPEIT5. Female individuals were phased as autosomes.

### Close relative clustering

We use pairwise kinship estimates computed using the KING software (v2.2.4)[42] to identify related individuals up to the fourth degree in the UK Biobank cohort. We identified parent–offspring duos and trios as having kinship between 0.1767 and 0.3535, IBS0 (proportion of SNPs with zero identity by state) below 0.0012 and age difference greater than 15 years[4,39], resulting in 1,071 trios and 4,136 duos (Supplementary Fig. 1a,b). We used Mendel error rate to ensure the accuracy of the identified parent–offspring trios (Supplementary Fig. 1c).

We next identified sibling pairs as those with kinship between 0.1767 and 0.3535, IBS0 above 0.0012 and age difference smaller than 15 years[4], resulting in 22,751 pairs for 41,661 unique individuals (Supplementary Fig. 1a,b).

For individuals with more distant relatives (up to the fourth degree), we utilized a clustering approach to segregate relatives by parental sides using the igraph package in R[43], as previously done[4]. Through this method, we identified 274,525 UK Biobank participants whose relatives could be clustered into parental groups on the basis of their relatedness (Supplementary Fig. 1d). In our analysis, we refer to those close relatives as surrogate parents.

We finally identified individuals who have both available parental genomes and inferred surrogate parents, and thus can be used for validating our method. This allowed us to perform the entire analysis using surrogate parents and excluding parental genomes, effectively simulating the remaining individuals without available parental genomes, and later reintroduced the parental genomes solely for validation and accuracy estimation. In subsequent sections, we refer to those as the validation cohort. A total of 2,141 individuals met these criteria, resulting in 3,160 target–relative pairs (Supplementary Fig. 2a). For each target–relative pair, we used KING pairwise kinship estimates between the relative of the target and the parents of the target to assign the relative to a parental side (Supplementary Fig. 2b). This validation cohort was subsequently used to assess the accuracy of our inferences and to derive probabilities for PofO assignments.

### Interchromosomal phasing from available parental genomes

For individuals with available parental genomes (1,071 trios and 4,136 duos), we performed interchromosomal phasing using the –pedigree option of SHAPEIT5 (ref. 11). This method requires a three-column pedigree file listing offspring, fathers and mothers. In cases in which one parent was unavailable, the missing parent was indicated by 'NA'. This procedure resulted in phased genotype data (that is, haplotypes). The order of the haplotypes corresponded to the order of parents in the pedigree file (here, first haplotypes were paternally inherited, and second haplotypes were maternally inherited). Consequently, this method allowed for direct inference of the PofO of haplotypes from statistical phasing when parental genomes were available. The key advantage of using pedigree-based statistical phasing over traditional Mendelian logic is its ability to infer the PofO of alleles even when both parents are heterozygous. This is accomplished by statistical phasing, leveraging haplotype information from the broader population, leading to more accurate results. In this study, the PofO inferred from parental genomes serves as the ground truth. This ground truth is critical for validating the accuracy of PofO inference derived from surrogate parents in the validation cohort (see subsequent sections).

### Interchromosomal phasing from surrogate parents

IBD mapping is a powerful approach for identifying haplotype segments co-inherited from a common ancestor among pairs of relatives. By analysing haplotype segments' shared IBD with the same set of surrogate parents across the 22 autosomes for a given target individual, we could determine which haplotype segments were inherited from the same parent. This enabled the construction of partial parental haplotype sets, which could then be used to perform interchromosomal phasing that segregates haplotypes inherited from each parent across the genome. This approach goes beyond traditional phasing methods, which are limited to resolving haplotypes within individual chromosomes (intrachromosomal phasing). For 274,525 individuals with identified surrogate parents, we used the THORIN tool[4] to map haplotype segments' shared IBD between the target individuals and their surrogate parents. These IBD-shared segments served as the foundation for both intrachromosomal and interchromosomal phasing, building on previous methodologies[4]. The key principle here is that haplotypes' shared IBD with the same surrogate parents originate from the same parent. Consequently, these haplotypes should consistently appear on the same parental haplotype across autosomes. In practice, we filtered for shared haplotype segments longer than 3 centimorgans (cM), which we incorporated into a scaffold file. This scaffold was then used as input for the SHAPEIT5 phase_common tool. This step allowed us to refine and re-estimate haplotypes from genotype data, while simultaneously correcting intrachromosomal phasing switch errors and performing interchromosomal phasing by assigning all haplotypes shared with the same surrogate parents to the same parental haplotype (for example, first or second) across all 22 autosomes.

### Parental side determination of surrogate parents

**X chromosome.** IBD mapping on the X chromosome has been shown as an accurate approach to identify maternal relatives for male target individuals[4]. Here we built on this previous approach to probabilistically infer the parental side of close relatives. To do so, we utilized the THORIN tool to map IBD segments between a target individual and its surrogate parents. When a target individual has several surrogate parents from the same family side, we combined them into a single surrogate parent set, which allowed the THORIN tool to merge overlapping IBD segments from different relatives within the same set. We then retained only the largest IBD segment (size computed in Morgans) between the target and each surrogate parent set. Unlike the previous approach that only determine maternal relationships using a strict threshold in the X chromosome IBD[4], we probabilistically determined the parental side of surrogate parents. To derive probabilities of a surrogate parents set being on the paternal or maternal side, we used the validation cohort and computed the length of the X chromosome IBD haplotype segment shared with the target individual. This method effectively maximizes the number of surrogate parents with determined parental status.

Let $l(i, j)$ mark the length in cM of the longest haplotype segment shared on the X chromosome between individuals $i$ and $j$. Let $I_{pat}(i)$ refer to the set of all $N_{pat}(i)$ paternal relatives of individual $i$ and similarly for $I_{mat}(i)$ and its cardinality $N_{mat}(i)$. In addition, we define

$l_{\text{pat}}(i) := \frac{1}{N_{\text{pat}}(i)} \cdot \sum_{j \in I_{\text{pat}}(i)} l(i,j)$ as the (average) length of X chromosome haplotype sharing of an individual $i$ with its surrogate fathers, in the training set. We define $l_{\text{mat}}(i)$ analogously for surrogate mothers. As we know that surrogate fathers should not share an X chromosome segment and surrogate mothers are much more likely to share longer segments, we expect $l_{\text{mat}}(i)$ to be much larger than $l_{\text{pat}}(i)$.

Finally, let $I_{\text{pat}}$ refer to the set of $N_{\text{pat}}$ target individuals that have paternal relatives in the training set. In the same way, $I_{\text{mat}}$ refer to the set of $N_{\text{mat}}$ target individuals that have paternal relatives in the training set.

For any fixed length $l$, let us define the proportion of an event of two individuals sharing X chromosome haplotypes of length shorter than $l$ when one individual is a paternal relative of the other one:

$$\widehat{Pr}(l(i,j) < l | j \in I_{\text{pat}}(i)) = \frac{1}{N_{\text{pat}}} \sum_{i \in I_{\text{pat}}} I(l_{\text{pat}}(i) < l)$$

where $I(A)$ takes the value of 1 if statement $A$ is true and zero otherwise. When a shared segment length is very large, the proportion is expected to be very small, that is, it is unlikely to share large segments with surrogate fathers. Analogously, we can define

$$\widehat{Pr}(l(i,j) < l | j \in I_{\text{mat}}(i)) = \frac{1}{N_{\text{mat}}} \sum_{i \in I_{\text{mat}}} I(l_{\text{mat}}(i) < l)$$

As,

$$Pr(j \in I_{\text{pat}}(i) | l(i,j) < l) = Pr(l(i,j) < l | j \in I_{\text{pat}}(i)) \cdot \frac{Pr(j \in I_{\text{pat}}(i))}{Pr(l(i,j) < l)}$$

and

$$Pr(j \in I_{\text{mat}}(i) | l(i,j) < l) = Pr(l(i,j) < l | j \in I_{\text{mat}}(i)) \cdot \frac{Pr(j \in I_{\text{mat}}(i))}{Pr(l(i,j) < l)}$$

and they sum up to 1, that is,

$$Pr(j \in I_{\text{pat}}(i) | l(i,j) < l) = Pr(l(i,j) < l | j \in I_{\text{pat}}(i)) \cdot \frac{Pr(j \in I_{\text{pat}}(i))}{Pr(l(i,j) < l)}$$

Assuming a uniform prior, that is, $Pr(j \in I_{\text{pat}}(i)) = Pr(j \in I_{\text{mat}}(i))$, we have

$Pr(j \in I_{\text{pat}}(i) | l(i,j) < l)$

$$= \frac{Pr(j \in I_{\text{pat}}(i) | l(i,j) < l)}{Pr(j \in I_{\text{pat}}(i) | l(i,j) < l) + Pr(j \in I_{\text{mat}}(i) | l(i,j) < l)}$$

$$= \frac{Pr(l(i,j) < l | j \in I_{\text{pat}}(i)) \cdot \frac{Pr(j \in I_{\text{pat}}(i))}{Pr(l(i,j) < l)}}{Pr(l(i,j) < l | j \in I_{\text{pat}}(i)) \cdot \frac{Pr(j \in I_{\text{pat}}(i))}{Pr(l(i,j) < l)} + Pr(l(i,j) < l | j \in I_{\text{mat}}(i)) \cdot \frac{Pr(j \in I_{\text{mat}}(i))}{Pr(l(i,j) < l)}}$$

$$= \frac{Pr(l(i,j) < l | j \in I_{\text{pat}}(i))}{Pr(l(i,j) < l | j \in I_{\text{pat}}(i)) + Pr(l(i,j) < l | j \in I_{\text{mat}}(i))}$$

Thus,

$\widehat{Pr}(j \in I_{\text{pat}}(i) | l(i,j) < l)$

$$= \frac{\frac{1}{N_{\text{pat}}} \sum_{i \in I_{\text{pat}}} I(l_{\text{pat}}(i) < l)}{\frac{1}{N_{\text{pat}}} \sum_{i \in I_{\text{pat}}} I(l_{\text{pat}}(i) < l) + \frac{1}{N_{\text{mat}}} \sum_{i \in I_{\text{mat}}} I(l_{\text{mat}}(i) < l)}$$

We can similarly derive that

$\widehat{Pr}(j \in I_{\text{mat}}(i) | l(i,j) > l)$

$$= \frac{\frac{1}{N_{\text{mat}}} \sum_{i \in I_{\text{mat}}} I(l_{\text{mat}}(i) > l)}{\frac{1}{N_{\text{mat}}} \sum_{i \in I_{\text{mat}}} I(l_{\text{mat}}(i) > l) + \frac{1}{N_{\text{pat}}} \sum_{i \in I_{\text{pat}}} I(l_{\text{pat}}(i) > l)}$$

Note that when $l$ is large $\frac{1}{N_{\text{mat}}} \sum_{i \in I_{\text{mat}}} I(l_{\text{mat}}(i) > l)$ will dominate and hence the ratio will approach 1. Hence, when $l$ is large, it is a faithful measure of the probability that if we observed $l$ long X chromosome sharing between two individuals ($i$ and $j$), how likely is it that $j$ is on the maternal side of $i$. However, as $l$ decreases, it does not inform us about the chances of $j$ being on the maternal side of $i$.

In theory, we could have used $\widehat{Pr}(j \in I_{\text{pat}}(i) | l(i,j) = l)$ and $\widehat{Pr}(j \in I_{\text{mat}}(i) | l(i,j) = l)$, but in practice, these measures turned out to be too noisy when not enough $(i,j)$ training data pairs were available with $l(i,j)$ being close enough to $l$.

We initially applied this approach to the male individuals of the validation cohort ($n = 857$). We found that haplotype segments' shared IBD with surrogate fathers are all shorter than 11.3 cM, and 99% of them are shorter than 3 cM (Supplementary Fig. 6a). We then evaluated the accuracy of our probabilistic assignment. To do so, we proceeded with a 'leave one sample out' approach: we removed a given individual from the training set and predicted its paternal and maternal sides by deriving probabilities using the $N - 1$ individuals of the training set (Supplementary Fig. 6b). We repeated this for the $N$ individuals of the training set. To compute accuracy at a given maternal probability $P$, we defined true positives as the number of surrogate mothers with probability $>P$, and false positives as the number of surrogate fathers with probability $>P$. We then computed accuracy as TP/(TP + FP) (Supplementary Fig. 6c).

Applying this method to the UK Biobank male individuals with available surrogate parents and unknown parental assignment, we were able to probabilistically assign parental sides to surrogate parents for a total of 115,027 male individuals. Of note, 34% of these individuals ($n = 39,100$) achieved a parental assignment probability of 1, indicating a robust and reliable classification (Supplementary Fig. 6d).

**mtDNA.** mtDNA, which is inherited exclusively from the mother, serves as a valuable tool for identifying maternal relatives. However, the UK Biobank Axiom array data proved inadequate for this purpose due to the limited number of genotyped variants ($n = 265$). To overcome this limitation, we used the WGS GraphTyper cram file available on the UK Biobank Research and Analysis Platform for 500,000 individuals. We called variants from the mtDNA WGS cram files for 274,525 UK Biobank individuals (those with interchromosomal phasing) and their surrogate parents using the MitoHPC software[44] with default parameters. We then aimed to determine the parental side of each surrogate parent on the basis of genetic similarities of their mtDNA. However, most IBD mapping software relies on the Li and Stephens hidden Markov model, which models the human recombinant genome. Owing to the lack of recombination of mtDNA and its higher mutation rate than autosomes, these kinds of software proved unsuitable. As a result, we adopted an alternative approach to evaluate non-recombinant DNA sharing between pairs of individuals. This approach, inspired by the Jaccard index, introduces the term minor variant sharing (MVS), which captures the proportion of shared minor alleles between relative pairs. The approach overcomes the limitations of traditional IBD mapping for mtDNA by replacing IBD with identity by state (IBS).

Let $V_t$ denote the vector of genotypes for the $M$ mtDNA variants in the target individual, and $V_r$ the corresponding vector for the relative of the target. In both vectors, genotypes are coded as 0 for the major allele and 1 for the minor allele. Thus, we computed mtDNA MVS as:

$$\mathrm{MVS}(t, r) = \frac{V_t^\top \times V_r}{M}$$

Using the validation cohort, we found that maternal relatives exhibit a higher average MVS than paternal relatives (Supplementary Figs. 7–9a). The effectiveness of this metric depends on the degree of relatedness and is particularly precise for second-degree relatives. For these relatives, MVS values strongly correlate with the probability of the surrogate parent being on the maternal side. However, for more distant relatives (due to the possible presence of an intermediary male relative in the genetic lineage disrupting mtDNA inheritance), low MVS values do not translate to paternal relationship.

To predict the parental side of relatives from the mtDNA MVS, we adopted a perfectly analogous procedure to the one outlined for chrX sharing whereby we replaced chrX IBD segment length with the mtDNA MVS value. To evaluate the accuracy of this approach, we stratified target samples by their degree of relatedness to their closest relatives in the cohort (Supplementary Figs. 7–9C).

This methodology was applied to predict the relative parental side for 19,022 second-degree, 114,965 third-degree and 312,447 fourth-degree target–relative pairs (Supplementary Figs. 7d–9d). For targets with multiple relatives of the same degree, we retained only the relative with the highest predicted accuracy, resulting in 17,625 second-degree, 95,151 third-degree and 204,924 fourth-degree target–relative pairs. Of note, 69,580 of these pairs were assigned to a parental side with a probability greater than 0.99 (Supplementary Fig. 10a). In addition, parental side assignments were supported by multiple relatives for 9,948 individuals (Supplementary Fig. 10b).

## PofO determination

**From close relatives.** We integrated interchromosomal phasing with inferred parental side information from relatives to determine the PofO for each parental haplotype set. Specifically, a haplotype set was classified as maternally inherited if its haplotypes shared IBD segments with maternal relatives, and as paternally inherited if they shared IBD segments with paternal relatives. When the assignment was possible for only one of the two haplotypes in a target individual, we assumed the absence of uniparental disomy and inferred the parental origin of the second haplotype by exclusion.

**From crossovers to PofO inference in siblings.** By analysing IBD segments shared by siblings, we can infer crossover locations in parental haplotypes. Together with sex-specific genetic recombination maps[9], Qiao et al.[10] have demonstrated that we can estimate the likelihood of a set of crossovers originating from either the mother or the father, allowing us to determine the PofO of the haplotype carrying this set of crossovers. This method is reliable exclusively for sibling pairs, as they guarantee that the detected crossover events occurred in the parents. We identified IBD-shared haplotypes between sibling pairs using the THORIN tool and kept only segments larger than 3 cM. We used IBD segment breakends as crossover positions. However, we were very unlikely to capture the exact crossover positions using this approach. First, because we were restricted to genotyped markers. Second, owing to the nature of the Li and Stephens hidden Markov model used to map IBD segments in the THORIN tool, which does not transition at homozygous sites, we identified crossovers at heterozygotes only. Therefore, we used a 1,000-bp window around each crossover position to increase chances of including the true position. We then used recombination maps to extrapolate the probability of a crossover occurring within this window on the basis of the genetic distance in Morgans between the start and end positions of the window. We repeated this process for both the male-specific and the female-specific recombination maps for all the crossover events identified on the same haplotype.

Let $D_f(p)$ be the distance in Morgan of the 1,000-bp window around the crossover inferred at position $p$, originating from the female-specific recombination map, and $D_m(p)$ the distance originating from the male-specific recombination map position. To determine the most likely parent giving rise to an observed crossover at position $p$, we can compute the difference on log(Morgan) scale between male and female recombination probability:

$$\Delta(p) = \log_{10}(D_f(p)) - \log_{10}(D_m(p))$$

Assuming perfect intrachromosomal phasing, we can deduce that all crossovers identified on a given haplotype are inherited from the same parent. We can therefore aggregate Morgan differences for the $n_0$ and $n_1$ crossover positions of haplotype 0 and haplotype 1 on chromosome $c$, respectively. Let us define a score ($S$) for chromosome $c$ using both haplotypes as:

$$S_c = \sum_{i=1}^{n_0} \Delta_0(p(i)) - \sum_{i=1}^{n_1} \Delta_1(p(i))$$

This approach can be sensitive to misidentified crossovers, which may arise from various sources, such as inaccuracies in IBD segment boundaries due to errors in the IBD mapper, genotyping errors or consanguinity. These errors can inflate or deflate the global score. To mitigate this and remove confounders, we compared the observed number of crossover $N_{\mathrm{obs}}(\mathrm{CO})$ events to the expected number $N_{\mathrm{exp}}(\mathrm{CO})$. $N_{\mathrm{exp}}(\mathrm{CO})$ depends on the chromosome length in Morgan $l_c$. As we aggregated crossovers across both siblings first and second haplotypes, the expected number of crossovers $N_{\mathrm{exp}}(\mathrm{CO})$ on these four haplotypes follows a Poisson distribution with $\lambda = 4l_c$. However, given that the observed number of crossovers $N_{\mathrm{obs}}(\mathrm{CO})$ inferred using our approach depends on IBD sharing patterns, $N_{\mathrm{obs}}(\mathrm{CO})$ can range from zero when siblings are in IBD2 or IBD0 for the entire chromosome, to $4l_c$. However, artefacts from the IBD mapper can inflate this number. Therefore, we evaluated the distribution of observed crossover for each chromosome, and individuals strongly deviating from this distribution (more than 10 times the interquartile range) were removed.

The accuracy of determining the PofO of haplotypes depends on the number of crossovers analysed, which varies with chromosome length (in Morgans; Supplementary Fig. 11a). Using intrachromosomal phasing, we could only determine the PofO for crossovers within individual chromosomes by computing $S_c$, requiring separate calculations for each chromosome. By contrast, interchromosomal phasing enabled us to group crossovers occurring across the entire genome on the same parental haplotype set, allowing us to infer the PofO for the entire set at once, where the final score ($S$) was simply adding up the scores across chromosomes (Supplementary Fig. 11b):

$$S = \sum_{c=1}^{22} S_c$$

We initially assigned the PofO of haplotype using this score as follows: a negative score means that the first haplotype of the individual is paternally inherited, and a positive score means that it is maternally inherited. We evaluated the efficiency of our approach using a subset of the validation cohort that included 88 individuals with at least one sibling and available interchromosomal phasing from close relatives. As interchromosomal phasing might not be available for all chromosomes, we simulated varying conditions by altering the number of chromosomes analysed per individual, resulting in different numbers of inferred crossovers depending on the genomic length analysed (Supplementary Fig. 11b). We found that interchromosomal phasing substantially increases accuracy compared with single chromosome analysis (Supplementary Fig. 11c–e). Moreover, the PofO of crossovers could be assigned for 96.4% of individuals using interchromosomal phasing versus only 51.3% with intrachromosomal phasing (Supplementary Fig. 11f).

To avoid relying on arbitrary thresholds, we utilized the validation cohort to derive probabilistic determinations of the parental origin of haplotypes using inferred crossovers from interchromosomally phased data. Let us consider a score $S_t$ for the target individual $t$ that has been computed using a specific number of interchromosomally phased chromosomes. Together, these chromosomes have a genomic length of $l_t$. As the number of observed crossovers $N_{obs}(CO)$ depends on $l_t$ and that $S_t$ depends on $N_{obs}(CO)$, we define $I_{pat}(t)$ as the set of individuals of the validation cohort for which $H_0$ is paternally inherited and for which $l$ is within the window $(l_t - 3, l_t + 3)$, and $I_{mat}(t)$ as the set of individuals of the validation cohort for which $H_0$ is maternally inherited and for which $l$ is within the window $(l_t - 3, l_t + 3)$. The cardinality of these sets is denoted by $N_{pat}(t)$ and $N_{mat}(t)$, respectively. This allowed us to compare our obtained total score to the score obtained for a subset of the validation cohort that had a similar available genomic length as $l_t$.

As negative scores are compatible with paternal assignment and positive scores with maternal assignment, we could then define the probability of paternal origin of a genome-wide haplotype set of target sample $t$ as follows

$$Pr_{pat}(t) = \frac{\frac{1}{N_{pat}(t)} \sum_{i \in I_{pat}(t)} I(S_i < S_t)}{\frac{1}{N_{pat}(t)} \sum_{i \in I_{pat}(t)} I(S_i < S_t) + \frac{1}{N_{mat}(t)} \sum_{i \in I_{mat}(t)} I(S_i < S_t)}$$

and $Pr_{mat}(t)$ is defined as $1 - Pr_{pat}(t)$.

We evaluated the accuracy of this approach using the validation cohort. To explore all dependencies, we simulated varying conditions by altering the number of chromosomes analysed per individual, ranging from only one chromosome used to the full set of interchromosomally phased chromosomes for each individual. As expected, we observed better accuracy for scores strongly deviating from zero (Supplementary Fig. 12a). In addition, we observed that larger available genomic lengths (that is, more chromosomes) resulted in less errors, and that no errors were detected when using more than 10 Morgan (Supplementary Fig. 12b). Of note, when using the full set of available interchromosomally phased chromosomes per individual in the validation cohort (that is, the most realistic scenario), we achieved 100% accuracy (Supplementary Fig. 12a,b).

We applied this approach on 26,635 individuals with available interchromosomal phasing and at least one sibling. The specific configurations of our sample set (genomic length and number of crossovers) matched the highest accuracy (Supplementary Fig. 12c,d).

As interchromosomal phasing is not available for all siblings, we additionally use 14,695 individuals without interchromosomal phasing available and at least one sibling. We applied the single-chromosome approach using strict threshold at scores of −2 and 2 to determine the PofO of each target haplotypes separately (see Supplementary Fig. 11 for accuracy), and assessed the different characteristics of this inference (Supplementary Fig. 13).

**Accuracy of combined PofO predictors.** We used our PofO predictors to attempt to assign the PofO for 272,384 individuals with interchromosomally phased data and unavailable parental genomes. Of these, 5,312 were filtered out due to sex chromosome aneuploidy, excess of X chromosome heterozygosity for male individuals, lack of mtDNA WGS data, or withdrawal of individual or relative in-between the beginning of the analysis and the use of the WGS data. For the 267,072 remaining individuals, we attempted to assign the PofO using our predictors. A given individual can have several predictors of its haplotype PofO: X chromosome IBD, mtDNA MVS from different relatives and sib-score. For each individual, we kept the predictor with the highest estimated accuracy to determine parental side. As a result, we could unambiguously determine the parental side of relatives for 99.1% of the target individuals ($n = 264,597$). For ambiguous cases ($n = 2,475$), we then

prioritized predictors in the order indicated by overall accuracy of the predictor: X chromosome, sib-score and MVS for second-degree relatives. We obtained 209 individuals with parental sides undetermined because of conflicting mtDNA-based PofO predictions from different (third and fourth degree) relatives in the same surrogate parent cluster. In total, we successfully assigned the PofO for 266,863 individuals, and the PofO remained unassigned for 5,521 individuals that had available interchromosomal phasing (5,312 filtered out in the quality control and 209 with conflicting PofO predictors). In addition to assigning the PofO for individuals with available interchromosomally phased data, we also used single-chromosome sibling score for 14,596 individuals with at least one sibling and unavailable interchromosomally phased data, bringing our total number of individuals with PofO assigned from predictors to 281,459. Finally, we also added PofO prediction from available parental genomes for 1,071 trios and 4,136 duos, for a total of 286,666 individuals.

We compared our PofO determination to the one obtained from parental genomes at each heterozygous site of the validation cohort, revealing an accuracy of 97.94% (Supplementary Fig. 14).

## Parental haplotypes imputation and encoding

To infer the PofO for untyped alleles, we used haploid imputation as previously done[4] to separately impute each parental haplotype using the –out-ap-field option of IMPUTE5 (v1.2.1)[45] and the Haplotype Reference Consortium as a reference panel. This provides an AP field (alternative allele probabilities per haplotype), indicating imputed paternal and maternal allele dosages. We then kept only variants with an INFO score greater than 0.8 and minor allele frequency greater than 1%.

We encoded parental haplotype in separate files. For each target and at each variant, we weighted the parental allele imputed dosage with the parental assignment probability. Let us consider an imputed variant $v$ for a target individual $t$ with haploid imputed dosages $AP_{mat}(t, v)$ and $AP_{pat}(t, v)$ for maternal and paternal haplotype, respectively. Let us also consider the parental assignment probability $P_t$ for the target $t$ given by combined predictors. We re-encoded our data as:

$$DS_{mat}(t, v) = AP_{mat}(t, v) \cdot p_t + AP_{pat}(t, v) \cdot (1 - p_t)$$

$$DS_{pat}(t, v) = AP_{pat}(t, v) \cdot p_t + AP_{mat}(t, v) \cdot (1 - p_t)$$

From this, we adapted a haploid genotype posterior probability field $GP$ (probability triplets for the three possible genotypes) as:

$$GP_{mat}(t, v) = (1 - DS_{mat}(t, v), DS_{mat}(t, v), 0)$$

$$GP_{pat}(t, v) = (1 - DS_{pat}(t, v), DS_{pat}(t, v), 0)$$

Finally, we also combined maternal and paternal alleles at heterozygous sites only to directly test for differential effect (all homozygous sites are encoded as missing):

$$GP_{diff}(t, v) = (DS_{mat}(t, v)/DS_{dip}(t, v), DS_{pat}(t, v)/DS_{dip}(t, v), 0)$$

where

$$DS_{dip}(t, v) = DS_{mat}(t, v) + DS_{pat}(t, v)$$

The data were re-encoded as VCF files, which were then converted into PGEN files using PLINK (v2.00a4.3).

## Identification of POEs

POEs is a generic term encompassing many types of POE. In this article, we claimed POE if the maternal and paternal effect estimates of an allele were significantly different. To detect these, we assessed the phenotypic difference between those heterozygous individuals who carry the

effect allele paternally versus maternally. Once such a difference was detected, we further classified POEs on the basis of the size and direction of the paternal versus maternal effects (Supplementary Fig. 16).

**Phenotype processing.** We processed the 59 selected traits (Supplementary Table 1) as follows. For quantitative traits, we averaged out phenotype values across all available time points to obtain one value per individual. We then inverse-normal quantile transformed each trait using the rntransform function from the GENABEL R package[46]. For T2D, we defined cases on the basis of the ICD-10 (International Classification of Diseases, Tenth Revision) code 'E11' (non-insulin-dependent diabetes mellitus). To enhance specificity, we excluded from the case and control groups any individuals who also had ICD-10 code 'E10' (insulin-dependent diabetes mellitus). Furthermore, individuals diagnosed with other forms of diabetes (for example, gestational diabetes) were removed from both cases and controls.

**Association tests.** To perform GWAS analysis, we used REGENIE (v3.2.9)[38]. Only genotyped variants were used for model fitting, as recommended (that is, REGENIE step 1). We then tested paternal, maternal and differential haplotypes (that is, $GP_{pat}$, $GP_{mat}$ and $GP_{diff}$) separately (that is, REGENIE step 2). We denote the $P$ values resulting from these GWAS $P_P$, $P_M$ and $P_D$ for paternal, maternal and differential GWASs, respectively. We used the first 20 principal components, age and sex as covariates. In addition, we restricted our association tests to individuals who self-identified as 'white British' and have a similar genetic ancestry determined by principal component analysis (UK Biobank field 22006), and we restricted the differential scan to variants with imputed parental dosages greater than 0.99, as previously done[4].

Given the complexity and diverse nature of POEs, we used a multi-purpose approach to identify novel putative associations. Besides the traditional genome-wide scan, we focused on known imprinted regions, that is, within a 500-kb window around known imprinted genes[7], which are more likely to exhibit POEs. Second, we expanded our analysis to perform a GWAS scan focusing on regions exhibiting additive association with the examined trait. For these analyses, we selected a total of 59 phenotypes, including anthropometric traits, growth-related measures, cardiometabolic traits and blood biomarkers. This comprehensive approach allowed us to maximize the potential for discovering novel POE signals across a broad spectrum of traits. To account for the number of traits tested, we computed phenotypic correlation between each pair of traits. We then used the function simpleM from the R package hscovar to compute the effective number of traits tested, which resulted in $N_{eff}$(traits) = 48. All associations were pruned using genetic distance smaller than 500 kb and linkage disequilibrium < 0.01.

To identify POE genome wide, we used a stringent POE significance threshold (that is, differential $P$ value $P_D$) of $5 \times 10^{-8}/48 = 1 \times 10^{-9}$ to correct for the number of independent tests performed across our 59 selected traits.

To identify POE within additive regions, we first selected significant additive associations ($P < 5 \times 10^{-8}/48 = 1.04 \times 10^{-9}$), which we then tested for POEs using the differential GWAS. We corrected our POE significance threshold for the number of identified additive association such that the differential $P$ value $P_D$ threshold $= 0.05/N_a$, where $N_a$ is the number of independent additive associations.

To avoid overly conservative type I error control, we adjusted the POE significance threshold by accounting for the linkage disequilibrium structure within imprinted regions. After calculating the number of effective independent tests, we applied a significance threshold of $P_D < 0.05/16,574 = 3.01 \times 10^{-6}$, where 16,574 is the number of independent tests performed within imprinted regions. POEs identified at this threshold are referred to as suggestively significant POEs. In addition, we verified whether the identified associations would pass the more stringent threshold of $6.27 \times 10^{-8}$, which corresponds to $3.01 \times 10^{-6}/48$,

where 48 is the effective number of traits tested. POEs identified at this threshold are referred to as significant POEs.

**Classification of POEs.** To classify POEs, we categorized associations on the basis of the relative magnitudes and directions of the maternal ($Z_{Mat}$) and paternal ($Z_{Pat}$) $Z$-scores. The classification criteria are as follows:
- Bipolar effects: the maternal and paternal effects are of opposite signs (sign($Z_{Mat}$) ≠ sign($Z_{Pat}$)) and have similar magnitudes ($|Z_{Mat}| > 0.5 \times |Z_{Pat}|$ and $|Z_{Pat}| > 0.5 \times |Z_{Mat}|$).
- Maternal effect: the maternal $Z$-score is at least twice as large as the paternal $Z$-score in absolute value ($|Z_{Pat}| < 0.5 \times |Z_{Mat}|$).
- Maternal asymmetric: the maternal and paternal effects have the same sign, the maternal effect is dominant ($|Z_{Mat}| > |Z_{Pat}|$) but the paternal effect is still substantial ($|Z_{Pat}| > 0.5 \times |Z_{Mat}|$).
- Paternal effect: the paternal $Z$-score is at least twice as large as the maternal $Z$-score in absolute value ($|Z_{Mat}| < 0.5 \times |Z_{Pat}|$).
- Paternal asymmetric: the paternal and maternal effects have the same sign, the paternal effect is dominant ($|Z_{Pat}| > |Z_{Mat}|$), but the maternal effect remains notable ($|Z_{Mat}| > 0.5 \times |Z_{Pat}|$).

**Conditional POE analyses.** To explore the presence of secondary, independent POEs near primary loci, we performed conditional analyses using REGENIE[38]. This involved including the lead linkage disequilibrium-pruned variant as a covariate in the model to account for its effect, to obtain a conditional differential $P$ value $P_{D_c}$ for the remaining variants. By conditioning on the lead variant, we aimed to identify secondary POEs that might otherwise remain undetected due to linkage disequilibrium or proximity filtering. This approach ensures that any secondary signals are independent of the primary association.

**Replication of known PofO associations**
We used publicly available summary data for eight studies reporting POEs or separate paternal and maternal GWAS estimates[4,5,7,12-16]. When differential GWAS coefficients were not directly available, we computed a differential scan similar to the one performed in our study:

$$Z_D = \frac{\widehat{\beta}_P^2 - \widehat{\beta}_M^2}{\sqrt{s.e._P^2 + s.e._M^2}}$$

$$P_D = 2 \times (1 - \Phi(|Z_D|))$$

where $\Phi$ is the cumulative distribution function of the standard normal distribution and $|Z_D|$ is the absolute value of the $Z$-score.

**Sex-specific differences in POEs**
We performed a differential GWAS scan separately for male and female individuals to obtain sex-specific coefficients. We then assessed significant differences of effects using a $Z$-score approach:

$$Z_D = \frac{\widehat{\beta}_{D_{males}} - \widehat{\beta}_{D_{females}}}{\sqrt{s.e._{D_{males}}^2 + s.e._{D_{females}}^2}}$$

$$P_{Z_D} = 2 \times (1 - \Phi(|Z_{compl}|))$$

**SNP heritability**
We performed linkage disequilibrium score regression[47] to compute SNP heritability ($h^2$) using our GWAS association summary statistics and publicly available linkage disequilibrium scores from the 1000 Genome Project. We showed that the linkage disequilibrium based on the parental dosage difference equals the linkage disequilibrium of additive genotype dosage (see Supplementary Note 21). We munged GWAS summary data and estimated heritability with default parameters.

When $h^2$ was computed for subsets of the genome (that is, within and outside imprinted regions), we weighted the estimates by the proportion of variants included in the subset. To test the difference between heritability estimates, we combined estimates into a $Z$-score:

$$Z_{h^2} = \frac{h_a^2 - h_b^2}{\sqrt{s.e._a^2 + s.e._b^2}}$$

$$P_{h^2} = 2 \times (1 - \Phi(|Z_{h^2}|))$$

where $a$ and $b$ represent (1) differential estimates within imprinted regions and outside imprinted regions (Extended Data Fig. 7a), or (2) paternal and maternal estimates (Extended Data Fig. 7c,d).

## Replication analyses with childhood and infant traits

**Comparative traits at age 10 in the UK Biobank.** In the UK Biobank, we tested comparative body height at age 10, comparative body size at age 10 and birth weight. For comparative height size at age 10 (phenotype code 1697), we encoded the data as shorter = 0, about average = 1, and taller = 2. For comparative body size at age 10 (phenotype code 1687), we encoded the data as thinner = 0, about average = 1, and plumper = 2. We treated these three traits as quantitative traits and inverse-normal quantile transformed each trait using the rntransform function from the GENABEL R package[46].

**Infant height and BMI in MoBa.** MoBa[23,48] is a prospective cohort study that recruited pregnant women in Norway between 1999 and 2008. The study enrolled approximately 114,500 children, 95,200 mothers and 75,000 fathers from 50 hospitals across Norway. Anthropometric measurements of children were collected at hospitals at birth and during routine visits at 6 weeks, 3 months, 6 months, 8 months, 1 year, 1.5 years, 2 years, 3 years, 5 years, 7 years and 8 years of age. Parents provided these measurements via questionnaires[23]. Parental origin of alleles was inferred using parental genomes. Parent–offspring trios were excluded if they met any of the following criteria: stillborn, deceased, twins, missing data in the Norwegian Medical Birth Registry, missing anthropometric measurements at birth, pregnancies for which the mother did not respond to the first questionnaire, or missing parental DNA samples. Genotyping was performed on multiple sets of trios randomly selected from the study biobank, for a total of 45,402 offsprings with both parents genotyped and at least one BMI or height measurement available. We estimated POEs on infancy and childhood BMI and height at 11 time points (at 6 weeks, 3, 6 and 8 months, and 1, 1.5, 2, 3, 5, 7 and 8 years of age; Supplementary Table 8) using REGENIE and including genotyping batch effects for the child, mother and father, the first 10 principal components, and the sex of the child as covariates. Similarly to the UK Biobank and Estonian Biobank cohort, we tested parental alleles in the second step. We then estimated the differential coefficients by comparing paternal versus maternal coefficients using a $Z$-score approach. Depending on the time point, up to 42,346 individuals were included in the association analyses.

**Multiple testing correction.** We restricted this analysis to seven loci found to be associated with obesity-related traits, and three height-associated loci in the UK Biobank cohort. In the UK Biobank, we tested the three height-associated loci for association with comparative height size and the seven obesity-related loci for association with both comparative body size and birth weight, for a total of $3 \times 1 + 7 \times 2 = 17$ tests.

In Moba, we tested the three height-associated loci for association with infant height and the seven obesity-related loci for association with infant BMI. As BMI and height measurements at different time points are correlated, we estimated the number of effective tests to apply appropriate multiple testing correction. This resulted in five

independent measurements for both BMI and height, for a total of $3 \times 5 + 7 \times 5 = 50$ independent tests.

In total, we set our significant threshold to $0.05/(17 + 50) = 7.4 \times 10^{-4}$ when evaluating our original POEs for association with childhood traits.

## Estonian Biobank genotype processing

We used the quality-controlled genotype data genome built GRCh38 in VCF provided by the Estonian Biobank team[25]. Similar to when processing the UK Biobank data, we used SHAPEIT5 (ref. 11) to phase autosomes and X chromosome, and KING[42] to estimate relatedness. To impute the Estonian Biobank genotype data, we used the Estonian Biobank WGS reference panel[49], including 2,695 individuals. We performed phasing of the reference panel using the two-step process of SHAPEIT5 (ref. 11). We used the resulting reference panel to impute the SNP array data using IMPUTE5 (ref. 45). For all the remaining analysis, we proceed as for the UK Biobank. To infer the PofO of haplotypes for individuals without available parental genomes, we used only IBD sharing on the X chromosome and sibling score, as mtDNA WGS were not available to compute MVS. For replication analysis, we focused on traits with at least 20,000 individuals in the Estonian Biobank. This resulted in ten traits: standing height, glucose levels, T2D, HbA1c, HDL-C and triglyceride levels, hip circumference, creatinine and cystatin C levels, and platelet count. We derived phenotypes from electronic health records, unless NMR metabolomics measurements were available[50], which were available in a larger sample size than blood biomarkers (for glucose, triglyceride, HDL-C and creatinine levels). Depending on the phenotype, up to 85,050 individuals were included in the association analyses.

## Ethics statement

The activities of the Estonian Biobank were regulated by the Human Genes Research Act, which was adopted in 2000 specifically for the operations of the Estonian Biobank. Individual-level data analysis in the Estonian Biobank was carried out under ethical approval 1.1-12/295 from the Estonian Committee on Bioethics and Human Research (Estonian Ministry of Social Affairs), using data according to release application nr T38 from the Estonian Biobank. For MoBa, informed consent was obtained from all study participants. The administrative board of MoBa led by the Norwegian Institute of Public Health approved the study protocol. The establishment of MoBa and initial data collection was based on a license from the Norwegian Data Protection Agency and approval from The Regional Committee for Medical Research Ethics. The MoBa cohort is currently regulated by the Norwegian Health Registry Act. The study was approved by The Regional Committee for Medical Research Ethics (no. 2012/67).

## Reporting summary

Further information on research design is available in the Nature Portfolio Reporting Summary linked to this article.

## Data availability

The summary data are publicly accessible for download from our repository[51]. The UK Biobank genetic data are available under restricted access. Access can be obtained by application via the UK Biobank Access Management System (https://www.ukbiobank.ac.uk/enable-your-research/apply-for-access). The Estonian Biobank data are also available under restricted access. Access to the Estonian Biobank data must be approved by the Scientific Advisory Committee of the Estonian Biobank and by the Estonian Committee on Bioethics and Human Research. More details are available at https://genomics.ut.ee/en/content/estonian-biobank#dataaccess. Data produced as part of this study (that is, interchromosomally phased data and PofO information) will be returned to their respective biobanks and access will be granted to approved researchers. The publicly available subset of the Haplotype Reference Consortium dataset is available from the

European Genome-Phenome Archive at the European Bioinformatics Institute, under accession EGAS00001001710. We used additional publicly available databases that have been consulted multiple time between September 2023 and December 2024: GeneImprint (http://www.geneimprint.com/) and the UK Biobank phenotype correlations (https://ukbb-rg.hail.is/).

## Code availability

The THORIN IBD mapping software (v1.2) is available under an MIT license (https://github.com/RJHFMSTR/THORIN)[52]. Detailed documentation, tutorials and pipeline to infer the PofO of alleles are available (https://rjhfmstr.github.io/THORIN/). All other custom codes used as part of this study are provided within the PofO inference pipeline (https://rjhfmstr.github.io/THORIN/docs/tutorials/pofo_pipeline.html).

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

**Acknowledgements** We thank the participants of all biobanks for sharing their data. The Estonian Biobank data analysis was carried out in part in the High-Performance Computing Center of University of Tartu, Estonia. The UK Biobank data analysis was carried out in part in the High-Performance Computing Center of University of Lausanne, Switzerland. Metabolomics data of the Estonian Biobank have been generated by as part of a collaboration between Nightingale Health Plc and the Institute of Genomics, University of Tartu. The development of the quality-controlled dataset was supported by the Estonian Research Council grant no. PRG1291. This research has been conducted using the UK Biobank application number 16389 and 66995, and funded by the Swiss National Science Foundation (SNSF) project grants SNSF 310030-189147 and 315230-219587. S.J. was supported by the Helse Vest's Open Research Grant (grant nos. 912250 and F-12144), the Novo Nordisk Foundation (grant NNF19OC0057445) and the Research Council of Norway (grant no. 315599). M.V. was supported by the Research Council of Norway (grant no. 301178). The Norwegian Institute of Public Health provided genomic data for the MoBa study in collaboration with the HARVEST collaboration supported by the Research Council of Norway (no. 229624), the NORMENT Centre (no. 223273) in collaboration with deCODE Genetics, the Center for Diabetes Research at the University of Bergen funded by the ERC AdG project SELECTionPREDISPOSED, and the MoBaPsychGen team supported by funding from the South-Eastern Norway Regional Health Authority (nos. 2021045, 2020022, 2022083 and 2018058).

**Author contributions** R.J.H. and Z.K. designed the study and wrote the manuscript. R.J.H. performed all experiments. R.J.H. and Z.K. designed all methods and statistical tests. R.J.H. and O.D. designed THORIN. O.D. contributed to the supervision of the statistical phasing and sex chromosome integration. T.C. contributed to the implementation of the sibling model. S.R. contributed to the imputation experiments. A.v.d.G. contributed to optimizing haplotype data encoding. L.M., R.M. and F.-D.P. contributed to the analyses in the Estonian Biobank cohort. A.M., L.M., T.E., R.M., M.N. and G.H. performed data collection, genotyping and quality control of the Estonian Biobank genetic data. J.K. and N.T. developed quality control criteria for the Estonian Biobank metabolite data. R.K., M.V. and S.J. contributed to the analyses in the MoBa cohort. All authors reviewed the final manuscript. The project was supervised by Z.K.

**Competing interests** O.D. is a current employee of Regeneron Genetics Center. The other authors declare no competing interests.

**Additional information**
**Correspondence and requests for materials** should be addressed to Robin J. Hofmeister or Zoltán Kutalik.

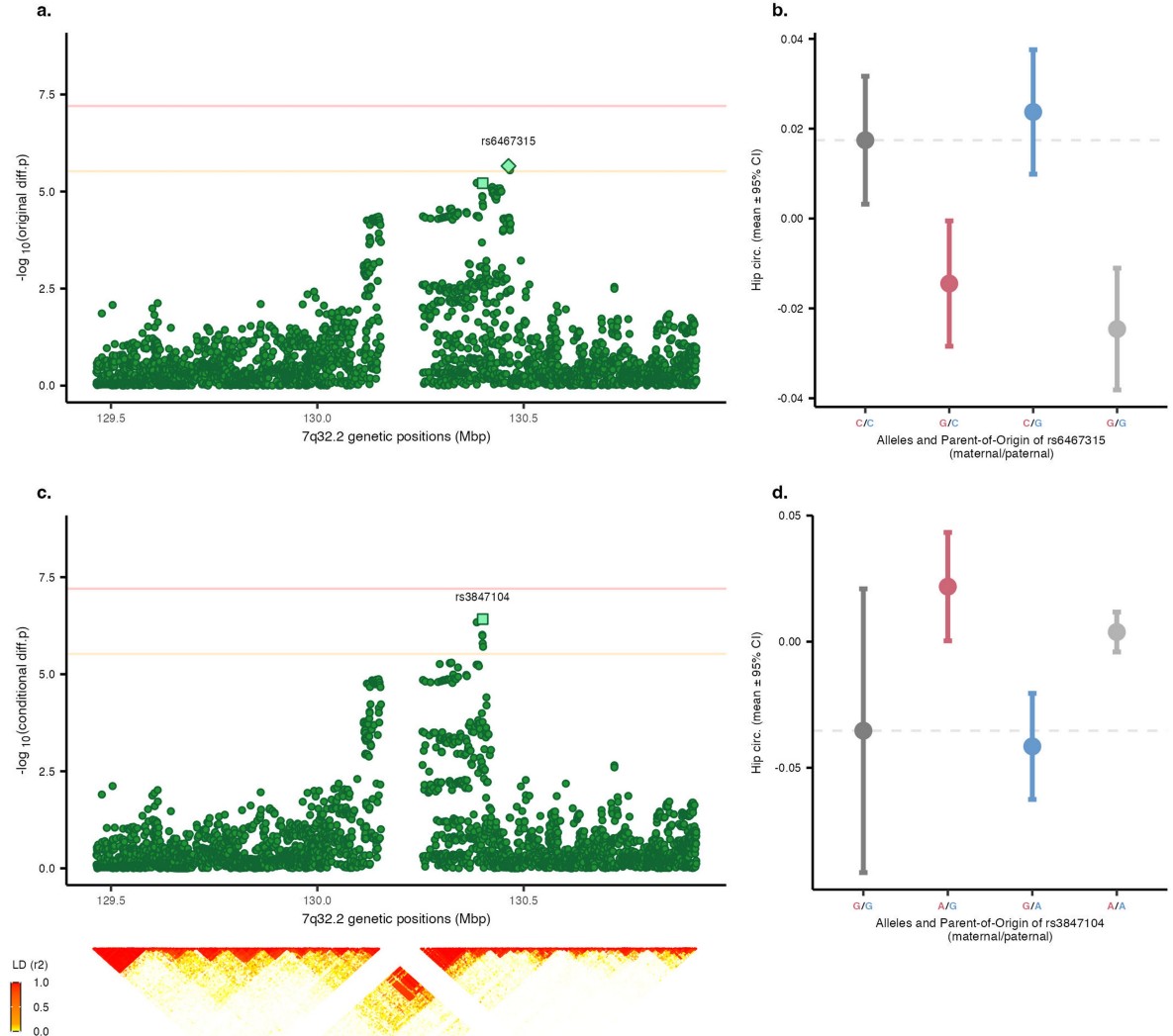

**Extended Data Fig. 1 | Conditional analysis on hip circumference at 7q32.2.**
**a)** Original differential GWAS on hip circumference. **b)** Genotypes and PofO of alleles (x-axis) effects of rs6467315 on hip circumference. **c)** Conditional differential GWAS on hip circumference. **d)** Genotypes and PofO of alleles (x-axis) effects of rs3847104 on hip circumference. **a)** and **c)** show the association strength ($-log_{10}(p-value)$, y-axis) against the genomic position (x-axis) at the 7q32.2 region. Each point represents a genetic variant. The diamond indicates the variant with significant POE on hip circumference found in this study in the primary GWAS scan. The square indicates the variant with significant and independent POE identified in the conditional analysis. Red and orange lines represent the significance threshold used to identify significant and suggestive POEs (see Methods). Bottom panel shows the Linkage Disequilibrium (LD) pattern, ranging from LD = 0 (white) to LD = 1 (red). **b)** and **d)** show the genotypes and PofO of alleles (x-axis) effects on normalized hip circumference (y-axis). Red and blue alleles indicate maternal and paternal alleles, respectively. Dots show mean values, error bars show 95% confidence intervals computed as $\beta \pm 1.96 \times SE$ and derived from n = 109,385 individuals. Grey dashed lines represent the mean value of individuals carrying no alternative allele (i.e., genotype 0). Dots and bars colors indicate maternal heterozygotes (red), paternal heterozygotes (blue), and homozygotes (dark and light grey).

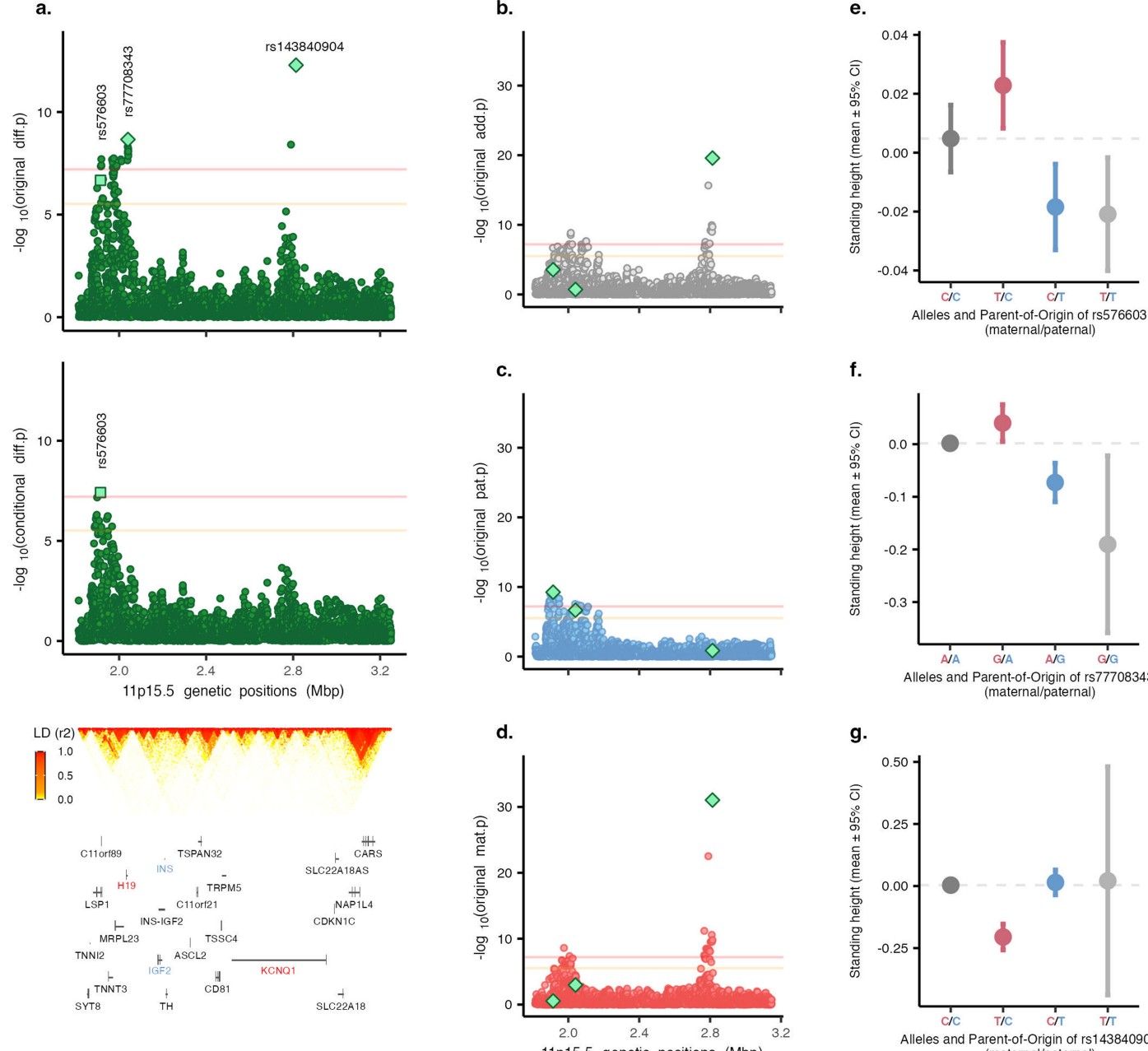

**Extended Data Fig. 2 | Parent-of-Origin associations on standing height.**
**a)** The two top panels represent the original and conditional differential GWAS (top) locuszoom plot and show the association strength ($-log_{10}(p-value)$, y-axis) against the genomic position (x-axis). Each point represents a genetic variant. Diamonds indicate the variants with independent, significant POE on standing height found in this study in the primary GWAS scan. The square represents the variant with significant and independent POE identified in the conditional analysis. Red and orange lines represent the significance threshold used to identify significant and suggestive POEs (see Methods). Linkage Disequilibrium (LD) pattern (middle) ranges from LD = 0 (white) to LD = 1 (red). Gene positions (bottom) along the 11p15.5 imprinted region (x-axis). Horizontal lines show the gene start and end positions. Vertical lines show the exons start and end positions. Gene names are shown below the corresponding gene coordinates. **b-d)** Additive (b, grey), Paternal (c, blue) and Maternal (d, red) associations on standing height shown as association strength ($-log_{10}(p-value)$, y-axis) against the genomic position (x-axis). Green diamonds indicate POEs selected in the differential scan. Orange lines represent the significance threshold used in this study when focusing on imprinted regions ($3.1 \times 10^{-06}$). **e-g)** Genotypes and PofO of alleles (x-axis) effects on standing height (y-axis). Red and blue alleles indicate maternal and paternal alleles, respectively. Dots show mean values, error bars show 95% confidence intervals computed as $\beta \pm 1.96 \times s.e.$ and derived from n = 109,385 individuals. Grey dashed lines represent the mean value of individuals carrying no alternative allele (i.e., genotype 0). Dots and bars colors indicate maternal heterozygotes (red), paternal heterozygotes (blue), and homozygotes (dark and light grey).

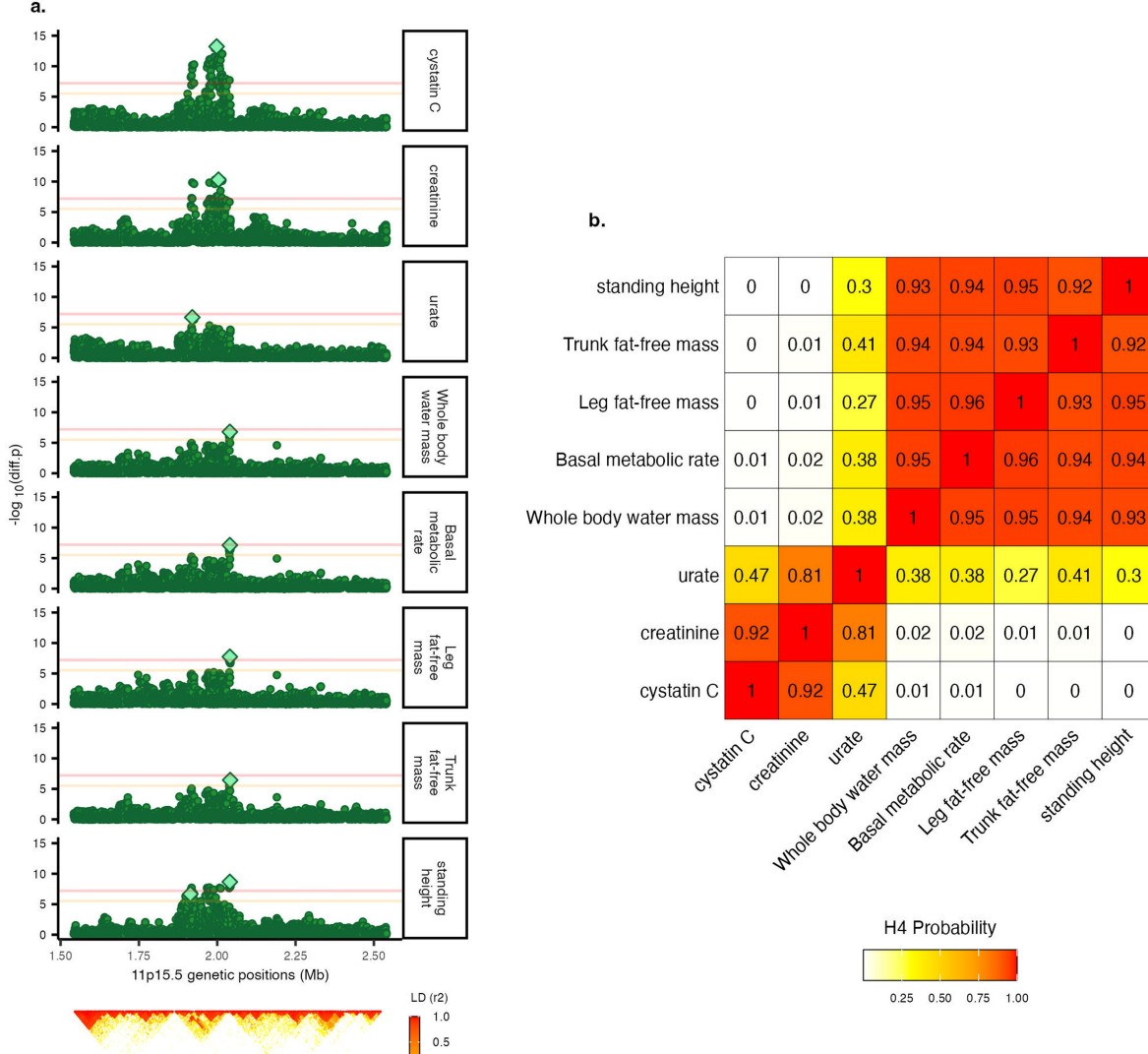

**Extended Data Fig. 3 | Co-localization analyses at 11p15.5. a)** Parent-of-origin associations across traits at 11p15.5 show the differential GWAS association strength ($-log_{10}(p-value)$, y-axis) against the genomic position (x-axis). Red and orange lines represent the significance threshold used to identify significant and suggestive POEs (see Methods). Each point represents a genetic variant. Light green diamonds shows lead POE reported in Table 1. **b)** Co-localization probability heatmap (i.e, shared causal variant probability $H_4$).

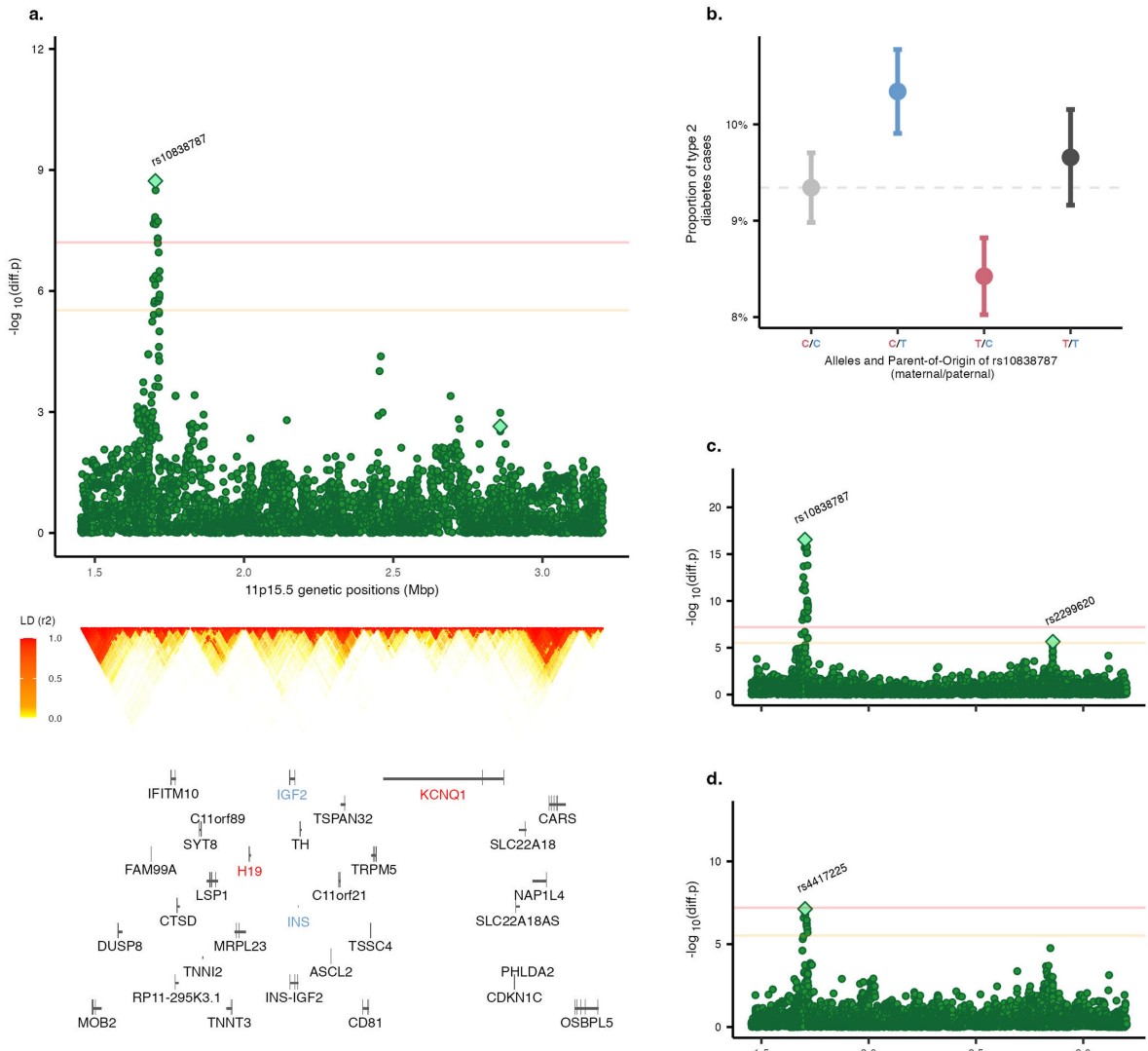

**Extended Data Fig. 4 | Parent-of-Origin associations with type 2 diabetes.**
Differential GWAS show the association strength ($-log_{10}(p-value)$, y-axis) against the genomic position (x-axis) on **a**) type 2 diabetes, **c**) HbA1c and **d**) glucose. Each point represents a genetic variant. Diamonds indicate variants with independent POE found in this study and listed in Table 1. Red and orange lines represent the significance threshold used to identify significant and suggestive POEs (see Methods). In **a**), linkage Disequilibrium (LD) pattern (middle) ranges from LD = 0 (white) to LD = 1 (red). Gene positions (bottom) are shown along the genomic positions (x-axis). Horizontal lines show the

gene start and end positions. Vertical lines show the exons start and end positions. Gene names are shown below the corresponding gene coordinates. **b**) Genotypes and PofO of alleles (x-axis) effects on type 2 diabetes incidence (y-axis). Red and blue alleles indicate maternal and paternal alleles, respectively. Dots show mean values, error bars show 95% confidence intervals computed as $\beta \pm 1.96 \times s.e.$ and derived from n = 108,196 individuals. Grey dashed lines represent the mean value of individuals carrying no alternative allele (i.e., genotype 0). Dots and bars colors indicate maternal heterozygotes (red), paternal heterozygotes (blue), and homozygotes (dark and light grey).

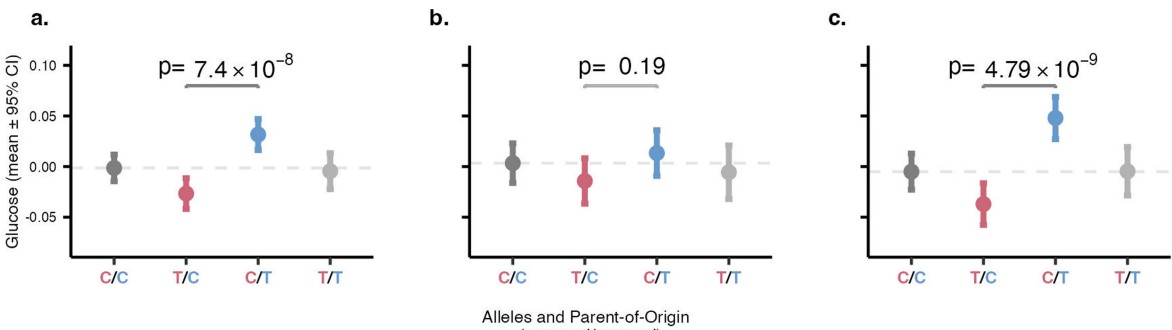

**Extended Data Fig. 5 | Sex-specific POE of rs4417225 on glucose levels.**
Effects of genotypes and PofO of alleles (x-axis) on glucose level (y-axis) for
(**a**) both sexes combined, (**b**) females only, and (**c**) males only. Red markers
represent maternal heterozygotes, blue markers represent paternal
heterozygotes, and dark and light grey represent homozygotes. The data
points show the mean glucose levels, with error bars indicating the 95%
confidence intervals. A grey dashed line represents the mean glucose level
for individuals with the reference genotype (C/C). Significance values from
the differential GWAS tests are annotated within each panel.

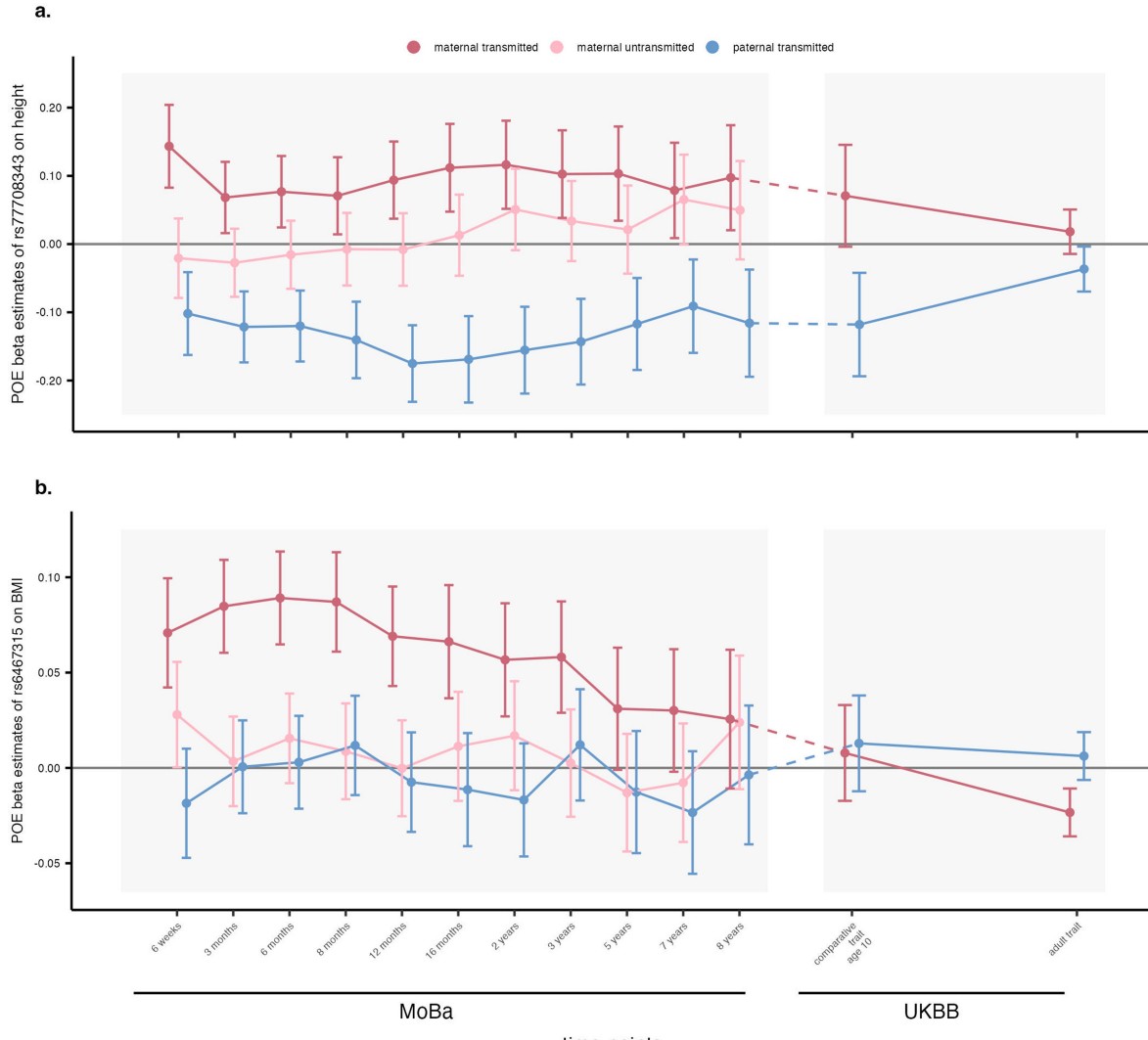

**Extended Data Fig. 6 | Parent-of-origin effects in early life.** Dots show beta estimates and error bars show 95% confidence intervals computed as $\beta \pm 1.96 \times$ s.e. (y-axis) of paternal transmitted (blue), maternal transmitted (red) and maternal untransmitted (pink) alleles effects on **a**) height and **b**) BMI time points (x-axis). Beta and 95% confidence intervals estimates were derived using REGENIE[38] and using a different number of individual at each time point, as indicated in Supplementary Table 8: n = 29,791; 41,865; 42,346; 36,790; 36,721; 27,595; 27,435; 27,985; 22,667; 23,280 and 17,888 for 6 weeks, 3 months, 6 months, 8 months, 1 year, 16 months, and 2 to 8 years, respectively.

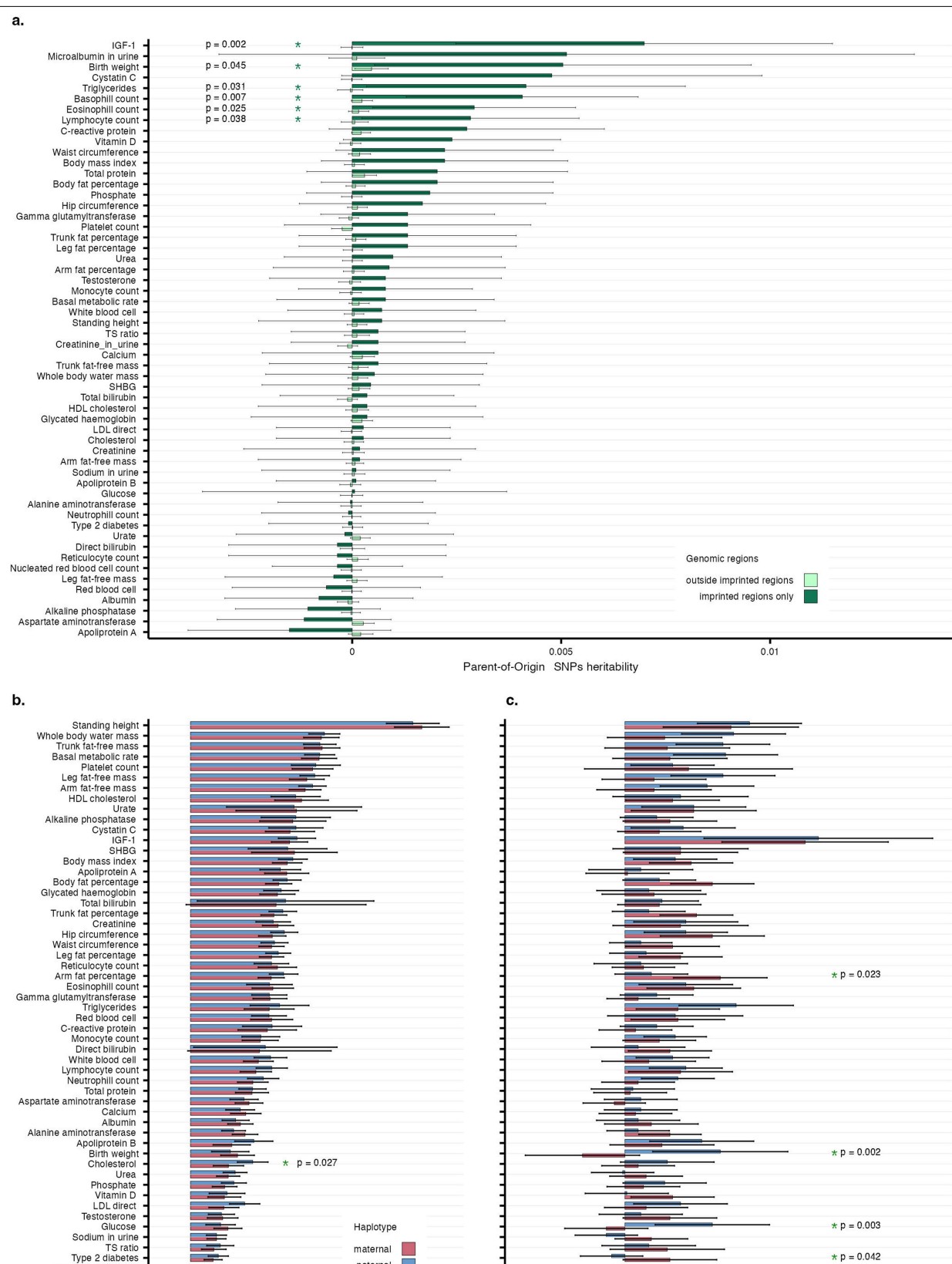

**Extended Data Fig. 7** | See next page for caption.

**Extended Data Fig. 7 | SNPs heritability. a** Parent-of-Origin SNPs heritability $h_{POE}^2$ was estimated within and outside imprinted regions. Stars and p-values indicate nominal significant differences between $h_{POE}^2$ within imprinted regions *vs* outside imprinted regions. Paternal (blue) and maternal (red) $h^2$ was estimated (**b**) genome-wide and (**c**) within imprinted regions. Stars and p-values indicate nominal significant differences between paternal and maternal $h^2$. Three out of the 59 selected traits were not represented here due to disproportionate standard errors: lipoprotein A, rheumatoid factor, and oestradiol.

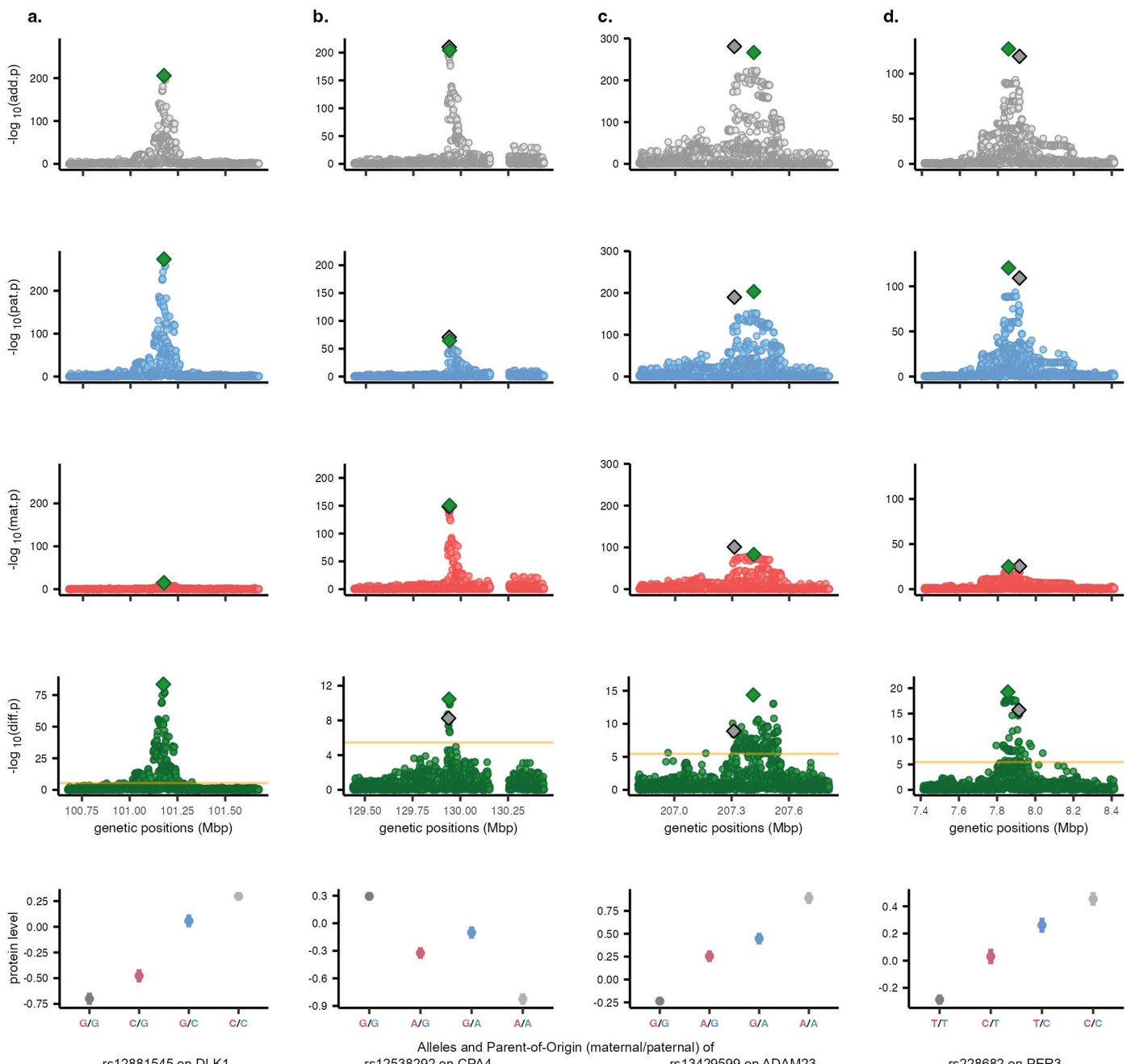

**Extended Data Fig. 8 | Significant Parent-of-Origin pQTLs.** Association strength ($-log_{10}(p-value)$, y-axis) against the genomic position (x-axis) for additive (grey), paternal (blue), maternal (red) and differential (green) GWAS on (**a**) DLK1, (**b**) CPA4, (**c**) ADAM23 and (**d**) PER3. Diamonds show the lead additive association previously reported[24] (grey) and the lead POE-pQTL detected in this study (green). Orange line indicate the significance threshold ($P_D < 0.05/14285 = 3.5 \times 10^{-6}$). **Bottom panel** show the effects of alleles and PofO on normalized protein levels. Dots show beta estimates and error bars show 95% confidence intervals estimates that were derived using REGENIE[38] and using n = 9,168; 7,488; 9,151 and 7,614 individuals for *DLK1*, *CPA1*, *ADAM23* and *PER3*, respectively.

## Extended Data Table 1 | Significant Parent-of-Origin pQTLs

| CHR | POS | SNP ID | A0 | A1 | A1FREQ | BETA PAT | SE PAT | P PAT | BETA MAT | SE MAT | P MAT | BETA DIFF | SE DIFF | P DIFF | BETA ADD | SE ADD | P ADD | TRAIT | STATUS |
|---|---|---|---|---|---|---|---|---|---|---|---|---|---|---|---|---|---|---|---|
| 14 | 101176212 | rs12881545 | G | C | 0.668 | 0.826 | 0.023 | 2.26E-274 | 0.187 | 0.023 | 1.24E-15 | 0.642 | 0.033 | 3.03E-84 | 0.505 | 0.016 | 2.34E-206 | DLK1 | primary pQTL and POE-pQTL |
| 7 | 129938598 | rs34587586 | G | T | 0.404 | -0.434 | 0.024 | 2.77E-71 | -0.634 | 0.024 | 4.44E-149 | 0.201 | 0.034 | 5.60E-09 | -0.532 | 0.017 | 1.53E-210 | CPA4 | primary pQTL |
| 7 | 129940561 | rs12538292 | G | A | 0.327 | -0.432 | 0.025 | 4.85E-65 | -0.667 | 0.025 | 2.32E-151 | 0.238 | 0.036 | 3.51E-11 | -0.548 | 0.018 | 7.76E-205 | CPA4 | primary POE-pQTL |
| 2 | 207311349 | rs72933203 | T | A | 0.104 | 1.017 | 0.035 | 1.45E-190 | 0.723 | 0.034 | 9.02E-102 | 0.293 | 0.048 | 1.29E-09 | 0.865 | 0.024 | 1.01E-281 | ADAM23 | primary pQTL |
| 2 | 207413625 | rs13429599 | G | A | 0.270 | 0.719 | 0.024 | 5.93E-204 | 0.457 | 0.024 | 9.30E-84 | 0.263 | 0.034 | 4.34E-15 | 0.576 | 0.017 | 4.18E-267 | ADAM23 | primary POE-pQTL |
| 1 | 7914177 | rs228647 | G | A | 0.570 | 0.506 | 0.023 | 6.87E-110 | 0.241 | 0.023 | 4.13E-26 | 0.264 | 0.032 | 1.95E-16 | 0.375 | 0.016 | 7.50E-120 | PER3 | primary pQTL |
| 1 | 7856346 | rs228682 | T | C | 0.401 | 0.540 | 0.023 | 4.34E-121 | 0.241 | 0.023 | 1.09E-25 | 0.297 | 0.032 | 5.41E-20 | 0.395 | 0.016 | 4.32E-128 | PER3 | primary POE-pQTL |

CHR: chromosome; POS: genetic position (hg19); SNP ID: variant rs id; A0: reference allele; A1: assessed allele; A1FREQ: A1 allele frequency. BETA, SE and P denote effect sizes, standard errors and P-values, computed using REGENIE[38] (two-sided test); PAT, MAT, DIFF and ADD denote paternal, maternal, differential and additive tests. TRAIT: phenotype name.

Zoltán Kutalik

# Reporting Summary

## Statistics

For all statistical analyses, confirm that the following items are present in the figure legend, table legend, main text, or Methods section.

| n/a | Confirmed | |
|---|---|---|
| ☐ | ☒ | The exact sample size (*n*) for each experimental group/condition, given as a discrete number and unit of measurement |
| ☒ | ☐ | A statement on whether measurements were taken from distinct samples or whether the same sample was measured repeatedly |
| ☐ | ☒ | The statistical test(s) used AND whether they are one- or two-sided<br>*Only common tests should be described solely by name; describe more complex techniques in the Methods section.* |
| ☐ | ☒ | A description of all covariates tested |
| ☐ | ☒ | A description of any assumptions or corrections, such as tests of normality and adjustment for multiple comparisons |
| ☐ | ☒ | A full description of the statistical parameters including central tendency (e.g. means) or other basic estimates (e.g. regression coefficient) AND variation (e.g. standard deviation) or associated estimates of uncertainty (e.g. confidence intervals) |
| ☐ | ☒ | For null hypothesis testing, the test statistic (e.g. *F*, *t*, *r*) with confidence intervals, effect sizes, degrees of freedom and *P* value noted<br>*Give P values as exact values whenever suitable.* |
| ☐ | ☒ | For Bayesian analysis, information on the choice of priors and Markov chain Monte Carlo settings |
| ☒ | ☐ | For hierarchical and complex designs, identification of the appropriate level for tests and full reporting of outcomes |
| ☒ | ☐ | Estimates of effect sizes (e.g. Cohen's *d*, Pearson's *r*), indicating how they were calculated |

*Our web collection on statistics for biologists contains articles on many of the points above.*

## Software and code

Policy information about availability of computer code

Data collection | No software was used for the data collection of this study.

Data analysis | Publicly available code used in this study :
THORIN v1.2 (https://github.com/RJHFMSTR/THORIN), for which we provide a full documentation and tutorials on a dedicated website (https://rjhfmstr.github.io/THORIN/).
BCFtools v1.8 (based on HTSlib v1.8)
SHAPEIT v5 (https://github.com/odelaneau/shapeit5)
REGENIE v3.2.9 (https://rgcgithub.github.io/regenie/)
PLINK v1.90b5
PLINK v2.00a4.3
R v4.3.1
R 'igraph' package v1.2.2
KING  v2.2.4

Non publicly available code used in this study:
IMPUTE5 v1.2.1

For manuscripts utilizing custom algorithms or software that are central to the research but not yet described in published literature, software must be made available to editors and reviewers. We strongly encourage code deposition in a community repository (e.g. GitHub). See the Nature Portfolio guidelines for submitting code & software for further information.

## Data

Policy information about availability of data

All manuscripts must include a data availability statement. This statement should provide the following information, where applicable:

- Accession codes, unique identifiers, or web links for publicly available datasets
- A description of any restrictions on data availability
- For clinical datasets or third party data, please ensure that the statement adheres to our policy

The summary data are publicly accessible for download on our webpage (http://poedb.dcsr.unil.ch/). The UK Biobank genetic data are available under restricted access. Access can be obtained by application via the UK Biobank Access Management System (https://www.ukbiobank.ac.uk/enable-your-research/apply-for-access). The Estonian Biobank data are also available under restricted access. The access to the Estonian Biobank data must be approved by the Scientific Advisory Committee of the Estonian Biobank and by the Estonian Committee on Bioethics and Human Research. More details are available at https://genomics.ut.ee/en/content/estonian-biobank#dataaccess. Data produced as part of this study (i.e inter-chromosomally phased data and PofO information) will be returned to their respective biobanks and access will be granted to approved researchers.
The publicly available subset of the Haplotype Reference Consortium (HRC) dataset is available from the European Genome-Phenome Archive at the European Bioinformatics Institute, accession EGAS00001001710. We used additional publicly available databases that have been consulted multiple time between September 2023 and December 2024: GeneImprint (http://www.geneimprint.com/) and the UK Biobank phenotype correlations (https://ukbb-rg.hail.is/).
- UK Biobank phenotype correlations (https://ukbb-rg.hail.is/)

## Research involving human participants, their data, or biological material

Policy information about studies with human participants or human data. See also policy information about sex, gender (identity/presentation), and sexual orientation and race, ethnicity and racism.

| | |
|---|---|
| Reporting on sex and gender | We use only genetically determined sex, that we refer to as "sex". |
| Reporting on race, ethnicity, or other socially relevant groupings | For the UK Biobank cohort, we used only individuals who self-identified as "White British" and have a similar genetic ancestry determined by principal components analysis (UK Biobank field 22006).<br>For the Estonian Biobank, we used only individuals determined at "Estonians" by principal components analysis.<br>For the MoBa cohort, outliers individuals, as defenied by principal components analysis, were removed. |
| Population characteristics | The UK Biobank population includes individuals between 37 and 73 years old at recruitment.<br>The Estonian biobank cohort includes individuals born between 1905 and 2005.<br>The MoBa cohort includes longitudinal measurements for children between 0 and 8 year of age. |
| Recruitment | Main analyses were performed in the UKBB, a volunteer-based cohort of 500,000 individuals from the general UK population for which participants signed a broad informed consent form.<br>MoBa is an open-ended cohort study that recruited pregnant women in Norway from 1999 to 2008. |
| Ethics oversight | The UK Biobank data was accessed under the project 66995 and 16389.<br>The Estonian Biobank data was accessed under the under ethical approval 1.1-12/295.<br>The use of the MoBa cohort was approved by The Regional Committee for Medical Research Ethics (#2012/67). |

Note that full information on the approval of the study protocol must also be provided in the manuscript.

## Field-specific reporting

Please select the one below that is the best fit for your research. If you are not sure, read the appropriate sections before making your selection.

☒ Life sciences          ☐ Behavioural & social sciences          ☐ Ecological, evolutionary & environmental sciences

For a reference copy of the document with all sections, see [nature.com/documents/nr-reporting-summary-flat.pdf](http://nature.com/documents/nr-reporting-summary-flat.pdf)

## Life sciences study design

All studies must disclose on these points even when the disclosure is negative.

| | |
|---|---|
| Sample size | The sample size of our study for the different cohorts and phenotypes are available in supplementary table 1 (for UK Biobank cohort), supplementary table 8 (MoBa cohort) and supplementary table 11 (Estonian Biobank cohort). Sample sizes for each cohort were determined as the number of individuals with both parent-of-origin information available and phenotype measurement available. |
| Data exclusions | As mentioned in the Methods section, we initially filtered out all variants that were not included for the phasing of the original UK Biobank release. These were determined from the SNPs QC file provided as part of the UK Biobank, ressource 1955. For the Estonian Biobank and Moba, we used pre-filtered data provided by the data management team of each respective cohort. |
| Replication | We attempted to replicate a total of 16 associations (53%) discovered in this study for which we had the corresponding genetic and phenotypic data available in an additional cohort (Estonian Biobank and Moba). For reproducibility, most of the softwares used for this study are publicly available and we provided details tutorial of our inference pipeline. |

| Randomization | Randomization was not used since there are no experimental groups. |
| Blinding | Blinding is not relevant to this study since no group allocation occurs. |

# Reporting for specific materials, systems and methods

We require information from authors about some types of materials, experimental systems and methods used in many studies. Here, indicate whether each material, system or method listed is relevant to your study. If you are not sure if a list item applies to your research, read the appropriate section before selecting a response.

## Materials & experimental systems

| n/a | Involved in the study |
|---|---|
| ☒ | Antibodies |
| ☒ | Eukaryotic cell lines |
| ☒ | Palaeontology and archaeology |
| ☒ | Animals and other organisms |
| ☒ | Clinical data |
| ☒ | Dual use research of concern |
| ☒ | Plants |

## Methods

| n/a | Involved in the study |
|---|---|
| ☒ | ChIP-seq |
| ☒ | Flow cytometry |
| ☒ | MRI-based neuroimaging |

## Plants

| Seed stocks | *Report on the source of all seed stocks or other plant material used. If applicable, state the seed stock centre and catalogue number. If plant specimens were collected from the field, describe the collection location, date and sampling procedures.* |
| Novel plant genotypes | *Describe the methods by which all novel plant genotypes were produced. This includes those generated by transgenic approaches, gene editing, chemical/radiation-based mutagenesis and hybridization. For transgenic lines, describe the transformation method, the number of independent lines analyzed and the generation upon which experiments were performed. For gene-edited lines, describe the editor used, the endogenous sequence targeted for editing, the targeting guide RNA sequence (if applicable) and how the editor was applied.* |
| Authentication | *Describe any authentication procedures for each seed stock used or novel genotype generated. Describe any experiments used to assess the effect of a mutation and, where applicable, how potential secondary effects (e.g. second site T-DNA insertions, mosiacism, off-target gene editing) were examined.* |

