## [Peer Review file · Nature]

Parent-of-origin effects on complex traits in up to 236,781 individuals

Corresponding Author: Dr Zoltán Kutalik

Version 1:

Reviewer comments:

Referee #1

(Remarks to the Author)

We start by noting that this report is prepared by two people.

The authors have extended and improved a method they developed for determining the parent-of-origin of alleles of genetic markers when a genealogy is unavailable, which is often the case, particularly with large populations. Applying their method to the UK Biobank and the Estonia Biobank, they separated out the paternally and maternally inherited haploid genomes for 123,716 and 97,346 individuals respectively. Together with 45,402 offspring from the Norwegian Mother, Father and Child Cohort Study, they were able to replicate a substantial number of previously reported genetic markers with parent-of-origin effects (POEs), and also revealed some possible false positives. Importantly, they have also discovered novel POEs, many exhibiting bipolar dominant behavior, i.e. both the paternally and maternally inherited allele have effects, but in opposite directions.

Overall, we are impressed with this work. Worth emphasizing is that PO information is not only useful for identifying POEs, but could play a key role in many other areas of genetic research. Moreover, the effectiveness and yield (percentage of samples for which PO information can be determined) of the method can improve as datasets become larger (Specific Comment 2).

In contrast to what the authors have achieved, we feel that the writing and presentation are somewhat uneven. One example is the paragraph that starts with (line 229) “We identified novel associations with type 2 diabetes (T2D), HbA1c, rs10838787 and rs4417225,” and ends with “Although these variants are novel, they are in high LD ($r^2 > 0.9$) with a variant previously associated with T2D in a parent-of-origin manner (ref 5). If the variants identified here are in such high LD with a previously reported variant, how is it a novel association? Calling the variants novel would also be misleading. Novelty aside, there is the unaddressed question of whether the two variants reported here could be a refinement of the original SNP reported, which we believe is rs2334499. Interestingly, rs2334499 falls within 350 Kb of a large cluster of imprinted genes, but was not considered to be within an imprinted region, at least in 2009. Maybe we are not being completely impartial, but this association is particular worth noting for several reasons. T2D has more impact on health than many of these other associations. For individuals with European descent, the variance of T2D risk explained by this SNP may only be second to the strongest variant TCF7L2. The opposite direction effects observed might be what motivated ref 14 to coin the term bipolar dominance. Maybe most important to the method described here, T2D is a late onset disease so many of the parent-offspring trios ascertained would not be useful. Indeed, we are not aware of a strong replication of this association since the original publication, and thus the strong results reported here are important even if not entirely novel.

As noted above, PO information is useful for many things other than POEs. As such, to maximize the impact of this manuscript, we hope that the authors would (a) do their best to make it easier for people to implement the method themselves and (b) to make the PO information on the UK Biobank samples available. Their tool THORIN is available on github with documentation. We understand that the authors are currently working on (a). That is, the authors are working on a tutorial to describe their entire parent-of-origin pipeline with the 1000 Genome Project as example data. We look forward to the completion of that tutorial and will likely try to use THORIN ourselves in the near future. With (b), one possibility is to

provide the information through the UKBB research analysis platform. However, that might not be easy or may take too much time. We do appreciate that it is very difficult to provide scripts that individual UKBB user can simply run and reproduce the PO information. But providing user friendly code of the method and any tips regarding the processing of the UKBB data on the research analysis platform would be very welcomed.

The manuscript is currently very long. Among other things, we believe the paper would read much better if the later part of the POEs results are mostly moved to the Supplement. These results are just not that interesting, we believe, for the general reader. Many of them are either borderline significant, insubstantial in effect and/or the trait is relatively unimportant. Unintentionally, they could actually dilute the impact of the genuinely impressive results and let the reader feel a little letdown when finishing. The Discussion section should probably also be shortened. By contrast, without necessarily increasing the length substantially, some additional details about the method, and the accuracy and yield of PO information, could be helpful.

Specific Comments

1. Lines 47 to 55. While the authors overcome the lack of genealogical information by clever use of the information from X chromosomes and mitochondrial DNA, it seems that the conceptual relationship with the method described by reference 5 should be made more explicit. After all, the concept of 'surrogate parenthood' in this context was, we believe, introduced in the long-range-phasing paper, the prequel to ref 5 which did not directly make use of the genealogy. A natural way to understand the authors' approach is to use the information from the X chromosomes and mitochondrial DNA to partially reconstruct the genealogy. The use of sex-specific recombination maps for this purpose with sibling data is however distinct, and something we are not aware of before.
2. The question of (percentage) yield for the UKBB samples. From the main text, if we understand correctly, 274,525 (line 89) have one or more surrogate parents identified, and at the end reliable PO information for close to the entire genome are determined for 123,716 individuals. But a higher number of 286,666 is mentioned on line 126. The question is what is the right denominator for percentage yield as the UKBB data set is clearly substantially larger than the numbers mentioned. Knowing the current UKBB percentage yield can allow us to project how it could improve as the number of UK genotyped individuals increases, due to having a larger number of surrogate parents per individual.
3. In the summary of the methods in the main text, the key steps start with inter-chromosomal phasing. The question is what about intra-chromosomal phasing which we assume is done using SHAPEIT5 (line 923). The legend of Fig. 1 mentions correcting for intra-chromosomal errors. It is of interest what is the intra-chromosomal phasing error rate to start with, and what complications that creates for the whole process.
4. Heritability. h^2_{POE} (line 504) should be explicitly defined. We presume it is the variance explained by POEs in addition to that of an additive effect model. If so, it looks rather modest (Figure 6c). It seems that should be the first point to make. If indeed the overall POE heritability is quite modest, the difference between paternal and maternal, and imprinted regions and not, are secondary issues. For the main text, it seems that a short paragraph would suffice.
5. The authors noted on page 6 that there is currently no consensus on how to identify POEs. Here, the authors have a great opportunity to set a rigorous standard. Achieving genome-wide significance for at least one of the association tests could be a useful criterion. The study presented some strong results, but including suggestive signals in the main text dilutes that message and having various significance thresholds makes the paper more difficult to read. In our opinion, the paper would benefit from moving the weaker/suggestive results to the supplement. Especially, the sub-chapter "Parent-of-origin specificity of additively associated regions". The rationale for using a more lenient significance threshold for variants in imprinted regions is more understandable. However, it is noted on page 40 in the paper, that the significance threshold for the variants in imprinted regions should be $6.27E-8$ when the number of phenotypes is taken into account. For results presented in the main text, this threshold could be more appropriate. Suggestive signals could be shown in the supplement.
6. Supplementary table 2 depicts replication results of previously published POEs. On closer examination, we believe that there are only 5 POE height signals in supplementary table 2, not 8. If we are not mistaken, the three variants that are shown under Zoledziewska et al. (2015) are all part of the same height signal which was replicated and refined to rs143840904 in 2016 (variant shown twice in the table). As a result, it would be more correct to say that the table is showing 19 previously reported POEs, of which 9 are replicated in the current study.

(Remarks on code availability)

We have not yet tried out the code, but plan to do so in the near future.

Referee #2

(Remarks to the Author)

We start by noting that this report is prepared by two people.

The authors have extended and improved a method they developed for determining the parent-of-origin of alleles of genetic markers when a genealogy is unavailable, which is often the case, particularly with large populations. Applying their method to the UK Biobank and the Estonia Biobank, they separated out the paternally and maternally inherited haploid genomes for

123,716 and 97,346 individuals respectively. Together with 45,402 offspring from the Norwegian Mother, Father and Child Cohort Study, they were able to replicate a substantial number of previously reported genetic markers with parent-of-origin effects (POEs), and also revealed some possible false positives. Importantly, they have also discovered novel POEs, many exhibiting bipolar dominant behavior, i.e. both the paternally and maternally inherited allele have effects, but in opposite directions.

Overall, we are impressed with this work. Worth emphasizing is that PO information is not only useful for identifying POEs, but could play a key role in many other areas of genetic research. Moreover, the effectiveness and yield (percentage of samples for which PO information can be determined) of the method can improve as datasets become larger (Specific Comment 2).

In contrast to what the authors have achieved, we feel that the writing and presentation are somewhat uneven. One example is the paragraph that starts with (line 229) "We identified novel associations with type 2 diabetes (T2D), HbA1c, rs10838787 and rs4417225", and ends with "Although these variants are novel, they are in high LD ($r^2 > 0.9$) with a variant previously associated with T2D in a parent-of-origin manner (ref 5). If the variants identified here are in such high LD with a previously reported variant, how is it a novel association? Calling the variants novel would also be misleading. Novelty aside, there is the unaddressed question of whether the two variants reported here could be a refinement of the original SNP reported, which we believe is rs2334499. Interestingly, rs2334499 falls within 350 Kb of a large cluster of imprinted genes, but was not considered to be within an imprinted region, at least in 2009. Maybe we are not being completely impartial, but this association is particular worth noting for several reasons. T2D has more impact on health than many of these other associations. For individuals with European descent, the variance of T2D risk explained by this SNP may only be second to the strongest variant TCF7L2. The opposite direction effects observed might be what motivated ref 14 to coin the term bipolar dominance. Maybe most important to the method described here, T2D is a late onset disease so many of the parent-offspring trios ascertained would not be useful. Indeed, we are not aware of a strong replication of this association since the original publication, and thus the strong results reported here are important even if not entirely novel.

As noted above, PO information is useful for many things other than POEs. As such, to maximize the impact of this manuscript, we hope that the authors would (a) do their best to make it easier for people to implement the method themselves and (b) to make the PO information on the UK Biobank samples available. Their tool THORIN is available on github with documentation. We understand that the authors are currently working on (a). That is, the authors are working on a tutorial to describe their entire parent-of-origin pipeline with the 1000 Genome Project as example data. We look forward to the completion of that tutorial and will likely try to use THORIN ourselves in the near future. With (b), one possibility is to provide the information through the UKBB research analysis platform. However, that might not be easy or may take too much time. We do appreciate that it is very difficult to provide scripts that individual UKBB user can simply run and reproduce the PO information. But providing user friendly code of the method and any tips regarding the processing of the UKBB data on the research analysis platform would be very welcomed.

The manuscript is currently very long. Among other things, we believe the paper would read much better if the later part of the POEs results are mostly moved to the Supplement. These results are just not that interesting, we believe, for the general reader. Many of them are either borderline significant, insubstantial in effect and/or the trait is relatively unimportant. Unintentionally, they could actually dilute the impact of the genuinely impressive results and let the reader feel a little letdown when finishing. The Discussion section should probably also be shortened. By contrast, without necessarily increasing the length substantially, some additional details about the method, and the accuracy and yield of PO information, could be helpful.

Specific Comments

1. Lines 47 to 55. While the authors overcome the lack of genealogical information by clever use of the information from X chromosomes and mitochondrial DNA, it seems that the conceptual relationship with the method described by reference 5 should be made more explicit. After all, the concept of 'surrogate parenthood' in this context was, we believe, introduced in the long-range-phasing paper, the prequel to ref 5 which did not directly make use of the genealogy. A natural way to understand the authors' approach is to use the information from the X chromosomes and mitochondrial DNA to partially reconstruct the genealogy. The use of sex-specific recombination maps for this purpose with sibling data is however distinct, and something we are not aware of before.
2. The question of (percentage) yield for the UKBB samples. From the main text, if we understand correctly, 274,525 (line 89) have one or more surrogate parents identified, and at the end reliable PO information for close to the entire genome are determined for 123,716 individuals. But a higher number of 286,666 is mentioned on line 126. The question is what is the right denominator for percentage yield as the UKBB data set is clearly substantially larger than the numbers mentioned. Knowing the current UKBB percentage yield can allow us to project how it could improve as the number of UK genotyped individuals increases, due to having a larger number of surrogate parents per individual.
3. In the summary of the methods in the main text, the key steps start with inter-chromosomal phasing. The question is what about intra-chromosomal phasing which we assume is done using SHAPEIT5 (line 923). The legend of Fig. 1 mentions correcting for intra-chromosomal errors. It is of interest what is the intra-chromosomal phasing error rate to start with, and what complications that creates for the whole process.
4. Heritability. h^2_{POE} (line 504) should be explicitly defined. We presume it is the variance explained by POEs in addition

to that of an additive effect model. If so, it looks rather modest (Figure 6c). It seems that should be the first point to make. If indeed the overall POE heritability is quite modest, the difference between paternal and maternal, and imprinted regions and not, are secondary issues. For the main text, it seems that a short paragraph would suffice.

5. The authors noted on page 6 that there is currently no consensus on how to identify POEs. Here, the authors have a great opportunity to set a rigorous standard. Achieving genome-wide significance for at least one of the association tests could be a useful criterion. The study presented some strong results, but including suggestive signals in the main text dilutes that message and having various significance thresholds makes the paper more difficult to read. In our opinion, the paper would benefit from moving the weaker/suggestive results to the supplement. Especially, the sub-chapter "Parent-of-origin specificity of additively associated regions". The rationale for using a more lenient significance threshold for variants in imprinted regions is more understandable. However, it is noted on page 40 in the paper, that the significance threshold for the variants in imprinted regions should be $6.27E-8$ when the number of phenotypes is taken into account. For results presented in the main text, this threshold could be more appropriate. Suggestive signals could be shown in the supplement.

6. Supplementary table 2 depicts replication results of previously published POEs. On closer examination, we believe that there are only 5 POE height signals in supplementary table 2, not 8. If we are not mistaken, the three variants that are shown under Zoledziewska et al. (2015) are all part of the same height signal which was replicated and refined to rs143840904 in 2016 (variant shown twice in the table). As a result, it would be more correct to say that the table is showing 19 previously reported POEs, of which 9 are replicated in the current study.

(Remarks on code availability)

Referee #3

(Remarks to the Author)

In this manuscript, Robin et al., proposed a method that combines inter-chromosomal phasing, mitochondrial and chromosome X data, and sibling-based crossover evidence to determine parent of origin inheritance of alleles in samples without parental genome information available. They applied the method to obtain parent-of-origin allele information in $\frac{1}{4}$ samples in UKBiobank, which allowed them to perform POE-specific association tests for 59 complex traits. They also replicated the application of the method and the identified associations in two external cohorts.

In my view, this POE-inference method is both innovative and holds promising application potential, given that the sample sizes of biobanks globally increase quickly but parental genome information is often incomplete or lacking. I also think the authors did a great job to show the practical usage of the method in identifying and replicating POE-specific associations.

In addition to these, I have some concerns on the rationale of applying several methods designed for additive genetic effects in a study of POE. Those include LDSC-based SNP heritability analyses and co-localization analyses, both heavily relying on the LD structure in the human genome. When these methods are applied to a POE study that essentially aims to differentiate genetic effects between maternal and paternal chromosomes, I think we need to consider the LD difference between parental chromosomes and the influence of such difference. It is well known that crossover (recombination) is one of the driving factors to shape the LD structure. And importantly, the crossover event can vary between sex (it is also based on this the authors proposed their crossover-inference strategy to assign parent-of-origin alleles using sibling pairs). Given the crossover positions that vary between genders, LD structure between parental chromosomes could vary too. These differences are averaged out when using unphased data in analyses for additive genetic effect, but should not be neglected in POE studies.

For example, in heritability analysis, the authors compared the heritability contributed from maternal and paternal haplotypes. Were the LD difference between parental chromosomes accounted in the calculation? How was the parent-of-origin SNP heritability calculated? In colocalization analysis, how was the maternally/paternally varied LD structure accounted?

My other comments are the following

1. The readership should be interested in seeing the estimated sample size required to reach reliable parent-of-origin inference using each of three methods under different data structure settings.

2. Line 153-155: "Therefore, we first pre-filtered claimed POEs by rigorously testing for a differential paternal vs maternal alleles effect and kept only published hits where the differential test P-value PD was lower than 10^{-3} , to ensure more robust evidence POE."

Is the PD provided in original publication, or recalculated by the authors using original samples from those studies, or calculated using samples from current study?

3. Line 1302: height studies.

Eight studies?

4. Line 158: "We successfully replicated 12 of them (54.5%) at $PD < 0.05/22$. Interestingly, of the 8 POEs reported for birth weight, none were replicated."

Any reason for that? Was the birth weight adjusted by gestational age?

5. Figure 2b: legend suggests “Unlabeled points correspond to bi-polar POEs” however in the figure itself “ZNF597, KFL14” were labeled but seem to have a bi-polar POE pattern?

6. Line 384-386: “the additively associated regions approach, which assesses the POE specificity of the lead additive variant but may not always pinpoint the true lead POE variant.”

Why the leading additive variant is different from the leading POE variant, if the additive model just dilute/average out the POE signal?

7. Line 707-709, “These associations are particularly notable in the context of bi-polar dominance, as the lead variants act as eQTLs for both maternally expressed genes (KLF14 and CPA4) and the paternally expressed MEST” and line 712-713, “This interplay between maternally and paternally expressed genes exemplifies the intricate regulatory mechanisms underlying bi-polar POEs”.

1) Were those eQTL analyses performed based on additive model?

2) Do these genes interact with one another, and do they exert activating or repressive effects on the phenotype? Observing significant associations between maternal and paternal alleles with different genes does not necessarily imply that they have opposite or distinct functional roles, and therefore does not adequately support the conflict theory of bipolar-type parent-of-origin effects.”

8. Lines 167-169: “traits related to growth and metabolism were particularly over-represented among pleiotropic loci, notably on chromosomes 7, 11, and 20 (Figure 2A), consistent with the expected roles of imprinted genes under the conflict theory.”

Conclusion of overrepresentation or pleiotropy should consider the fact that the number of phenotypes tested here are limited and those phenotypes are highly selective.

9. Lines 884-885: “our findings highlight the prevalence of bi-polar POEs, which we discovered to be far more common than previously recognized at imprinted loci”

Previous POE studies have reported a high proportion of bi-polar (complex imprinting) POE pattern
<https://www.nature.com/articles/s41467-019-09301-y> and <https://academic.oup.com/hmg/article/27/16/2927/5026426>

10. Line 554-555: “We found two POE associations where genes were located outside known imprinted regions: for ADAM23 and PER3 protein levels”.

POE on PER3 and POE-QTL SNP (both rs228647 and rs228682) regulation for PER3 have been previously reported in methylation study <https://www.nature.com/articles/s41467-019-09301-y>.

(Remarks on code availability)

Referee #4

(Remarks to the Author)

Hofmeister and colleagues propose a new method for determining parent of origin of alleles in large biobanks where parental genotypes are not available. By utilizing distant relatives (up to the 4th degree) in the UK Biobank and data from the X chromosome, mitochondrial DNA and sibling cross-overs, they were able to accurately infer the parent of origin of alleles. They subsequently performed association analyses with 29 phenotypic traits in the UK Biobank to identify parent of origin effects across the genome. Replication of these parent of origin effects were tested in the Estonian Biobank and the MoBa study. They also utilized the proteomics data in the UK Biobank to identify whether there were protein quantitative trait loci (pQTL) underlying their observed parent of origin effects on the phenotypic traits and to assess whether known pQTLs exhibited parent of origin effects.

This study will be of broad interest due to the novel method for inferring parent of origin of alleles and then the application to identify parent of origin effects on a wide range of traits (which is the largest study of its kind as far as I can tell). The method, which appears to be an extension of the authors previous work, was complex but made the most of the available data in the biobanks to infer parent of origin in as many participants as possible. It was nice to see how well the method performed across the three cohorts that differed substantially in terms of their family relatedness (from one or two distant relatives in the UK Biobank to larger families of closer relatives in the Estonian Biobank and predominantly parent/offspring trios in MoBa). However, the attempt at replicating the association results was a little disappointing as the majority of the phenotypes did not overlap across the cohorts. It was also somewhat surprising that only 30 parent of origin effects were identified across the 29 phenotypic traits, even with the large sample size available in the UK Biobank.

The manuscript was rather lengthy and could have been written more succinctly in places. A few more, relatively minor, aspects that require a little more clarity are mentioned below:

1. How does this method compare to the authors previous paper "Parent-of-Origin inference for biobanks"? How often are the PoE alleles the same between the two methods? Given there are several times the previous method is referenced, it would be good to have a formal comparison somewhere in the manuscript (supplementary material is fine).
2. There did not seem to be any information on the phenotypes that were analysed in any of the cohorts. What data cleaning (if any) was performed? Was it the baseline measure that was used in the UK Biobank or a combination of baseline and follow-up? How was type 2 diabetes defined?
3. In the results section, it is noted that six significant POEs from the additive analyses were identified (line 370) but Table 1 and Figure 2 only seem to show five (rs2237895 which is associated with HbA1c according to the results section is not presented in Table 1). Would the authors please clarify?
4. A flow diagram (in the supplementary material) would be useful to show how the samples were removed throughout the analysis pipeline – i.e. there are 274,525 UK Biobank individuals with up to 4th degree relatives, but only 109,385 analysed in most of the phenotype analyses; why were 165,140 individuals removed? Similarly for the replication cohorts.
5. In the discussion the authors speculate that some reasons for the PoEs not replicating in the Estonian Biobank could be due to statistical power, cohort characteristics, or environmental factors. It could also be false positives in UKB (perhaps related to the slightly lower imputation quality)?
6. How does the accuracy of PofO inference (i.e. errors in the PofO assignment) influence the downstream association analyses?
7. I'm struggling to understand how equations 1 and 2 were derived. As these equations are key to understanding the method, would the authors please provide an intuition behind the equations, or a little more of the derivation, to help the reader? Additionally, lines 1009-1015 the authors talk about test and training sets – can they please define what these are?
8. Why was only BMI across early life chosen as the phenotype for replication in MoBa and not height? Especially given none of the PoEs identified were associated with adult BMI but there were several associated with adult height.
9. The numbers are a little exaggerated in places. For example, the title claims that evidence of PoE is 'from ~265,000 individuals', whereas the actual numbers of individuals within each of the cohorts that have phenotype data are lower (~109K for UK Biobank, max N=85050 in Estonian Biobank and max N=42346 in MoBa). In the abstract 'more than 60 complex traits and over 2,400 protein levels' are investigated, but in reality, 59 traits and only previously associated proteins are tested. It would be worthwhile stating these more accurately.

(Remarks on code availability)

I have not run the code, but the author's website provides detailed instructions for setting up and running their software.

Version 2:

Reviewer comments:

Referee #2

(Remarks to the Author)

We note that this report is prepared by two people. Overall, we are satisfied with the revision of the manuscript and the responses of the authors to our comments.

(Remarks on code availability)

We successfully installed and compiled THORIN and were able to run the toy example without any issues. The installation instructions are clearly outlined and easy to follow.

Referee #3

(Remarks to the Author)

The authors have addressed my comments well. In some parts of the revised manuscript (mainly supplementary) there are mystery question marks, such as "Correlated these LD estimates at the 11p15.5 imprinted region (Supplementary Figure ?? A)."

(Remarks on code availability)

I didn't run the code. The authors have provided their tool in <https://rjhfmr.github.io/THORIN/> and the instructions seem to be clear to me.

Referee #4

(Remarks to the Author)

Many thanks to the authors for their thorough responses to my previous concerns. With the substantial changes, particularly to the results and discussion sections, I think the manuscript is a lot easier to digest and the key messages are more prominent. I have a couple of very minor questions:

1. Thank you for including Sup Fig 15, I think this makes the selection of UKB participants throughout the pipeline much clearer. However, a couple of the numbers don't seem to add up; individuals with at least one sibling (N=14,695) is less than the number of PofO assigned from intra-chrs sib score (N=14596) and the PoO un-assigned (N=315). Would you please be able to clarify?
2. Lines 254-257: It was nice to see the ORs for the maternally and paternally inherited alleles on T2D, as it shows the size

of the effect and can be more easily compared to the additive effects from previous GWAS. I wonder whether it would be worthwhile including the effect size estimates for the other described associations in the previous three sections (growth and metabolic traits/lipid metabolism and hip circumference/fat distribution)?

3. The legend for fig. 7 states that 7a) is for BMI and 7b) is of height, but the actual plot seems to be the other way around.

4. The shortened discussion is much more concise; however, I feel that a concluding statement is missing and think it would be valuable for readers to get one or two of take-home messages from the manuscript.

5. Some of the supplementary figures are not correctly referenced. For example, line 643 has "(Supplementary Figure ??A)". Same in the supplementary notes.

(Remarks on code availability)

Response to referees

We sincerely thank all the reviewers for their thoughtful and constructive feedback on our manuscript. We appreciate their careful evaluation, which has significantly contributed to refining our study. Their comments have helped us clarify key aspects of our methodology, substantially improve the presentation of our results, and ensure that our findings are robust and clearly communicated.

We acknowledge the global reviewers' concern regarding the length of the manuscript. Given the scope of our study—introducing a novel parent-of-origin (PofO) inference method, applying it at scale in multiple biobanks, and analyzing both complex traits and protein QTLs—it was challenging to strike the right balance between comprehensiveness and length. However, we fully recognize the importance of making the manuscript more accessible to a broad readership.

To address this concern, we have substantially revised the manuscript to enhance readability and streamline the presentation:

1. **Focusing the main text on key findings:** We prioritized the most significant results and removed exploratory findings or secondary associations from the main text. This ensures that the most impactful contributions of the study stay in focus.
2. **Relocating detailed analyses to supplementary materials:** Many technical details, secondary findings, and in-depth methodological comparisons have been moved to the Supplementary Notes. This allows interested readers to explore specific aspects while keeping the main text concise.
3. **Condensing the discussion:** We have refined the Discussion section to focus on the broader implications of our findings, while moving detailed technical discussions to the Supplementary Notes.
4. **Enhancing readability:** We have carefully revised the writing to improve clarity, remove redundancies, and ensure that key messages are conveyed in a more concise and accessible manner.

We believe these revisions have substantially improved the manuscript's readability and impact, while still making all details available in the Supplement for the interested readers. We again thank the reviewers for their insightful comments, which have helped us refine and strengthen this work. We hope that the revised manuscript fully addresses their concerns and look forward to their feedback.

As requested by the journal, we include a version of the manuscript with track changes. However, given the extensive modifications made to shorten and streamline the text, this version is difficult to read. Therefore, we also provide a clean version of the updated manuscript without track changes for ease of reference. Please note that all line numbers mentioned in our response refer to the clean version of the manuscript without track changes.

Referee #1/2 (Remarks to the Author):

We start by noting that this report is prepared by two people.

The authors have extended and improved a method they developed for determining the parent-of-origin of alleles of genetic markers when a genealogy is unavailable, which is often the case, particularly with large populations. Applying their method to the UK Biobank and the Estonia Biobank, they separated out the paternally and maternally inherited haploid genomes for 123,716 and 97,346 individuals respectively. Together with 45,402 offspring from the Norwegian Mother, Father and Child Cohort Study, they were able to replicate a substantial number of previously reported genetic markers with parent-of-origin effects (POEs), and also revealed some possible false positives. Importantly, they have also discovered novel POEs, many exhibiting bipolar dominant behavior, i.e. both the paternally and maternally inherited allele have effects, but in opposite directions.

Overall, we are impressed with this work. Worth emphasizing is that PO information is not only useful for identifying POEs, but could play a key role in many other areas of genetic research. Moreover, the effectiveness and yield (percentage of samples for which PO information can be determined) of the method can improve as datasets become larger (Specific Comment 2).

In contrast to what the authors have achieved, we feel that the writing and presentation are somewhat uneven. One example is the paragraph that starts with (line 229) “We identified novel associations with type 2 diabetes (T2D), HbA1c, rs10838787 and rs4417225,”, and ends with “Although these variants are novel, they are in high LD ($r^2 > 0.9$) with a variant previously associated with T2D in a parent-of-origin manner (ref 5). If the variants identified here are in such high LD with a previously reported variant, how is it a novel association? Calling the variants novel would also be misleading. Novelty aside, there is the unaddressed question of whether the two variants reported here could be a refinement of the original SNP reported, which we believe is rs2334499. Interestingly, rs2334499 falls within 350 Kb of a large cluster of imprinted genes, but was not considered to be within an imprinted region, at least in 2009. Maybe we are not being completely impartial, but this association is particular worth noting for several reasons. T2D has more impact on health than many of these other associations. For individuals with European descent, the variance of T2D risk explained by this SNP may only be second to the strongest variant TCF7L2. The opposite direction effects observed might be what motivated ref 14 to coin the term bipolar dominance. Maybe most important to the method described here, T2D is a late onset disease so many of the parent-offspring trios ascertained would not be useful. Indeed, we are not aware of a strong replication of this association since the original publication, and thus the strong results reported here are important even if not entirely novel.

Response:

We sincerely appreciate the reviewers’s feedback regarding the clarity, novelty, and importance of our findings. Regarding the type 2 diabetes (T2D) POE at the 11p15.5 region, the reviewer raises a crucial point regarding the interpretation of novelty in the presence of strong linkage disequilibrium (LD) with previously reported variants. We have carefully revised our manuscript to address these concerns, ensuring that our findings are framed

appropriately as a refinement and confirmation of the previously reported parent-of-origin (POE) association rather than a novel discovery.

1. Clarifying the refinement rather than novelty of POEs at 11p15.5

In our original text, we referred to our findings as novel POE associations, but we acknowledge that this phrasing was potentially misleading, given that our lead variant rs10838787 is in high LD ($r^2 > 0.9$) with rs2334499, a previously reported POE-associated variant. We have now explicitly reframed our findings as a refinement of this association rather than a new discovery in the Results section (line **258**):

“Although the imprinting status of this locus is not yet established, our lead-POE variant is located within 350 kb of a well-characterized imprinted gene cluster, and is in high LD ($r^2 > 0.9$) with rs2334499, a variant previously reported to exhibit a similar bi-polar effect on T2D⁶. Given the strong LD between these two T2D risk variants, we performed a conditional analysis, which identified rs10838787 as a more likely causal variant in the UKBB cohort (Supplementary Notes 6.7). Thus, our result refines rather than introduces a novel association at this locus, and provides robust confirmation of a bi-polar POE on T2D, an effect that had not been strongly replicated since its initial report⁶.“

By clarifying these points, we ensure that our presentation is precise, avoiding any misinterpretation of novelty.

2. Emphasizing the importance of this refinement in the context of T2D

The reviewer highlighted that rs2334499 is a key variant in T2D risk, potentially second only to TCF7L2 in individuals of European descent. We recognize the importance of this association and have now added in the Result section (line **265**):

“Importantly, this locus accounts for a substantial proportion of T2D risk in individuals of European ancestry, potentially ranking among the SNPs with the largest impact on T2D. Indeed, the odds of T2D are 1.25 times higher (95% CI: 1.16-1.35) when the A allele of rs10838787 is paternally vs maternally inherited (compared to OR~1.4 for rs7903146, intronic to TCF7L2, one of the most influential genetic factors for T2D²⁰)”

3. Addressing the concept of late-onset T2D

The reviewer correctly highlighted that T2D is a late-onset disease, which presents a challenge for traditional parent-offspring trio studies. In these studies, offspring may be too young at the time of data collection to have developed T2D, leading to underpowered analyses for detecting POEs on this trait. We emphasize now that our method is much less impacted by such limitations.

To address this point, we have added a dedicated discussion paragraph highlighting how our approach—leveraging large biobank datasets—overcomes these limitations and is particularly well-suited for detecting POEs in late-onset diseases like T2D. This addition underscores the advantage of our method in identifying POEs in traits where traditional trio-based designs may lack sufficient power. In addition, we also emphasize the importance of confirming and refining previous findings with stronger statistical evidence and larger datasets (line **419**):

“The key advantage of our approach - its applicability to biobank-scale data - enables PofO inference across individuals of all ages. This distinguishes it from trio-based studies, where parent-offspring trios are typically recruited together, resulting in a cohort predominantly

composed of young offspring. Therefore, our method enables the analysis of late-onset phenotypes, such as T2D, for which we confirmed and refined a POE at 11p15.5 locus, which went unreplicated for over 15 years.”

4. Strengthening the Interpretation of the KCNQ1 Locus

We also revised our text on rs2299620, a suggestive POE identified at the KCNQ1 locus, to ensure that the results are framed as a refinement rather than a novel discovery (Results section, line 278):

“In addition to the above-mentioned associations in the 11p15.5 region, we identified a suggestively significant maternal-only effect of rs2299620 on HbA1c, intronic to KCNQ1 (Figure 5C). This variant was in high LD with rs2237892 ($r^2 = 0.5$), another T2D-risk variant previously reported⁶. Consistently, we also found a moderate protective effect of the maternal T allele of rs2299620 on T2D ($OR_{mat} = 0.69$, [0.60–0.79]). In addition, conditional analysis confirmed rs2299620 as the more likely driver of the POE on HbA1c (Supplementary Note 6.7).”

These clarifications ensure that our presentation is aligned with the reviewer’s concerns about accuracy in describing novelty, clarity in interpretation, and the significance of this work within the broader field of T2D genetics.

As noted above, PO information is useful for many things other than POEs. As such, to maximize the impact of this manuscript, we hope that the authors would (a) do their best to make it easier for people to implement the method themselves and (b) to make the PO information on the UK Biobank samples available. Their tool THORIN is available on github with documentation. We understand that the authors are currently working on (a). That is, the authors are working on a tutorial to describe their entire parent-of-origin pipeline with the 1000 Genome Project as example data. We look forward to the completion of that tutorial and will likely try to use THORIN ourselves in the near future. With (b), one possibility is to provide the information through the UKBB research analysis platform. However, that might not be easy or may take too much time. We do appreciate that it is very difficult to provide scripts that individual UKBB user can simply run and reproduce the PO information. But providing user friendly code of the method and any tips regarding the processing of the UKBB data on the research analysis platform would be very welcomed.

Response:

We thank the reviewers for their interest in our parent-of-origin inference method and their enthusiasm for implementing it in future research. We fully agree that making the method as accessible as possible will maximize its impact beyond the study of POEs.

1. Providing a comprehensive tutorial for publicly available data

To facilitate broader adoption of our method, we completed the detailed tutorial that documents each step of the parent-of-origin inference pipeline, using publicly available data from the 1000 Genomes Project as an example. This will allow researchers to understand and reproduce the full process while adapting it to their own datasets. We are committed to ensuring that this tutorial is comprehensive and user-friendly, and we appreciate the reviewer's interest in using THORIN.

2. UK Biobank data availability and efforts to facilitate access

Unfortunately, due to UK Biobank data security policies, we are unable to transfer the inferred parent-of-origin data from our local secured HPC environment to the UK Biobank Research Analysis Platform (RAP). Additionally, returning the data through the official UKB data return process is not feasible, as the inferred genetic data is too large to meet their current data return requirements. For example, the inferred “differential” genetic data for chromosome 20 alone exceeds 20GB, and we also generated separate copies of maternal and paternal haplotypes, which are even larger. Given the size and complexity of these datasets, transferring them through the standard UKB data return channels is not possible under current constraints.

However, to address this, we have contacted the UK Biobank team to inform them that we have inferred parent-of-origin data available for all approved researchers and are willing to return it through an official channel if possible. In the event that direct return is not feasible, we have also asked whether UKB could provide funding support for us to reproduce the inference directly on the RAP platform, which would facilitate data sharing with all eligible researchers.

The manuscript is currently very long. Among other things, we believe the paper would read much better if the later part of the POEs results are mostly moved to the Supplement. These results are just not that interesting, we believe, for the general reader. Many of them are either borderline significant, insubstantial in effect and/or the trait is relatively unimportant. Unintentionally, they could actually dilute the impact of the genuinely impressive results and let the reader feel a little letdown when finishing. The Discussion section should probably also be shortened. By contrast, without necessarily increasing the length substantially, some additional details about the method, and the accuracy and yield of PO information, could be helpful.

Response:

We appreciate the reviewers's constructive feedback on the manuscript length and readability. We fully agree that keeping the focus on the most impactful findings will strengthen the paper and improve the reader's experience.

1. Condensing the Results section

To maintain focus on the most significant and substantial POEs, we have:

- Kept only the strongest POE results in the main text, prioritizing those that meet genome-wide significance and have clear biological relevance.
- Moved borderline-significant or lower-impact associations to Supplementary Notes, ensuring that readers remain engaged with the most compelling discoveries.
- Reorganized the results for better readability, making the progression of findings clearer and more impactful.

2. Shortening the Discussion Section

We have also streamlined the Discussion section by:

- Focusing on the key implications of our findings while moving more detailed points to Supplementary Notes.

- Retaining only the most relevant methodological considerations while keeping technical discussions in the Supplement for interested readers.
- Ensuring a strong conclusion that highlights the major contributions of our study without unnecessary length.

3. Expanding on methodology and POE accuracy

As suggested, we have added additional details about the method, including yield of PofO information (see **Supplementary Figure 15**, referenced in line **126**) and accuracy of the method (see **Supplementary Note 1 and 2**, referenced in lines **95** and **133**, and the **Supplementary Figure 5** associated to **Supplementary Note 1**), without significantly increasing main manuscript length. This ensures that readers gain a clearer understanding of the method's reliability and advantages, while keeping the main text concise.

Specific Comments

1. Lines 47 to 55. While the authors overcome the lack of genealogical information by clever use of the information from X chromosomes and mitochondrial DNA, it seems that the conceptual relationship with the method described by reference 5 should be made more explicit. After all, the concept of 'surrogate parenthood' in this context was, we believe, introduced in the long-range-phasing paper, the prequel to ref 5 which did not directly make use of the genealogy. A natural way to understand the authors' approach is to use the information from the X chromosomes and mitochondrial DNA to partially reconstruct the genealogy. The use of sex-specific recombination maps for this purpose with sibling data is however distinct, and something we are not aware of before.

Response:

We thank the reviewers for their comment and fully agree that the two referenced works were pioneering in the development of surrogate parenthood approaches and should be more explicitly acknowledged in our manuscript.

In response, we have revised the introduction to better highlight these foundational contributions, and further emphasize how our method builds on these ideas (line **47**):

“Studying POEs at genome-wide scale requires large cohorts with parent-of-origin (PofO) information, traditionally derived from parent-offspring trios or duos⁴. An alternative approach uses close relatives as surrogate parents to estimate haplotypes⁵, which allows PofO inference when combined with genealogical data⁶. However, large-scale biobanks often lack both parental genomic data and detailed genealogies⁷. Building on the surrogate parent concept, our previous work introduced a method leveraging chromosome X sharing to identify maternal relatives without genealogical information³, which opened new research avenues and significantly increased the sample size. However, this method was limited to male participants constraining sample size and hence reducing statistical power.”

2. The question of (percentage) yield for the UKBB samples. From the main text, if we understand correctly, 274,525 (line 89) have one or more surrogate parents identified, and at the end reliable PO information for close to the entire genome are determined for 123,716 individuals. But a higher number of 286,666 is mentioned on line 126. The question is what is the right denominator for percentage yield as the UKBB data set is clearly substantially larger than the numbers mentioned. Knowing the current UKBB percentage yield can allow us to project how it could improve as the number of UK genotyped individuals increases, due to having a larger number of surrogate parents per individual.

Response:

We thank the reviewers for this constructive comment. We agree that the current description of the number of individuals for whom PofO could be inferred made it difficult to project our method's applicability to other datasets (e.g., Our Future Health). This concern was also raised by Referee #4 (comment #4), who suggested adding a supplementary figure for clarity, as well as by Referee #3 (comment #1).

To address this, we have added **Supplementary Figure 15** (referenced in line 126), which provides a detailed overview of the inference process.

- Panel A illustrates the number of individuals at each step of the analysis.
- Panels B and C show the proportion of individuals for whom PofO can be inferred (y-axis) based on the availability of relatives (x-axis).
- Panel B presents results when sequencing data is available, allowing the use of mtDNA as an additional predictor.
- Panel C shows the scenario where only SNP array data is available, meaning PofO inference primarily relies on chromosome X sharing and sibling-based crossover information.

These additions provide a clearer framework for estimating PofO inference yield in different datasets, depending on the availability of relatives per individual. This should facilitate projections for large-scale biobank initiatives, such as Our Future Health, as their genotyped sample size continues to grow.

3. In the summary of the methods in the main text, the key steps start with inter-chromosomal phasing. The question is what about intra-chromosomal phasing which we assume is done using SHAPEIT5 (line 923). The legend of Fig. 1 mentions correcting for intra-chromosomal errors. It is of interest what is the intra-chromosomal phasing error rate to start with, and what complications that creates for the whole process.

Response:

Thank you for your insightful comment regarding the comparison of inter-chromosomal phasing and intra-chromosomal phasing, and how it improves the accuracy of parent-of-origin (PofO) inference. We acknowledge that the initial manuscript did not fully highlight the benefit of inter-chromosomal phasing beyond switch error rate (SER) reduction. A more relevant measure for our association study context is the fraction of the genome that is correctly phased. To address this, we have now added a dedicated **Supplementary Note 1** (referenced in line 95) and the associated **Supplementary Figure 5**, where we directly compare intra- and inter-chromosomal phasing in a subset of 548 UK Biobank individuals with both close relatives (enabling inter-chromosomal phasing) and both parental genomes (allowing direct validation

of phasing accuracy). This analysis revealed that while inter-chromosomal phasing only marginally reduced the SER (from 0.15% to 0.14%), it had a substantial impact on correcting large-scale misphased haplotype blocks. Specifically, we found that:

- 6.24% of the genome was incorrectly phased with intra-chromosomal phasing.
- Inter-chromosomal phasing reduced this to 3.82%, nearly a two-fold reduction in the proportion of misphased genome-wide segments.
- When restricting to chromosomes where IBD-sharing relatives were available, the error rate was further reduced to 2.53%.

While the improvement in SER may seem small in absolute terms, it is particularly critical for PofO inference. The largest phasing errors occur over long haplotype segments, meaning that intra-chromosomal phasing alone can result in entire chromosomal regions being assigned to the wrong parental haplotype. This can lead to systematic misclassification of maternal and paternal alleles, affecting POE detection and downstream associations. By leveraging IBD information across autosomes, inter-chromosomal phasing corrects these large-scale phasing errors, significantly improving the accuracy of PofO inference and downstream POE analyses.

Additionally, since the reviewer specifically referenced Figure 1, we have updated its legend to explicitly refer to the **Supplementary Note 1**, ensuring that readers are directed to the more detailed explanation of inter-chromosomal phasing and its benefits.

4. Heritability. h^2_{POE} (line 504) should be explicitly defined. We presume it is the variance explained by POEs in addition to that of an additive effect model. If so, it looks rather modest (Figure 6c). It seems that should be the first point to make. If indeed the overall POE heritability is quite modest, the difference between paternal and maternal, and imprinted regions and not, are secondary issues. For the main text, it seems that a short paragraph would suffice.

Response:

We appreciate the reviewers's feedback on clarifying h^2_{POE} and improving the presentation of its results. In response, we have:

1. Explicitly defined h^2_{POE} in the main text (line 344):

“ Specifically, h^2_{POE} represents the variance explained by differential parental effects at heterozygous sites, beyond that of the additive model.”

2. Reorganized the section to emphasize that h^2_{POE} is modest overall (line 347), before discussing differences in maternal vs. paternal contributions and enrichment within imprinted regions. This adjustment aligns with the reviewer's suggestion that the primary takeaway should be the overall modest contribution of POEs to heritability, with secondary discussions following. Specifically, we now say:

“ Across most traits, h^2_{POE} was modest compared to additive effects”

3. Moved more detailed results (e.g., comparisons between maternal and paternal SNP heritability, specific trait-level patterns) to **Supplementary Note 11**, while keeping

the main text focused on the key findings. This improves readability and prevents dilution of the central message.

4. Clarified statistical significance considerations in the **Supplementary Note 11**. While some traits showed nominally significant differences between maternal and paternal heritability estimates, these did not remain significant after correction for multiple testing, indicating that larger datasets may be needed to confirm these differences.

Together, these revisions ensure that the key conclusions are clear and appropriately contextualized, while also allowing readers to explore more details in the supplementary materials if desired.

5. The authors noted on page 6 that there is currently no consensus on how to identify POEs. Here, the authors have a great opportunity to set a rigorous standard. Achieving genome-wide significance for at least one of the association tests could be a useful criterion. The study presented some strong results, but including suggestive signals in the main text dilutes that message and having various significance thresholds makes the paper more difficult to read. In our opinion, the paper would benefit from moving the weaker/suggestive results to the supplement. Especially, the sub-chapter “Parent-of-origin specificity of additively associated regions”. The rationale for using a more lenient significance threshold for variants in imprinted regions is more understandable. However, it is noted on page 40 in the paper, that the significance threshold for the variants in imprinted regions should be $6.27E-8$ when the number of phenotypes is taken into account. For results presented in the main text, this threshold could be more appropriate. Suggestive signals could be shown in the supplement.

Response:

We appreciate the reviewer’s suggestion to establish a rigorous standard for identifying POEs and to ensure that the significance thresholds used in our study are clear and consistent. We fully agree that setting a standardized approach for POE detection will strengthen the field and improve comparability across studies, and we thank the reviewer for this suggestion.

1. Establishing a standard for POE identification

To address this, we have now included a dedicated section in the main text ("Establishing a Standard for Identifying Parent-of-Origin Effects", line **146**), where we formally define a uniform statistical framework for POE detection. Specifically, we propose that:

- A true POE should be defined based on a significant differential test (P_D), ensuring that maternal and paternal effects differ significantly rather than relying on arbitrary thresholds applied to each parental allele separately.
- A genome-wide, or region-focused, significance threshold should be applied for POE detection, ensuring findings are identified with the same level of confidence as traditional GWAS signals.
- For previously reported POEs, where P_D values may not have been explicitly tested, we propose a Z-score approach to retrospectively assess the difference between maternal and paternal effects.

We believe that by implementing this consistent methodology, our study not only strengthens the reliability of POE discoveries but also provides a framework for future studies in the field.

2. Application of a strict Genome-wide significance threshold

To ensure clarity and consistency in the presentation of results, we have followed the reviewer's suggestion and now only report loci in the main text where at least one primary POE was identified at a significance threshold of $P_D < 6.27 \times 10^{-8}$, as determined by both the number of variants and phenotypes tested. All other suggestive findings that did not meet this threshold have been moved to Supplementary Notes.

This revision ensures that:

- The main text remains focused on the most robust POE associations, avoiding dilution of key findings.
- The significance thresholds applied are consistent across analyses, addressing concerns about multiple thresholds making the results harder to interpret.
- Findings at a more lenient threshold, referred to as “suggestive associations” (such as those in the additively associated regions analysis) are still reported but in supplementary materials, where they can be appropriately discussed, unless they are directly related to a primary POE discussed in the main text.

3. In addition to this, we took the opportunity to also set a standard on how to classify POE patterns, based on the relative magnitude and direction of the maternal and paternal effects. We now mentioned this into the main text (line **158**) and detailed the procedure in the method section (section “Classification of parent-of-origin effects”, line **903**).

By implementing these revisions, we believe that we have addressed the reviewer's concerns regarding standardization, significance thresholds, and clarity of the results.

6. Supplementary table 2 depicts replication results of previously published POEs. On closer examination, we believe that there are only 5 POE height signals in supplementary table 2, not 8. If we are not mistaken, the three variants that are shown under Zoledziewska et al. (2015) are all part of the same height signal which was replicated and refined to rs143840904 in 2016 (variant shown twice in the table). As a result, it would be more correct to say that the table is showing 19 previously reported POEs, of which 9 are replicated in the current study.

Response:

Thank you for your careful review of Supplementary Table 2 and for pointing out that three of the height-related variants reported in Zoledziewska et al. (2015) correspond to the same signal, refined to rs143840904 in a later study. In addition, we also reported two snp-trait pairs that represent the same associations with platelet count ($R^2=0.5$). We agree with this assessment and have made the necessary adjustments to ensure accurate reporting of the number of independent signals.

Specifically, we revised our replication section to state that we identified 18 independent previously reported POEs (22 reported associations), rather than 22 independent POEs. We also updated our replication rate accordingly (line **164**):

“We identified 18 independent known POEs (22 reported associations) meeting this criterion that could be tested for replication in our cohort (same variant-trait pair). We successfully replicated 8 of them (44.4%) at $P_D < 0.05/18$.”

To improve clarity in **Supplementary Table 2**, we:

- Bolded replicated associations.
- Marked in superscript (¹ and ²) SNP-trait pairs corresponding to the same association.
- According to Referee #3 (comment #2), we indicated by an asterisk (*) studies for which the differential p-value was not available, and re-computed by us.
- Updated the table legend to clearly describe these adjustments.

Referee #1/2 (Remarks on code availability):

We have not yet tried out the code, but plan to do so in the near future.

Referee #3 (Remarks to the Author):

In this manuscript, Robin et al., proposed a method that combines inter-chromosomal phasing, mitochondrial and chromosome X data, and sibling-based crossover evidence to determine parent of origin inheritance of alleles in samples without parental genome information available. They applied the method to obtain parent-of-origin allele information in $\frac{1}{4}$ samples in UKBiobank, which allowed them to perform POE-specific association tests for 59 complex traits. They also replicated the application of the method and the identified associations in two external cohorts.

In my view, this POE-inference method is both **innovative and holds promising application potential**, given that the sample sizes of biobanks globally increase quickly but parental genome information is often incomplete or lacking. I also think the authors did a **great job** to show the practical usage of the method in identifying and replicating POE-specific associations.

In addition to these, **I have some concerns on the rationale of applying several methods designed for additive genetic effects in a study of POE**. Those include LDSC-based SNP heritability analyses and co-localization analyses, both heavily relying on the LD structure in the human genome. When these methods are applied to a POE study that essentially aims to differentiate genetic effects between maternal and paternal chromosomes, I think we need to consider the LD difference between parental chromosomes and the influence of such difference. It is well known that crossover (recombination) is one of the driving factors to shape the LD structure. And importantly, the crossover event can vary between sex (it is also based on this the authors proposed their crossover-inference strategy to assign parent-of-origin alleles using sibling pairs). Given the crossover positions that vary between genders, LD structure between parental chromosomes could vary too. These differences are averaged out when using unphased data in analyses for additive genetic effect, but should not be neglected in POE studies.

For example, in heritability analysis, the authors compared the heritability contributed from maternal and paternal haplotypes. Were the LD difference between parental chromosomes accounted in the calculation? How was the parent-of-origin SNP heritability calculated? In colocalization analysis, how was the maternally/paternally varied LD structure accounted?

Response:

We thank the reviewer for their concern regarding the potential influence of sex-specific recombination on LD structure and its implications for POE analyses. Indeed, differences in male and female recombination rates could, in principle, lead to deviations in LD structure between maternally and paternally inherited haplotypes. However, each individual's haplotype is a recombined product of multiple past generations, meaning that paternal haplotypes have undergone recombination not only under the male-specific map in the most recent generation but also under both male and female recombination maps in earlier generations. This results in an effective LD structure that is largely averaged across sexes over time.

To empirically validate our theoretical reasoning, we computed LD separately for maternally and paternally inherited haplotypes in the UK Biobank dataset. Our analysis demonstrated a strong correlation between maternal LD and paternal LD, with only minor deviations. These deviations were comparable to the variation observed when computing traditional LD (from

unphased genotype data) across two randomly sampled groups of 100,000 individuals, suggesting that any systematic differences between parental LD structures are compatible with the amount of noise present in these estimates. This has been included in **Supplementary Note 21** and **Supplementary Figure 29**.

Beyond empirical validation (**Supplementary Note 21.1**), we also derived a theoretical framework (**Supplementary Note 21.2**) to formally test whether the LD structure of differential genotypes remains equivalent to that of traditional additive genotypes. Our derivations demonstrate that, under realistic assumptions, the correlation structure of differential genotypes follows the same LD properties as traditional additive coding. These findings confirm that standard LD reference panels can be used without modification for POE-specific heritability estimation using LD Score Regression (LDSC) and colocalization analyses.

In response to this comment, we have updated our methods section to explicitly refer to the supplementary note that clarifies why standard LD scores are appropriate for POE analyses (lines **942**):

*“We showed that the LD based on the parental dosage difference equals the LD of additive genotype dosage (see **Supplementary Note 21**).”*

My other comments are the following

1. The readership should be interested in seeing the estimated sample size required to reach reliable parent-of-origin inference using each of three methods under different data structure settings.

Response: We appreciate this valuable suggestion, which enhances the clarity of our method’s applicability under different data structure settings. A similar point was also raised by Referee #1-2 (comment #2) and Referee #4 (comment #4). As mentioned in our response to Referee #1-2, we have now added **Supplementary Figure 15** (referenced in line **126**) to provide a detailed assessment of the estimated sample size required for reliable parent-of-origin inference using each predictor (i.e., chromosome X sharing, mtDNA, and sibling-based crossover analysis).

- Panel A illustrates the number of individuals at each step of the analysis.
- Panels B and C show the proportion of individuals for whom PofO can be inferred (y-axis) based on the availability of relatives (x-axis).
- Panel B presents results when sequencing data is available, allowing the use of mtDNA as an additional predictor.
- Panel C shows the scenario where only SNP array data is available, meaning PofO inference primarily relies on chromosome X sharing and sibling-based crossover information.

2. Line 153-155: “Therefore, we first pre-filtered claimed POEs by rigorously testing for a differential paternal vs maternal alleles effect and kept only published hits where the differential test P-value PD was lower than 10^{-3} , to ensure more robust evidence POE.”

Is the PD provided in original publication, or recalculated by the authors using original samples from those studies, or calculated using samples from current study?

Response:

Thank you for raising this point. The differential p-value (P_D) used in our replication analysis was either directly taken from the original publications when available or re-computed by us when it was not explicitly reported.

For studies that did not provide P_D , but did report maternal- and paternal-specific effect estimates along with their standard errors, we applied a Z-score approach to formally test for differential parental effects. These cases are now clearly indicated in Supplementary Table 2 with an asterisk (*).

3. Line 1302: height studies. Eight studies?

Response:

Thank you for catching this typo. The correct wording should indeed be "eight studies" rather than "height studies." We have corrected this in the revised manuscript.

4. Line 158: "We successfully replicated 12 of them (54.5%) at $P_D < 0.05/22$. Interestingly, of the 8 POEs reported for birth weight, none were replicated."

Any reason for that? Was the birth weight adjusted by gestational age?

Response:

Thank you for your insightful comment regarding the lack of replication for previously reported POEs on birth weight. Several factors likely contributed to this outcome:

1. Differences in birth weight data collection: In the original studies, birth weight was reported by the mother for her offspring, which is generally considered reliable. In contrast, in the UK Biobank we used data on self-reported (own) birth weight via a touchscreen questionnaire at ages ~40-70. This retrospective self-reporting introduces recall bias and increases measurement noise, which can reduce the power to detect genetic associations, particularly for subtle POEs.
2. Lack of gestational age adjustment: Some birth weight studies adjust for gestational age when maternal pregnancy records are available. However, UK Biobank did not collect information on participants' own gestational age, preventing us from incorporating this important adjustment. Notably, even in the original study that reported POEs on birth weight using Icelandic data, the authors acknowledged that while their Icelandic birth weight data were adjusted for gestational age, the UK Biobank birth weight data were not, due to the limited availability of gestational age information (Juliusdottir et al., Nature Genetics 2021). This highlights a general limitation of using UK Biobank for birth weight analyses, even in additive GWAS.

Given these limitations, we believe that the lack of replication of POEs for birth weight in UK Biobank is primarily due to the noisier phenotype rather than a fundamental inconsistency with prior findings.

To provide further details, we have added **Supplementary Note 4** (referenced in line 167), where we elaborate on these challenges and their implications for POE detection in UK Biobank.

5. Figure 2b: legend suggests “Unlabeled points correspond to bi-polar POEs” however in the figure itself “ZNF597, KFL14” were labeled but seem to have a bi-polar POE pattern?

Response:

We appreciate the reviewer’s observation regarding the labeling of POEs in Figure 2B.

As there is no widely accepted consensus on how to classify POEs, we applied an approach based on the relative magnitude and direction of maternal and paternal effects, as now detailed in the Methods section (line 903):

903 **Classification of parent-of-origin effects**

904 To classify POEs, we categorized associations based on the relative magnitudes and direc-
905 tions of the maternal (Z_{MAT}) and paternal (Z_{PAT}) Z-scores. The classification criteria are
906 as follows:

- 907 • Bi-polar Effects: The maternal and paternal effects are of opposite signs ($\text{sign}(Z_{MAT}) \neq$
908 $\text{sign}(Z_{PAT})$) and have similar magnitudes ($|Z_{MAT}| > 0.5 \times |Z_{PAT}|$ and $|Z_{PAT}| >$
909 $0.5 \times |Z_{MAT}|$).
- 910 • Maternal Effect: The maternal Z-score is at least twice as large as the paternal Z-score
911 in absolute value ($|Z_{PAT}| < 0.5 \times |Z_{MAT}|$).
- 912 • Maternal Asymmetric: The maternal and paternal effects have the same sign, the ma-
913 ternal effect is dominant ($|Z_{MAT}| > |Z_{PAT}|$) but the paternal effect is still substantial
914 ($|Z_{PAT}| > 0.5 \times |Z_{MAT}|$).
- 915 • Paternal Effect: The paternal Z-score is at least twice as large as the maternal Z-score
916 in absolute value ($|Z_{MAT}| < 0.5 \times |Z_{PAT}|$).
- 917 • Paternal Asymmetric: The paternal and maternal effects have the same sign, the pa-
918 ternal effect is dominant ($|Z_{PAT}| > |Z_{MAT}|$), but the maternal effect remains notable
919 ($|Z_{MAT}| > 0.5 \times |Z_{PAT}|$).

To ensure clarity and accuracy, we have now added this classification in Figure 2 (colored areas) and in Table 1. Using this classification, associations such as near ZNF597 and KLF14 did not meet our predefined threshold for bi-polar effects, despite showing some level of opposing parental influences.

6. Line 384-386: “the additively associated regions approach, which assesses the POE specificity of the lead additive variant but may not always pinpoint the true lead POE variant.” Why the leading additive variant is different from the leading POE variant, if the additive model just dilute/average out the POE signal?

Response: The lead additive variant and the lead POE variant may differ because the variant with the strongest additive association is not necessarily the one with the strongest POE. Several factors contribute to this discrepancy:

1. Discrepancy due to estimation error

In any GWAS in general, effects are estimated with noise, hence the lead variant in two GWASs can be different even if there is one common causal variant. The LD between the two detected lead variants increases as the power in both studies to detect this variant increases. In our case, there are several scenarios that can lead to that situation: (i) If a variant has paternal and maternal effects in the opposite direction, the additive model captures only the average of these effects and hence will have low power. (ii) If the paternal and maternal effects have only a small difference, the POE will have low power.

2. Existence of distinct additive and POE effects

Allelic heterogeneity (when multiple independent variants contribute to a trait) is an extremely widespread feature of genetic associations. It is possible that at a given locus there are two signals in LD but contributing independently to a trait. One of these variants may have a stronger additive effect, while the other one has a stronger POE.

7. Line 707-709, “These associations are particularly notable in the context of bi-polar dominance, as the lead variants act as eQTLs for both maternally expressed genes (KLF14 and CPA4) and the paternally expressed MEST” and line 712-713, “This interplay between maternally and paternally expressed genes exemplifies the intricate regulatory mechanisms underlying bi-polar POEs”.

1) Were those eQTL analyses performed based on additive model?

2) Do these genes interact with one another, and do they exert activating or repressive effects on the phenotype? Observing significant associations between maternal and paternal alleles with different genes does not necessarily imply that they have opposite or distinct functional roles, and therefore does not adequately support the conflict theory of bipolar-type parent-of-origin effects.”.

Response:

We appreciate the reviewer’s careful consideration of this point and their request for clarification.

1. Yes, the eQTL associations are derived from publicly available datasets (GTEx, eQTLGen) that primarily assess eQTLs under an additive model. To clarify this, we have

now explicitly stated in the text that the reported eQTLs were identified under an additive model (see point 2). Since the vast majority of POE acting on gene expression represents a scenario whereby one of the two effects is zero (either maternal or paternal), the additive effects still can pick up POE signals.

2. We acknowledge the reviewer's concern regarding the interpretation of these findings. Our original wording may have unintentionally suggested a direct regulatory relationship between *KLF14*, *CPA4*, and *MEST*. To address this, we have revised the discussion section (line 461) to clarify that while these variants influence multiple imprinted genes with opposite parental expression patterns, the precise regulatory interactions between these genes remain unclear. Of note, to exert an antagonistic effect, it is not necessary that these genes interact in any way, we only claim that there is a dual effect of the same variant on two genes and we hypothesise that the expression levels of these genes are likely mediators of the variant's effect:

“Our follow-up analysis at the 7q32.2 imprinted region - exhibiting bi-polar POEs on triglycerides, HDL-C, and SHBG - revealed a possible explanation for this bi-polar phenomenon, namely the lead POE variant being an eQTL for maternally expressed KLF14 and CPA4 and also an eQTL for the paternally expressed MEST under an additive model. If those genes have opposite effects on the associated complex traits, such gene-expression mediated effects may explain certain bi-polar patterns for complex traits, suggesting a rather indirect antagonism. These findings underscore the complexity of POEs and the need for functional studies to elucidate imprinting mechanisms in metabolic traits.”

8. Lines 167-169: “traits related to growth and metabolism were particularly over-represented among pleiotropic loci, notably on chromosomes 7, 11, and 20 (Figure 2A), consistent with the expected roles of imprinted genes under the conflict theory.”

Conclusion of overrepresentation or pleiotropy should consider the fact that the number of phenotypes tested here are limited and those phenotypes are highly selective.

Response:

We thank the reviewer for his/her comment regarding the potential influence of phenotype selection on our conclusions about trait overrepresentation. To address this concern, we have refined our analysis by explicitly classifying all tested traits into broader functional categories: growth, metabolism, and other (**Supplementary Table 1**). We then formally tested for enrichment of POEs within these categories to assess whether the observed concentration of POEs in growth- and metabolism-related traits was statistically significant (lines 181), and added the details of this analysis in **Supplementary Note 5**:

*“To formally assess the enrichment, we assigned all tested traits into growth, metabolism, and other categories (Supplementary Table 1) and tested for POE enrichment. The analysis confirmed a significant overrepresentation of POEs in growth- and metabolic-related traits compared to additive effects (OR = 5.35, $p = 0.018$, **Supplementary Note 5**), reinforcing the expected role of imprinted genes in these processes.”*

These additions provide a more rigorous framework for evaluating the overrepresentation of POEs in growth- and metabolism-related traits while addressing the potential biases introduced by phenotype selection.

9. Lines 884-885: “our findings highlight the prevalence of bi-polar POEs, which we discovered to be far more common than previously recognized at imprinted loci”

Previous POE studies have reported a high proportion of bi-polar (complex imprinting) POE pattern <https://www.nature.com/articles/s41467-019-09301-y> and <https://academic.oup.com/hmg/article/27/16/2927/5026426>

Response:

We appreciate the reviewer’s comment and the references provided. Previous studies have indeed identified bi-polar POEs, particularly at DNA methylation sites and for metabolic traits. Our study extends these findings by systematically quantifying the frequency of bi-polar POEs in a genome-wide association study framework. Among the 18 independent POEs identified (i.e., independent SNP-trait pairs), we found that bi-polar POEs were at least as frequent as each type of uniparental POE, with 7 bi-polar effects, 7 maternal effects, and 4 paternal effects.

In contrast, a prior study reported a predominance of uniparental effects, as seen in Partida *et al.* (<https://academic.oup.com/hmg/article/27/16/2927/5026426>), where most identified POEs followed this pattern:

“Most of the loci identified displayed a DNA methylation distribution consistent with uniparental effects, where one of the alleles led to a larger average phenotypic value than the other and one of the chromosomes was putatively silenced.”

Zeng *et al.*, however, reported that *“a large proportion of the identified POE–mQTL associations followed a complex imprinting pattern”*. Indeed, they report that 606 out of the 984 POE-influenced CpG candidates identified exhibited a complex imprinting pattern (polar dominance or bi-polar pattern).

To better acknowledge these prior findings while contextualizing our results, we have modified the discussion to explicitly reference these studies and emphasize that bi-polar POEs are not rare at imprinted loci (line 459):

“Although bi-polar POEs have been previously reported^{6,16}, particularly in regions subject to methylation^{32,33}, our findings further support that such effects are relatively common at imprinted loci.”

10. Line 554-555: “We found two POE associations where genes were located outside known imprinted regions: for ADAM23 and PER3 protein levels”.

POE on PER3 and POE-QTL SNP (both rs228647 and rs228682) regulation for PER3 have been previously reported in methylation study <https://www.nature.com/articles/s41467-019-09301-y>.

Response:

We appreciate the reviewer bringing this to our attention. The cited study indeed reported three CpG sites near PER3 that exhibited POE and further identified POE-mQTLs regulating these sites. Notably, our lead POE-associated SNP was also found to act as a POE-mQTL for these CpG sites, providing an independent line of evidence supporting a potential parent-of-origin effect at this locus.

To acknowledge this, we have now revised our main text to specify that previous work has reported POE-associated DNA methylation patterns at PER3 and that our findings align with these observations (line 498):

“Two additional POE-pQTLs were found for ADAM23 and PER3, genes outside known imprinted regions but previously reported to exhibit paternal-biased expression^{34,35}, supporting potential incomplete or context-dependent imprinting. In addition, despite PER3 being outside known imprinted regions, our lead POE-pQTL was previously reported as POE methylation QTL (POE-mQTL) for PER3³³(Supplementary Note 19)”

This revision strengthens the interpretation of PER3 as a candidate for incomplete or context-dependent imprinting, highlighting the importance of integrating methylation, transcriptomic, and proteomic data when investigating POEs.

Referee #4 (Remarks to the Author):

Hofmeister and colleagues propose a new method for determining parent of origin of alleles in large biobanks where parental genotypes are not available. By utilizing distant relatives (up to the 4th degree) in the UK Biobank and data from the X chromosome, mitochondrial DNA and sibling cross-overs, they were able to accurately infer the parent of origin of alleles. They subsequently performed association analyses with 29 phenotypic traits in the UK Biobank to identify parent of origin effects across the genome. Replication of these parent of origin effects were tested in the Estonian Biobank and the MoBa study. They also utilized the proteomics data in the UK Biobank to identify whether there were protein quantitative trait loci (pQTL) underlying their observed parent of origin effects on the phenotypic traits and to assess whether known pQTLs exhibited parent of origin effects.

This study will be of broad interest due to the novel method for inferring parent of origin of alleles and then the application to identify parent of origin effects on a wide range of traits (which is the largest study of its kind as far as I can tell). The method, which appears to be an extension of the authors previous work, was complex but made the most of the available data in the biobanks to infer parent of origin in as many participants as possible. It was nice to see how well the method performed across the three cohorts that differed substantially in terms of their family relatedness (from one or two distant relatives in the UK Biobank to larger families of closer relatives in the Estonian Biobank and predominantly parent/offspring trios in MoBa). However, the attempt at replicating the association results was a little disappointing as the majority of the phenotypes did not overlap across the cohorts. It was also somewhat surprising that only 30 parent of origin effects were identified across the 29 phenotypic traits, even with the large sample size available in the UK Biobank.

The manuscript was rather lengthy and could have been written more succinctly in places. A few more, relatively minor, aspects that require a little more clarity are mentioned below.

Response:

We sincerely appreciate the reviewer's positive assessment of our method and its broad applicability for studying parent-of-origin effects (POEs) in large-scale biobanks. We are pleased that the reviewer found the methodological approach rigorous and that the performance of our PofO inference was well demonstrated across three cohorts with different family structures.

We acknowledge the reviewer's concern regarding the phenotype overlap across cohorts for replication. Indeed, we were also disappointed by the limited overlap in phenotypic traits between UK Biobank and the Estonian Biobank, which restricted the number of POEs that could be formally tested for replication. Unfortunately, this limitation arises from differences in the available phenotypic data across biobanks, which is outside our control. To maximize replication efforts, we leveraged all available data in EstBB, but we recognize that replication would be more robust with more harmonized phenotypic measurements across cohorts.

Regarding the number of significant POEs identified, we note that POEs generally exhibit smaller effect sizes than additive genetic effects, making them inherently more difficult to detect at a stringent genome-wide significance level. This was supported by our heritability analysis, which confirmed that POE effects are expected to be 10-fold smaller than classical additive effects. While 30 significant POEs may seem modest given the sample size, this number represents a substantial increase compared to previous studies of POEs, and we further validated multiple associations through replication in external cohorts.

Following the reviewer's feedback, we also made substantial efforts to streamline the manuscript. We have now:

- Shortened the Results and Discussion sections by focusing on the most relevant findings.
- Moved more exploratory and less statistically robust results to Supplementary Notes to ensure the main text remains concise and focused.

We hope these revisions enhance the clarity and impact of the manuscript and address the reviewer's concerns.

Specific comments:

1. How does this method compare to the authors previous paper "Parent-of-Origin inference for biobanks"? How often are the PoE alleles the same between the two methods? Given there are several times the previous method is referenced, it would be good to have a formal comparison somewhere in the manuscript (supplementary material is fine).

Response: We appreciate the reviewer's suggestion to formally compare our current method with our previous approach (*Parent-of-Origin Inference for Biobanks, Nature Communication 2022*). In response, we have added **Supplementary Note 2** (referenced in line 133) that outlines

the key methodological differences between the two methods, including the use of phasing sampling, relatedness thresholds, and PofO predictors.

A direct comparison of PofO calls from the two methods is not feasible due to UK Biobank data access restrictions, the previous method was conducted under a different UK Biobank application ID, with data stored on a separate high-performance computing (HPC) server.

However, to address the reviewer's request, we conducted an internal validation by evaluating the agreement between different PofO predictors. Specifically, we reanalyzed our data using only chromosome X, as in the previous study, and compared the results with those obtained using mtDNA and sib-score data in the current approach. This analysis demonstrated 97% concordance between chromosome X-based inference and the new predictors, indicating strong agreement between the methods despite technical differences (see **Supplementary Note 2.3**):

“The previous method inferred PofO using only chromosome X data, whereas the current method integrates additional predictors, including mtDNA and sib-score information. To assess the concordance of these predictors, we reanalyzed our data using only chromosome X to predict the maternal side of surrogate parents, replicating our prior study’s conditions. We present these results below and compare them to our novel approach.

Using the validation cohort, we found that no paternal relatives shared a haplotype segment larger than 11.3cM, and 99% of them shared only segments smaller than 2.8 cM. We therefore used a lenient threshold and defined maternal surrogate parents as sharing a haplotype segment larger than 2.8cM. In our validation cohort, we found that 65.5% of maternal relatives met this criterion.

We next applied this criterion to identify surrogate mothers for the remaining male of the UK Biobank (i.e those not included in the validation cohort). Among the surrogate mothers we could infer using this approach, 5,809 were related to the second-degree, 26,383 third-degree, and 59,610 fourth-degree. For these target-relative pairs, we compared the maternal side inferred from chromosome X, to the parental side inferred using mtDNA MVS. The results showed that chromosome X and mtDNA predictions were concordant in 97.5% of second-degree, 96.7% of third-degree, and 92.0% of fourth-degree pairs.

Additionally, we used chromosome X to identify surrogate mothers for 4,342 individuals who also had at least one sibling. We compared chromosome X-based assignment to the sibling-score approach, and found a 90.0% agreement.

The previous concordance rate compared chromosome X assignment to each of our additional predictors individually. However, our novel approach selects the most accurate predictor at the individual level rather than applying a uniform method to all cases. To directly compare the chromosome X-only approach with our multi-predictor method, we analyzed 45,056 individuals where maternal side was inferred using chromosome X ($cM > 2.8$) and at having at least one additional predictor. For 20,660 individuals, chromosome X was the most accurate predictor and remained the sole determinant of maternal side. For the remaining individuals, chromosome X and the additional predictor indicated the same parental side in 97% of cases. Notably, 56.1% of inconsistencies were attributed to mtDNA minor variant sharing in fourth-degree relatives.”

2. There did not seem to be any information on the phenotypes that were analysed in any of the cohorts. What data cleaning (if any) was performed? Was it the baseline measure that was used

in the UK Biobank or a combination of baseline and follow-up? How was type 2 diabetes defined?

Response:

We acknowledge the reviewer's request for clarification on phenotype selection and processing. We have now added a dedicated section in the Methods (method section "Phenotypes processing", line **853**) detailing the data cleaning and transformation procedures applied to the phenotypic traits.

3. In the results section, it is noted that six significant POEs from the additive analyses were identified (line 370) but Table 1 and Figure 2 only seem to show five (rs2237895 which is associated with HbA1c according to the results section is not presented in Table 1). Would the authors please clarify?

Response:

Thank you for pointing this out. The association of rs2237895 with HbA1c was not included in Table 1 and Figure 2 because it is likely tagging the same signal as rs2299620. These two variants are in low LD ($r^2 = 0.025$), and our conditional analysis suggested that neither is significantly associated with HbA1c conditional on the other variant. Therefore, we cannot distinguish which signal is the likely driver and chose to include rs2299620, which was the lead variant identified in our imprinted region-focused analysis (rs2299620).

To clarify this, we updated our original text, now included in the **Supplementary Note 7** (line **1483**) to explicitly state why rs2237895 was not included in Table 1 and Figure 2. The revised text provides details on the conditional analysis and ensures that the rationale for including only rs2299620 is clearly communicated:

"Conditional analysis suggested that neither variant met the significance threshold independently ($P_{Dc} = 3.5 \times 10^{-4}$ and 4.3×10^{-5} for rs2237895 and rs2299620, respectively), suggesting that they may represent the same association. To avoid redundancy, we only reported rs2299620 in Table 1 and Figure 2, as it is our lead POE-variant identified in the imprinted region-focused analysis."

4. A flow diagram (in the supplementary material) would be useful to show how the samples were removed throughout the analysis pipeline – i.e. there are 274,525 UK Biobank individuals with up to 4th degree relatives, but only 109,385 analysed in most of the phenotype analyses; why were 165,140 individuals removed? Similarly for the replication cohorts.

Response: Thank you for your suggestion. We recognize the importance of providing a clear overview of the sample selection process throughout the analysis pipeline. A similar concern was previously raised by Referee #1-2 (comment #2) and Referee #3 (comment #1), to which we have already responded by adding **Supplementary Figure 15** (referenced in line 126).

This supplementary figure provides a detailed flowchart of the inference process, illustrating:

- Panel A: The number of individuals at each step of the analysis, from the initial identification of individuals with relatives to the final set used in phenotype analyses.
- Panel B & C: The proportion of individuals for whom parent-of-origin inference was successful, depending on the availability of relatives and the type of genetic data used (WGS vs. SNP array).

To further clarify this process in the main text, we have now explicitly described the total number of individuals with PofO inference (286,666), as well as the subset with high-confidence assignments (N=123,716, PofO probability >0.99). Finally, we highlight that the final set of individuals used in phenotype analyses consisted of 109,385 white British individuals after applying ancestry-based restrictions. These clarifications are now included in the main text (section “Parent-of-origin determination for 109,385 white British individuals”, title of the section and lines 118-126).

For the Estonian Biobank, we have in addition added **Supplementary Figure 28**.

We hope this addition provides the necessary clarity and appreciate the reviewer’s feedback on improving the manuscript’s transparency.

5. In the discussion the authors speculate that some reasons for the POEs not replicating in the Estonian Biobank could be due to statistical power, cohort characteristics, or environmental factors. It could also be false positives in UKB (perhaps related to the slightly lower imputation quality)?

Response:

We acknowledge that some of the unreplicated POEs in the Estonian Biobank could be false positives from the UK Biobank. We have now explicitly mentioned this possibility in **Supplementary Note 20** (line 1896):

“Not all POEs identified in the UKBB were replicated in the EstBB, which could reflect differences in statistical power, cohort characteristics, or environmental factors, or false positive associations.”

However, we do not believe that false positives in UKBB are related to lower imputation quality. While the EstBB used an Estonian-specific reference panel, potentially yielding higher imputation accuracy, we applied stringent quality control in both cohorts by filtering variants based on imputation quality scores. Specifically, we excluded variants with an imputation INFO score below 0.8, ensuring that all results are based on high-quality imputed variants. Given this threshold, it is highly unlikely that imputation artifacts contribute significantly to false-positive associations.

Additionally, for imputation errors to induce false positives in POE analyses, they would need to be correlated with either the parent-of-origin of alleles or with phenotypes—an unlikely scenario, as imputation accuracy is primarily influenced by allele frequency and linkage disequilibrium rather than parental origin.

To address this concern, we elaborated on this aspect in **Supplementary Note 14** (line 1712).

6. How does the accuracy of PofO inference (i.e. errors in the PofO assignment) influence the downstream association analyses?

Response:

Similarly to imputation errors, errors in PofO assignment are unlikely to generate false-positive associations but primarily reduce statistical power. Since inference errors are expected to be randomly distributed with respect to the phenotype, they do not introduce systematic biases in POE detection. Instead, such errors dilute true POE signals by misattributing effects across both parental origins, thereby weakening association power. This was also discussed in our previous paper (Hofmeister et al., Nature Communication 2022):

“The presence of errors in the inference is unlikely to produce false positive PofO associations, but only decrease the statistical power of the study, since inference errors are expected to be drawn independently from the phenotypes. Instead, errors are expected to lead to false negatives as PofO signals get diluted onto the two parental origins and thus decrease association power.”

These points have now been explicitly addressed in **Supplementary Note 14**, where we discuss how both PofO inference and imputation accuracy influence our results and their interpretation:

“Errors in PofO inference are unlikely to generate false-positive associations but primarily reduce statistical power. Because PofO errors are expected to be randomly distributed with respect to the phenotype, they should not systematically bias association results. Instead, these errors lead to dilution of true POE signals and reduce the effective sample size, as effects are misattributed to both parental origins, thereby reducing the ability to detect significant associations (see Supplementary Note 3). A similar principle applies to imputation errors: while lower imputation quality may introduce noise and reduce statistical power, stringent quality control filtering (e.g., INFO > 0.8) minimizes this risk, and any remaining imputation inaccuracies are expected to reduce power rather than induce false-positive associations.

Notably, for imputation to introduce systematic errors in POE detection, imputation errors would need to be correlated with either the parent-of-origin of an allele or with the phenotype itself—an unlikely scenario given that imputation accuracy is primarily influenced by allele frequency and linkage disequilibrium rather than parental origin.”

In addition, we have provided a statistical derivation of the drop in statistical power due to PofO inference errors or imputation errors when comparing our current approach to our previous approach in **Supplementary Note 3**:

1201 **Supplementary Note 3 Impact of parent-of-origin inference errors**
1202 **and missingness**

1203 Let X be a random variable taking the value of -1 if the coded allele was inherited from
1204 the father and +1 if the same allele was inherited from the mother, similar to our data
1205 for the differential POE GWAS. In a sample of n heterozygous individuals, we expect to
1206 see the value +1 and -1 half of the time. When we infer this variable, let Z denote the
1207 estimate for the true value X . We are assumed to make mistakes and flip the origin with
1208 probability π , referred to as the error rate. These mistakes happen randomly, in the sense
1209 that it is expected that half of the time we flip a -1 to a +1 and half of the time the opposite.
1210 Thus, $E[X] = E[Z] = 0$. The variance of these random variables is $Var(X) = Var(Z) =$
1211 $E(X^2) - E^2(X) = E(X^2) = 1$. We first compute the correlation between X and Z :

$$\begin{aligned} cor(X, Z) &= \frac{cov(X, Z)}{\sqrt{Var(X) \cdot Var(Z)}} = cov(X, Z) = E((X - E(X)) \cdot (Z - E(Z))) \quad (28) \\ &= E(X \cdot Z) = \pi \cdot (-1) + (1 - \pi) \cdot 1 = 1 - 2\pi \quad (29) \end{aligned}$$

1212 Since the effective sample size equals the actual sample size multiplied by the squared cor-
1213 relation between the true and the noisy variable, with sample size n , at 1% error rate the
1214 effective sample size is $n \cdot (1 - 2 \cdot 0.01)^2 \approx .96 \cdot n$, while at 2% error rate the effective sample
1215 size is $n \cdot (1 - 2 \cdot 0.02)^2 \approx .92 \cdot n$. Thus, if we can increase the sample size by only 5% at
1216 the cost of increasing the error rate from 1% to 2%, we already increase statistical power to
1217 detect true associations.

7. I'm struggling to understand how equations 1 and 2 were derived. As these equations are key to understanding the method, would the authors please provide an intuition behind the equations, or a little more of the derivation, to help the reader? Additionally, lines 1009-1015 the authors talk about test and training sets – can they please define what these are?

Response:

We sincerely appreciate the reviewer's feedback regarding the clarity of Equations (1) and (2). This section is indeed crucial for understanding our method, and we acknowledge that it was not as intuitive as it could have been.

In response to the reviewer's comment, we have made two key improvements:

1. Correction of notation typos:
We identified and corrected misleading notation where "pat" was mistakenly used instead of "mat" in some instances. This correction ensures that the equations correctly reflect the intended definitions of maternal and paternal probability assignments.
2. Improved explanation with step-by-step breakdown:
To enhance readability and comprehension, we now explicitly decompose the main equation into its constituent terms, providing a clearer step-by-step explanation of each component. We define each term in words, explaining how it contributes to the probability estimation of a surrogate parent being on the maternal or paternal side.

These modifications (lines 610-640) should significantly improve the clarity of this section and make the methodology more accessible to readers. We thank the reviewer for pointing out the

difficulty in understanding these equations, as it prompted us to refine this explanation and improve the manuscript's readability.

8. Why was only BMI across early life chosen as the phenotype for replication in MoBa and not height? Especially given none of the PoEs identified were associated with adult BMI but there were several associated with adult height.

Response:

We initially focused on BMI in MoBa due to limited data access at the time of our original analysis. However, we have now extended our analysis to include height across early life in the MoBa cohort. We have updated the Results section accordingly and revised the corresponding figure to reflect these new findings, which we now included as main figure 7.

9. The numbers are a little exaggerated in places. For example, the title claims that evidence of PoE is 'from ~265,000 individuals', whereas the actual numbers of individuals within each of the cohorts that have phenotype data are lower (~109K for UK Biobank, max N=85050 in Estonian Biobank and max N=42346 in MoBa). In the abstract 'more than 60 complex traits and over 2,400 protein levels' are investigated, but in reality, 59 traits and only previously associated proteins are tested. It would be worthwhile stating these more accurately.

Response:

Thank you for your careful review. We have now corrected the reported numbers to ensure accuracy.

- For the UK Biobank, we adjusted the reported sample size to 109,385 individuals, reflecting the restriction to white British ancestry.
- For the Estonian Biobank, we corrected the number for "up to 85,050 individuals", which represents the maximum number of individuals we tested in GWAS.
- Similarly, in the MoBa cohort, we corrected the number for "up to 42,346 individuals".

Considering these revised numbers, our study included evidence from up to 236,781 individuals, and we updated the title accordingly.

Regarding the number of traits analyzed, our initial analysis was indeed focused on 59 traits. Although our follow-up analysis included three additional traits (sitting height to evaluate standing height POE-variants, and two comparative traits at age 10), we have now refined the text to consistently state 59 traits for clarity.

Additionally, we have updated the number of our pQTL analysis. Instead of referring to "over 2,400 protein levels" we now specify that we tested more than 14,000 pQTLs, providing a more precise representation of our analysis.

Referee #4 (Remarks on code availability):

I have not run the code, but the author's website provides detailed instructions for setting up and running their software.

Response to referees

We thank the referees for taking the time to review our revised manuscript and for their feedback. Below, we provide a point-by-point response to each of the remaining comments.

Referee #2 (Remarks to the Author):

We note that this report is prepared by two people. Overall, we are satisfied with the revision of the manuscript and the responses of the authors to our comments.

Referee #2 (Remarks on code availability):

We successfully installed and compiled THORIN and were able to run the toy example without any issues. The installation instructions are clearly outlined and easy to follow.

Referee #3 (Remarks to the Author):

The authors have addressed my comments well. In some parts of the revised manuscript (mainly supplementary) there are mystery question marks, such as "Correlated these LD estimates at the 11p15.5 imprinted region (Supplementary Figure ??A)."

Thank you for pointing this out. The issue arose because the supplementary information was compiled separately from the main text, during which references to elements such as main figures were not resolved correctly. When the full manuscript is compiled as a single document, these references appear as intended. We will ensure that all references are correctly resolved in the final submission.

Referee #3 (Remarks on code availability):

I didn't run the code. The authors have provided their tool in <https://rjhfmr.github.io/THORIN/> and the instructions seem to be clear to me.

Referee #4 (Remarks to the Author):

Many thanks to the authors for their thorough responses to my previous concerns. With the substantial changes, particularly to the results and discussion sections, I think the manuscript is a lot easier to digest and the key messages are more prominent. I have a couple of very minor questions:

1. Thank you for including Sup Fig 15, I think this makes the selection of UKB participants throughout the pipeline much clearer. However, a couple of the numbers don't seem to add up; individuals with at least one sibling (N=14,695) is less than the number of PofO assigned from intra-chrs sib score (N=14596) and the PoO un-assigned (N=315). Would you please be able to clarify?

Thank you for catching this. The numbers were correctly reported in the main text, but an error occurred during the preparation of Supplementary Figure. Upon verification, the correct number of unique individuals with unassigned parent-of-origin is $N = 99$, corresponding to those with an absolute score below 2 on all 22 autosomes. We have updated the figure accordingly to reflect the correct counts.

2. Lines 254-257: It was nice to see the ORs for the maternally and paternally inherited alleles on T2D, as it shows the size of the effect and can be more easily compared to the additive effects from previous GWAS. I wonder whether it would be worthwhile including the effect size estimates for the other described associations in the previous three sections (growth and metabolic traits/lipid metabolism and hip circumference/fat distribution)?

We believe that ORs are appropriate for case-control associations, but for quantitative traits, reporting beta and standard error estimates is more suitable. In the main text, we chose not to include these estimates for all associations, as doing so would substantially lengthen the

manuscript (and remove focus) —each association involves six values (paternal, maternal, and differential betas and SEs). Instead, we provide all effect size estimates in Table 1 and the Supplementary Note. We believe this approach maintains readability and ensures the main text remains focused and accessible.

3. The legend for fig. 7 states that 7a) is for BMI and 7b) is of height, but the actual plot seems to be the other way around.

Thank you for pointing this out, we corrected it.

4. The shortened discussion is much more concise; however, I feel that a concluding statement is missing and think it would be valuable for readers to get one or two of take-home messages from the manuscript.

Thank you for the suggestion. We have added a brief concluding paragraph at the end of the discussion to highlight key take-home messages from the study.

5. Some of the supplementary figures are not correctly referenced. For example, line 643 has “(Supplementary Figure ??A)”. Same in the supplementary notes.

Thank you for pointing this out. The issue arose because the supplementary information was compiled separately from the main text, during which references to elements such as main figures were not resolved correctly. When the full manuscript is compiled as a single document, these references appear as intended. We will ensure that all references are correctly resolved in the final submission.